# A bacterial CARD–NLR-like immune system controls the release of gene transfer agents

Emma J. Banks [1] ✉, Pavol Bárdy [2], Ngat T. Tran[1], Phuong M. Nguyen [2], Boris Stojilković [3], Kevin Gozzi[4], Abbas Maqbool[5] & Tung B. K. Le [1] ✉

Bacteria use immune systems to detect and defend against mobile genetic elements including phages. Gene transfer agents (GTAs) are domesticated prophages with phage-like characteristics including the ability to induce host cell lysis for gene transfer. Whether GTAs elicit or avoid bacterial immune systems is poorly understood. Here, a transposon mutagenesis with deep sequencing screen in *Caulobacter crescentus* identified a tripartite system, LypABC, essential for GTA-mediated cell lysis and gene transfer. LypABC resembles a caspase recruitment domain–nucleotide-binding leucine-rich repeat (CARD–NLR) anti-phage defence system. LypABC is dispensable for DNA packaging into GTA particles but required for host cell lysis, involving the peptidase domains of LypA and LypC, and the ATPase domain of LypB. As LypABC overproduction is toxic, strict regulation through the transcriptional repressor CdxB is required. CdxB binds the promoters of *lypABC* and of essential GTA activator genes, coupling GTA activation to host cell lysis. Our findings suggest that bacterial immune systems can be co-opted to support horizontal gene transfer by GTAs.

Mobile genetic elements (MGEs), such as bacteriophages, plasmids and transposons, are double-edged swords; while they can confer certain adaptive advantages to their host through horizontal gene transfer, they often act selfishly, exploiting the host for their own propagation[1]. Bacteria are therefore engaged in an arms race against MGEs and have evolved an extraordinary diversity of immune systems to detect and defend against MGEs, including 150 distinct anti-MGE systems that have been identified in recent years[2–8]. Although immune systems are traditionally considered antagonistic to MGEs, it remains unclear whether some immune systems might be versatile and, in certain contexts, may facilitate rather than prevent MGE propagation.

Amid the constant conflict between bacteria and phages, GTAs are exceptions. GTAs are selfless virus-like MGEs that have been domesticated from ancient viruses to provide beneficial functions to their hosts[9–11]. GTAs[9,12–14] are encoded by gene clusters within a wide variety of bacterial and archaeal genomes, and are deeply integrated with their host physiology[15,16]. GTAs transition through a series of life stages: GTA gene cluster activation[17], GTA particle assembly[18], non-selective

encapsulation of host DNA into GTA particles[19,20], GTA particle release by host cell lysis[21,22] and, finally, transfer of host DNA into recipient bacteria[18,23,24]. Most notably, unlike bacteriophages, GTA capsid heads are too small to package complete GTA gene clusters (for example, the *Caulobacter crescentus* GTA can package only ~8.3 kb of DNA yet its encoding GTA cluster is >15 kb (ref. [25])). Consequently, GTAs are unable to self-multiply and be infectious[26]. Despite GTA domestication, the phage-like origin and appearance of GTAs—together with life stages that include host cell lysis—are factors that may inadvertently trigger host immunity. It remains unclear how GTAs might avoid, subvert or even adopt host immune systems to complete their life stages.

Here, by studying GTA-mediated host cell lysis in *C. crescentus*, we identified a potential bacterial immune system, LypABC, that has been adopted to control the release of GTA particles. LypABC resemble components of CARD–NLR anti-phage defence systems[27]. First described in *Lysobacter enzymogenes*, CARD–NLR immunity occurs through abortive infection that involves sacrificial death of phage-infected cells, preventing the release of mature phage particles

[1]Department of Molecular Microbiology, John Innes Centre, Norwich, UK. [2]York Structural Biology Laboratory, University of York, York, UK. [3]Department of Cell and Developmental Biology, John Innes Centre, Norwich, UK. [4]The Rowland Institute at Harvard, Harvard University, Cambridge, MA, USA. [5]Department of Biochemistry and Metabolism, John Innes Centre, Norwich, UK. ✉e-mail: emma.banks@jic.ac.uk; tung.le@jic.ac.uk

and thereby curbing infection[27]. The *L. enzymogenes* CARD–NLR system senses phage infection, somehow activating a CARD-containing protein component, which interacts with an NLR-like protein[27]. Cell death occurs through proteolysis-based activation of a gasdermin effector, which directly causes cell lysis by forming membrane pores and permeabilizing the cell membrane[28,29]. In animals, many NLR-based inflammatory responses also contain CARD components that signal to caspases[30,31], which then proteolytically cleave and activate gasdermin effectors, leading to the release of pro-inflammatory cytokines and cell death[32–34]. Here we find that predicted anti-phage defence domains of LypABC are essential for cell lysis. We further show that LypABC specifically mediates cell lysis for GTA release, but is dispensable for DNA packaging into GTA particles. Overproduction of LypABC is highly toxic to both GTA-producing and non-producing cells, highlighting the need for this system to be tightly regulated. Lastly, we identify a transcriptional regulator, CdxB, that directly represses genes encoding GTA-activating factors and LypABC, thereby coupling GTA gene cluster activation and host cell lysis. In summary, we have identified a CARD–NLR-like system that may benefit MGEs and promote horizontal gene transfer.

## Results

### GTA-mediated host cell lysis results in ghost cell formation

*C. crescentus* GTA synthesis is repressed under standard laboratory conditions but can be activated by deleting the master repressor gene, *rogA* (ref. 25; Fig. 1a). This relieves RogA-mediated repression of the *gafYZ* operon, which is essential for GTA activation[25]. The transcriptional activator GafY, together with integration host factor (IHF), co-activates the expression of GTA gene clusters and accessory genes elsewhere on the chromosome. Meanwhile, GafZ enables RNA polymerase to bypass internal transcription terminators within the core GTA gene cluster, ensuring complete expression of an entire biosynthetic gene cluster[25,35,36] (Fig. 1a).

To investigate the consequences of GTA activation and how this leads to host cell lysis, we observed wild-type (GTA-off) and Δ*rogA* (GTA-on) *C. crescentus* strains during stationary phase by phase-contrast microscopy (Fig. 1b). While the wild-type strain comprised almost entirely phase-dark cells with only 0.1 ± 0.1% phase-light cells (that is, ghost remnants of lysed cells), the Δ*rogA* mutant population was heterogeneous, consisting of a mixture of phase-dark cells and a substantially higher proportion of phase-light ghost cells (51.6 ± 2.5%) than the wild-type strain. Complementation of the Δ*rogA* mutant restored the wild-type phenotype (0.5 ± 0.7% ghost cells), while deletion of the core GTA gene cluster in the Δ*rogA* background (Δ*rogA*Δ*GTA*) completely eliminated ghost cells (0.3 ± 0.4%), indicating that GTA activation is responsible for ghost cell formation (Fig. 1b).

Next, to monitor GTA activity from cluster activation to host cell lysis at the single-cell level, we engineered a fluorescent reporter by transcriptionally fusing the promoter of the core GTA cluster (P*gtaT*) to mNeonGreen (mNG) and integrated this construct ectopically at the vanillate-utilization (*vanA*) locus (Fig. 1c). As expected, no mNG-fluorescent cells were observed in the wild-type strain. By contrast, the Δ*rogA* mutant showed a mixture of mNG-fluorescent cells (30.4 ± 3.0%) and non-fluorescent cells (of which 21.5 ± 3.9% were phase-dark and 48.2 ± 2.3% were phase-light ghost cells) (Fig. 1d). Subsequent time-lapse microscopy revealed the emergence of an mNG signal, followed by cell death within ~90 min. Cell death typically involved the loss of cytoplasmic mNG signal, cell pole contraction and then a transition from a phase-dark to a phase-light ghost cell state—all within ~10 min (Fig. 1e, Extended Data Fig. 1a and Supplementary Videos 1–3). Notably, ghost cells maintained their vibrioid morphology, contrasting with modes of phage holin–endolysin-mediated lysis that often involve outer membrane blebbing and morphological deformation, culminating in explosive cell lysis[37–41]. Altogether, these observations highlight the heterogeneous nature of GTA production in *C. crescentus*

and suggest that GTAs may cause host cell lysis via a mechanism distinct from classical holin–endolysin pathways.

### LypABC are essential for GTA-mediated host lysis and gene transfer

Consistent with the observed non-explosive GTA-mediated lysis phenotype, by bioinformatic searches, we found no homologues of canonical holin–endolysin-encoding genes within the core GTA cluster or elsewhere on the *C. crescentus* genome. To identify candidate lysis genes in an unbiased manner, we conducted saturated transposon mutagenesis combined with deep sequencing (Tn-seq), comparing transposon insertion frequencies between wild-type (GTA-off) and Δ*rogA* (GTA-on) strains (Fig. 2a and Supplementary Table 1). We reasoned that transposon insertions disrupting either GTA activator-encoding genes or genes required for host cell lysis would prevent lysis, leading to a higher recovery of DNA for deep sequencing. Accordingly, we anticipated a higher frequency of insertions within genes required for GTA activation or cell lysis in Δ*rogA* compared with the wild-type background.

As expected, we found genes encoding known GTA activators, such as GafY, GafZ, and the α and β subunits of IHF, which showed 7.8-fold, 6.6-fold, 8.3-fold and 4.9-fold increases in transposon insertions, respectively, confirming that the Tn-seq experiments worked (Extended Data Fig. 2a). Differential analysis of the data identified 41 genes with significantly enriched Tn insertions in the Δ*rogA* background (log$_2$(fold change) ≥ 1.0, adjusted *P* < 0.01) (Fig. 2b; see also Extended Data Fig. 2b for Gene Ontology (GO) analysis). Among these, we identified an operon comprising the three genes *CCNA_03886*, *CCNA_00580* and *CCNA_00579* (renamed to *lypABC*, respectively, for putative lysis proteins ABC) that contained higher insertion frequencies in the Δ*rogA* background compared with the wild-type strain (4.2-fold, 3.9-fold and 5.0-fold, respectively) (Fig. 2b,c). This 6.9-kb operon, located ~2.4 Mb away from the GTA gene cluster, is predicted by the bioinformatic tool DefenseFinder[42] to encode an immune system resembling a CARD–NLR anti-phage defence complex. Deletion of either the entire *lypABC* operon or individual *lyp* genes in the Δ*rogA* background (GTA-on) completely abolished ghost cell formation (Fig. 2d). Meanwhile, complementation of each Δ*rogA*Δ*lyp* mutant by expressing wild-type gene copies from an ectopic xylose-utilization (*xylX*) locus restored the lytic phenotype (Fig. 2d).

To test whether the absence of *lypABC*—and thus cell lysis—abrogates GTA-mediated gene transfer, we performed a gene transfer assay, measuring transduction of a tetracycline resistance marker from different donor strains to a kanamycin-resistant recipient strain. We used a xylose-inducible *gafYZ* overproducer strain (pBXMCS-6::P$_{xyl}$-*gafYZ*) as the donor as this strain produces far more GTA-lysing cells (~70–80%) than a Δ*rogA* mutant. In contrast to the lysis-competent P$_{xyl}$-*gafYZ* donor strain, which had a transduction rate of ~2.35 × 10⁻⁶ per cell, the lysis-incompetent Δ*lypABC* P$_{xyl}$-*gafYZ* donor was incapable of gene transfer, equivalent to the GTA-off P$_{xyl}$-empty vector negative control donor strain (Fig. 2e). These data show that all three *lyp* genes are required for GTA-mediated host cell lysis and consequent transfer of DNA to recipient cells.

### LypABC are essential for GTA release but not DNA packaging or cell death

To investigate whether LypABC are involved in the production of DNA-packaged GTA particles within *C. crescentus* cells, we assayed for the presence of an ~8.3-kb DNA band in total DNA extractions[25]. The DNA band was visible in the Δ*rogA*Δ*lypABC* mutant as well as in each of the individual Δ*rogA*Δ*lyp* gene mutants, indicating that LypABC are not required for the encapsulation of host genomic DNA into GTA particles (Fig. 3a). To determine whether GTA particles are produced in the absence of *lypABC*, we performed immunoblotting using an antibody against the GTA head–tail connector structural protein, GtaL[35]. Although GtaL was detected in both the intracellular and

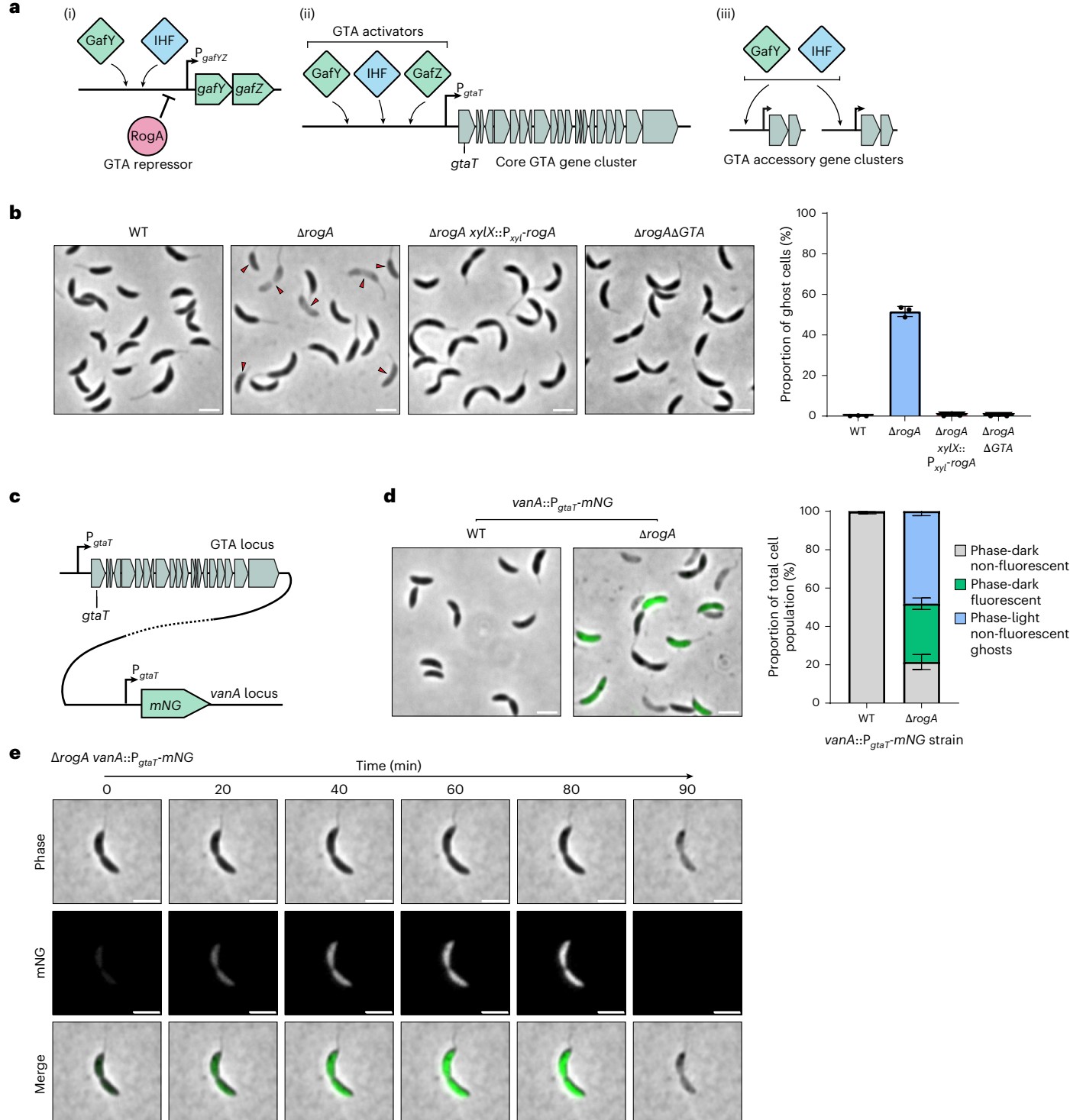

**Fig. 1 | GTA-mediated host cell lysis results in ghost cell formation.**
**a**, Schematic of GTA regulation in *C. crescentus*. (i) GTA-activating proteins, GafY and IHF, activate *gafYZ* transcription; however, RogA can repress transcription of *gafYZ*. (ii) GafY, IHF and GafZ bind to the core GTA gene cluster promoter, P$_{gtaT}$, and activate cluster transcription, resulting in GTA activation. (iii) GafY and IHF also activate transcription of accessory GTA gene clusters at different chromosomal loci. **b**, Left: phase-contrast microscopy of *C. crescentus* strains. The Δ*rogA* mutant was complemented by expression of *rogA* from the *C. crescentus* chromosomal *xylX* locus promoter (P$_{xyl}$). Images are representative of three independent repeats. Red arrows indicate ghost cells. Scale bars, 2 μm. Right: quantification of ghost cells as a proportion of the total population. Data represent the mean ± s.d. from three independent experiments (*n* = 400 cells analysed per experiment). **c**, Schematic of strain construction for an mNG reporter for GTA activation. The core GTA gene cluster was unaltered, and a

copy of the P$_{gtaT}$ promoter was inserted at the neutral *vanA* locus, immediately followed by the *mNG* gene. The dashed line indicates the large genomic distance separating the two loci. **d**, Left: fluorescence microscopy images of wild-type (WT) and Δ*rogA* strains containing the *vanA*::P$_{gtaT}$-*mNG* reporter fusion. Images are representative of three independent repeats. Scale bars, 2 μm. Right: quantification of three different cell types within each population: phase-dark non-fluorescent cells, phase-dark fluorescent cells and phase-light non-fluorescent ghost cells. Data represent the mean ± s.d. from three independent experiments (*n* = 400 cells analysed per experiment). **e**, Fluorescence microscopy images from a time-lapse with the Δ*rogA vanA*::P$_{gtaT}$-*mNG* strain showing GTA activation (mNG signal) followed by cell lysis (Supplementary Video 1). For further examples, see Extended Data Fig. 1a and Supplementary Videos 2 and 3. Data are representative of at least three independent repeats. Scale bars, 2 μm.

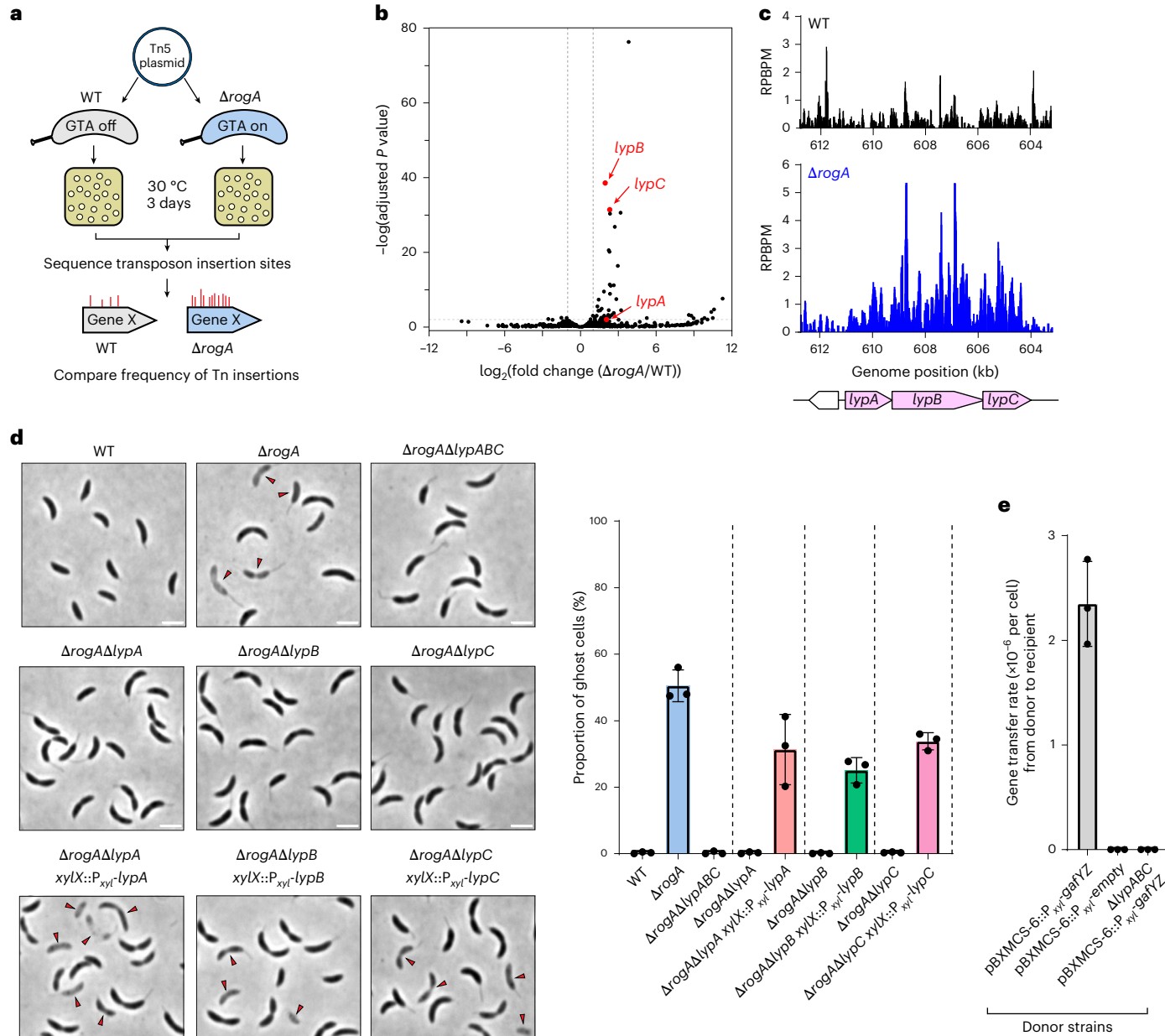

**Fig. 2 | LypABC are essential for GTA-mediated host lysis and gene transfer.**
**a**, Schematic of the Tn-seq method to identify lysis genes. **b**, Tn-seq volcano plot
showing the log₂-transformed fold change of transposon insertions comparing
Δ*rogA* with the WT. Grey lines indicate a two-fold change threshold. *lypA*, *lypB*
and *lypC*, which contain more transposons in Δ*rogA*, are annotated. DESeq2 was
used for differential expression analysis with default settings (Wald test followed
by *P* value adjustment for multiple comparisons using the Benjamini–Hochberg
method). Three independent repeats were performed. **c**, Plots showing the reads
per base pair per million mapped reads (RPBPM) at the *lypABC* locus, indicating
the higher frequency of transposon insertions throughout the *lypABC* genes
within Δ*rogA* compared with the WT. **d**, Left: phase-contrast microscopy images
of each Δ*rogA*Δ*lyp* gene deletion mutant. Mutants were complemented by
expressing the appropriate *lyp* gene from the *C. crescentus* chromosomal *xylX*

locus promoter (P*xyl*). Images are representative of three independent repeats.
Red arrows indicate ghost cells. Scale bars, 2 μm. Right: quantification of ghost
cells as a proportion of the total population. Data represent the mean ± s.d.
from three independent experiments (*n* = 400 cells analysed per experiment).
**e**, Gene transfer assay measuring GTA transduction of a chromosomally encoded
tetracycline resistance cassette from different *C. crescentus* donor strains into
a kanamycin-resistant recipient strain. Donor strains contain the replicative
plasmid pBXMCS-6 encoding either *gafYZ* (to activate GTA expression) or
nothing (empty vector control). The transfer rate was calculated by dividing
doubly antibiotic-resistant colony-forming units per ml (CFU ml⁻¹) by the
total recipient CFU ml⁻¹. Data show the mean ± s.d. from three independent
experiments.

extracellular fractions of the Δ*rogA* mutant, consistent with particle
release via cell lysis, GtaL was present only in the intracellular fraction
of the Δ*rogA*Δ*lypABC* mutant, suggesting that GTA particles assemble
but are not released from cells to the extracellular space (Fig. 3b). We
further visualized the formation and release of GTA particles directly
using cryo-electron tomography. GTA capsids were present in both

Δ*rogA* and Δ*rogA*Δ*lypABC* cells; however, membrane collapse and sub-
sequent cell lysis to release GTA particles occurred only in Δ*rogA* cells
(Fig. 3c and Supplementary Fig. 1), suggesting that LypABC are required
for host lysis but not for upstream intracellular particle assembly.
     To investigate whether GTA-producing cells lacking LypABC are
viable, we used dual labelling with propidium iodide, a fluorescent DNA

stain that can enter only membrane-compromised dead cells, and the P$_{gtaT}$-mNG reporter. Wild-type cells were phase-dark with no detectable mNG or propidium iodide signal (Fig. 3d) and, as observed previously (Fig. 1d), the Δ*rogA* population was heterogeneous, containing both mNG-fluorescent and non-fluorescent phase-dark cells in addition to non-fluorescent ghost cells. Weak propidium iodide signal was visible in most but not all Δ*rogA* ghost cells (36.3 ± 6.9%); propidium iodide fluoresces following DNA binding; thus, this weak propidium iodide signal may be due to the release of DNA-packed GTA particles. In the non-lysing Δ*rogA*Δ*lypABC* mutant, 85.7 ± 0.9% of all cells contained the mNG signal, confirming that GTA cluster expression occurs independently of LypABC (Fig. 3d). Importantly, 23.0 ± 8.5% of all cells shared both mNG and propidium iodide signal (Fig. 3d), indicating that these GTA-producing cells were non-viable despite the absence of visible lysis. This conclusion was further corroborated by observing cells growing on propidium iodide-supplemented agarose pads by time-lapse microscopy (Extended Data Fig. 1b and Supplementary Videos 4 and 5). Similar to Δ*rogA* cells (Fig. 1e), in Δ*rogA*Δ*lypABC* cells, the mNG signal appeared first, followed by cell pole contraction (Extended Data Fig. 1b and Supplementary Videos 4 and 5). However, unlike the Δ*rogA* strain, cells did not become phase-light ghost cells. Instead, the mNG signal was gradually replaced by the propidium iodide signal over ~500 min, suggesting that cells die in the absence of visible lysis (probably owing to host genome digestion and packaging into GTA capsids). Thus far, the collective evidence suggests that LypABC are essential for host cell lysis to release GTA particles, but are dispensable for GTA gene expression, DNA packaging and GTA-mediated cell death.

## LypABC resemble a CARD–NLR anti-phage defence system

Bioinformatic analysis by DefenseFinder[42] suggested that LypABC architecturally resemble a CARD–NLR anti-phage immune system (Fig. 4a). First identified in *L. enzymogenes*, CARD–NLR systems mediate abortive infection by sensing phage invasion and then activating a death-domain effector (for example, a pore-forming gasdermin protein in *L. enzymogenes*), leading to cell death and infection containment[27].

Foldseek-based[43] searches for structural homologues of LypA and LypB returned known CARD–NLR proteins including the trypsin-like serine peptidase and NLR-like protein components of the *L. enzymogenes* and *Azospirillum* sp. CARD–NLR systems[27], while LypC returned trypsin-like serine peptidase and endonuclease homologues (Fig. 4b). Moreover, AlphaFold3-generated models of LypA and LypB superimpose closely to their *L. enzymogenes* homologues (Extended Data Fig. 3a,b), and a phylogenetic tree built from the ATPase domains of CARD–NLR, bNACHT NLR, SWACOS and MalT-family proteins placed LypB within a distinct CARD–NLR clade (Fig. 4c). These data suggest that LypABC belong to the CARD–NLR protein family.

LypA contains three predicted domains: an N-terminal α-helical bundle domain (amino acids (aa) 1–80), a middle trypsin-like serine peptidase domain (aa 85–305) and a C-terminal ATPase domain (aa 324–592). The predicted N-terminal domain shares structural similarity with eukaryotic caspase recruitment domains (CARDs)[27] and

prokaryotic CARDs found in *L. enzymogenes* and *Azospirillum* sp. (Extended Data Fig. 3c). In *L. enzymogenes*, the CARD is suggested to activate a gasdermin effector protein that causes cell lysis[27]. To assess the relevance of the CARD-like domain in LypA, we generated a CARD-truncated *lypA* mutant; however, the resulting truncated protein was unstable in vivo (Extended Data Fig. 4a), preventing us from drawing further conclusions. Next, we assessed the role of the trypsin-like serine peptidase domain of LypA, discovering that expressing a *lypA* catalytic serine residue mutant (S262A) did not restore lysis in the Δ*rogA*Δ*lypA* background (Fig. 4d) despite robust protein production (Extended Data Fig. 4a). This indicates that LypA peptidase activity is critical for LypABC-mediated lysis. The C-terminal ATPase domain of LypA lacks conserved Walker A/B residues (Extended Data Fig. 5b), yet AlphaFold3 produced a high-confidence model of LypA bound to ATP and Mg$^{2+}$ (interface predicted template modelling (ipTM) score: 0.96; Extended Data Fig. 6a), suggestive of ATP binding. Deletion of the ATPase domain resulted in an unstable truncated protein (Extended Data Fig. 4a). We therefore identified three residues in the putative ATP-binding pocket that are predicted to form contacts with ATP and mutated these to alanine. This *lypA* triple mutant (T356A, S360A, F361A) was stably produced in vivo (Extended Data Fig. 4a) but failed to restore lysis in the Δ*rogA*Δ*lypA* mutant (Fig. 4d), suggesting that the divergent LypA ATPase domain might bind ATP in vivo, and this activity is essential for LypABC-mediated cell lysis.

LypB contains a predicted Orc1-like AAA+ ATPase domain (aa 193–392) containing conserved Walker A (Gx$_4$GKS) and Walker B (MVLD) motifs, followed by extended tandem tetratricopeptide (TPR)-like repeats (aa 393–771) (Fig. 4a and Extended Data Figs. 3b, 5b and 6b). This architecture is reminiscent of NLR-related proteins such as APAF1 (ref. 44), which may substitute the leucine-rich repeats of canonical NLRs for alternatives such as TPR or WD40 repeats[45]. In addition to Foldseek (Fig. 4b), HHPred searches further identified known bacterial NLR proteins as homologues of LypB including MalT from *Escherichia coli*[46] and Avs3 from *Salmonella enterica*[47]. NLR ATPase domains typically bind ATP and oligomerize into a large macromolecular structure[48]. Expressing *lypB* variants with point mutations in the Walker A (that is, K230A) and Walker B (that is, D335A) motifs failed to complement the Δ*rogA*Δ*lypB* mutant phenotype (Fig. 4d) despite wild-type levels of protein production (Extended Data Fig. 4b), suggesting that LypB ATPase activity is essential for LypABC-mediated host cell lysis.

Like the *L. enzymogenes* anti-phage system[27], AlphaFold3-Multimer predicted an interaction between the CARD domain of LypA and the N-terminal domain of LypB (ipTM score: 0.85; Extended Data Fig. 7a). Moreover, the LypA CARD was predicted to self-interact (Extended Data Fig. 7b), which is consistent with homomeric interactions observed for eukaryotic CARDs[49–51]. Bacterial two-hybrid assays confirmed that full-length LypA and LypB interact exclusively with each other and not with LypC (Extended Data Fig. 7c). Systematic pairwise testing of all possible LypA, LypB and LypC domain combinations revealed weak self-interactions for the LypB N-terminal domain and LypC endonuclease domain and validated

**Fig. 3 | LypABC are essential for GTA release but not DNA packaging or cell death. a**, Total genomic DNA extractions from *C. crescentus* strains visualized by agarose gel electrophoresis. GTA packaging of bacterial host DNA is determined by the presence of an 8.3-kb band. A representative gel from two independent experiments is presented. EtBr, ethidium bromide. **b**, Immunoblots of intracellular and extracellular supernatant fractions from *C. crescentus* strains using a polyclonal antibody targeting the GTA structural head–tail connector protein, GtaL. The immunoblot is representative of two independent experiments. **c**, Cryo-electron tomograms of Δ*rogA* and Δ*rogA*Δ*lypABC* cells with segmented tomograms for each image. The red arrows indicate GTA capsid heads; the blue arrow indicates a collapsed inner membrane; the yellow arrow indicates an intact inner membrane. Segmentation labels: inner membrane, blue; outer membrane, cyan; S-layer, green; ribosomes, orange; expanded capsids,

magenta; prohead capsids, yellow; phosphate granule, grey. Scale bars, 100 nm. Images are representative of two independent experiments. Additional images are shown in Supplementary Fig. 1. **d**, Left: fluorescence microscopy images of *C. crescentus* strains containing a *vanA*::P$_{gtaT}$-*mNG* fluorescent reporter for GTA activation. Cells were incubated with the dead cell-permeable dye propidium iodide (PI) and visualized on agarose pads. Images are representative of three independent repeats. Red arrows indicate ghost cells. Scale bars, 2 μm. Time-lapses of the Δ*rogA*Δ*lypABC vanA*::P$_{gtaT}$-*mNG* strain grown in the presence of propidium iodide can be viewed in Extended Data Fig. 1b and Supplementary Videos 4 and 5. Right: quantification of four different cell types within each population: non-fluorescent cells, mNG-fluorescent-only cells, mNG- and PI-fluorescent cells and PI-fluorescent-only cells. Data represent the mean ± s.d. from three independent experiments (*n* = 400 cells analysed per experiment).

that the LypA–LypB interaction occurs specifically between the LypA CARD and the LypB N-terminal domain (Extended Data Fig. 7d). To assess the functional relevance of a LypA–LypB interaction in vivo, we expressed a variant of *lypB* lacking the N-terminal domain in the

Δ*rogA*Δ*lypB* mutant. The truncated protein was stably produced (Extended Data Fig. 4b) but failed to complement the lysis defect (Fig. 4d), indicating that the LypA–LypB interaction is essential for LypABC-mediated cell lysis.

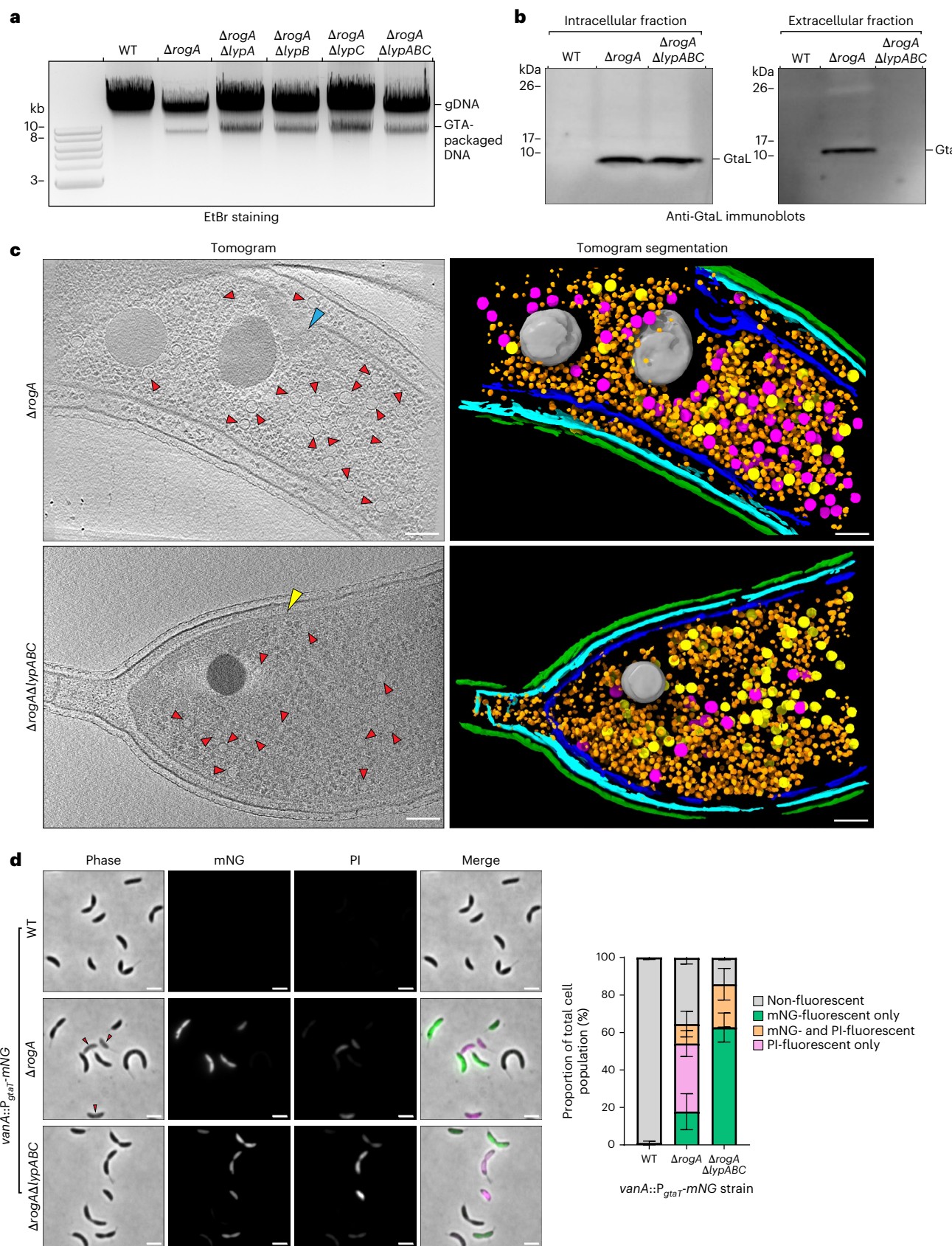

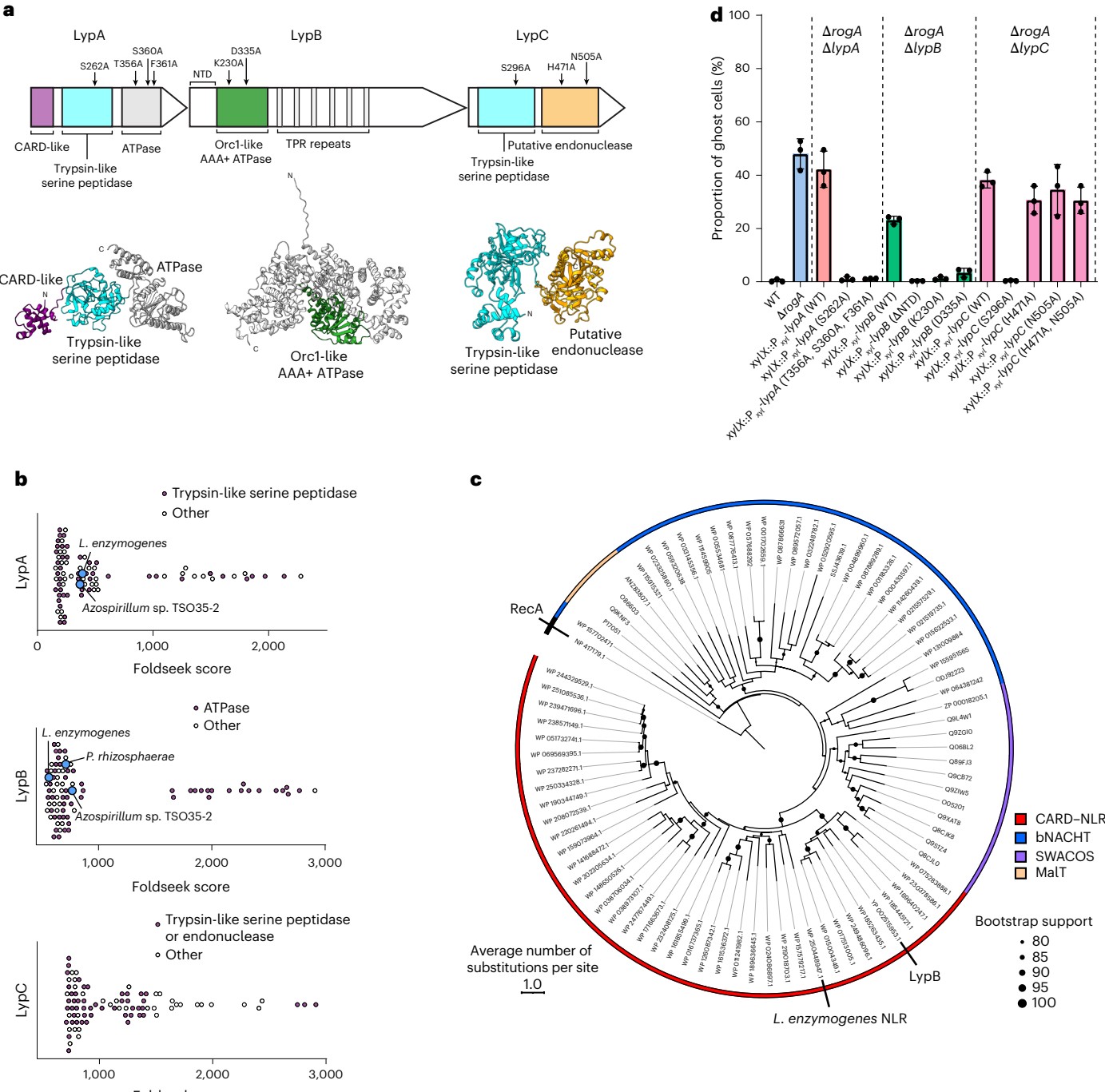

**Fig. 4 | LypABC resemble a CARD–NLR anti-phage defence system.**
**a**, Schematics of the predicted domain organization for LypABC (top) with AlphaFold3-predicted structures (bottom). **b**, AlphaFold-predicted structural homologues of LypA, LypB and LypC identified by Foldseek with a query coverage >80% and $E$ value < 5.88 × 10$^{-9}$, sorted according to the Foldseek score. Blue circles represent characterized CARD–NLR homologues; purple circles represent homologues sharing the same predicted catalytic domain as the query; white circles represent homologues with a different domain annotation or annotated as uncharacterized ('other'). **c**, Maximum likelihood phylogenetic tree constructed from the ATPase domains of predicted CARD–NLR proteins annotated by DefenseFinder (red clade), experimentally validated bNACHT

proteins (blue clade), and SWACOS (purple) and MalT-family (orange) STAND ATPases. The tree is rooted on the ATPase RecA sequence. Bootstrap support values >80% are presented. LypB clusters within the CARD–NLR clade. **d**, Quantification of ghost cells as a proportion of the total population within different ΔrogAΔlyp mutants complemented with either the WT lyp gene, lyp genes encoding point mutation(s) of predicted catalytic amino acid(s) or a lyp gene containing a truncation of the N-terminal domain (NTD). Locations of amino acid mutations are indicated by arrows above the LypABC protein schematics in **a**. Means ± s.d. of three independent repeats are shown (n = 400 cells analysed per repeat).

LypC is predicted to contain an N-terminal trypsin-like serine peptidase domain (aa 1–335) and a C-terminal endonuclease domain (aa 354–662), separated by a flexible linker (Fig. 4a and Extended Data Fig. 5a,c). Using a similar complementation approach, we found

that the serine peptidase activity of LypC was essential for host lysis, as assessed via the catalytic residue S296A mutation (Fig. 4d). By contrast, disrupting LypC putative endonuclease activity via mutating catalytic residues H471 or N505 to alanine (either individually or in

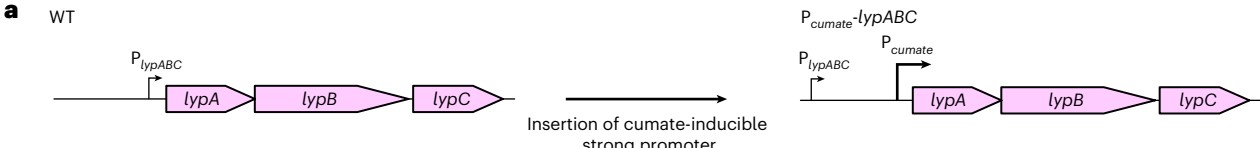

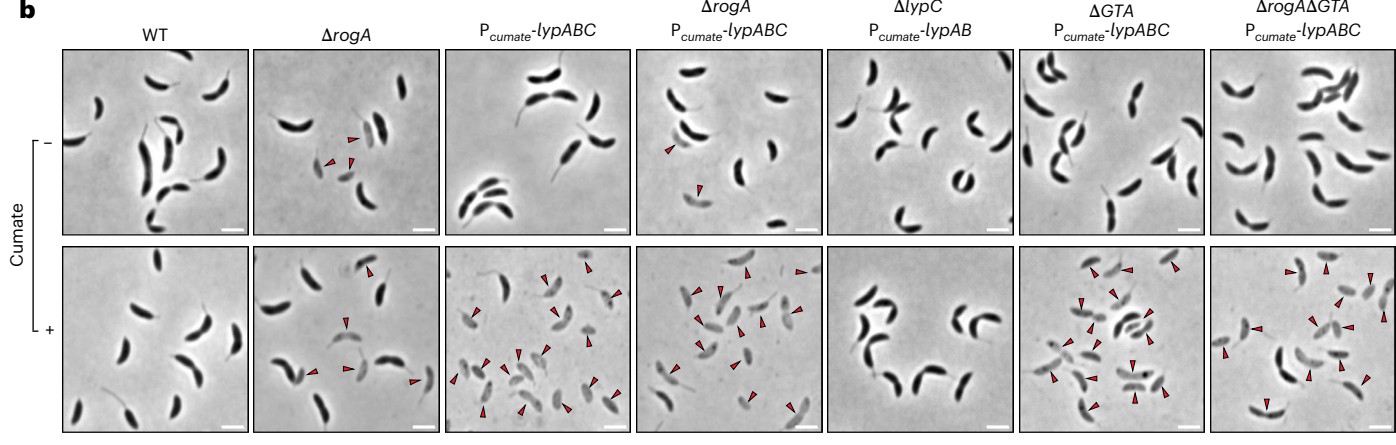

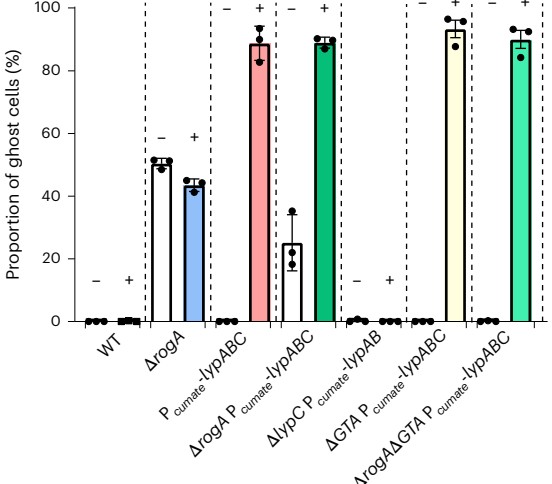

**Fig. 5 | Misregulation of *lypABC* is highly toxic to *C. crescentus*. a**, Schematic of the cumate-inducible promoter (P$_{cumate}$) insertion immediately upstream of the native *lypABC* locus (P$_{lypABC}$). The new strain contains both the original native promoter (P$_{lypABC}$) and the new P$_{cumate}$ promoter. **b**, Top: phase-contrast microscopy of *C. crescentus* strains. Images are representative of three independent repeats. Red arrows indicate ghost cells. Scale bars, 2 µm. Bottom: quantification of ghost cells as a proportion of the total population. White bars with the minus symbol indicate no cumate; the coloured bars with the plus symbol indicate cumate addition. Data represent the mean ± s.d. from three independent experiments (*n* = 400 cells analysed per experiment).

combination) had no effect on LypABC-mediated cell lysis (Fig. 4d), with immunoblotting confirming stable production of all mutant proteins (Extended Data Fig. 4c). We also deleted the endonuclease domain entirely; however, the resulting truncated variant was unstable in vivo, which prevented functional assessment yet suggests a possible contribution towards LypC folding or stability (Extended Data Fig. 4c).

The LypABC system does not encode a gasdermin; however, in other CARD–NLR variants, gasdermins are replaced by alternative death-domain effectors such as phospholipases or endonucleases[27]. While LypC encodes a putative endonuclease domain, our mutagenesis analysis suggests that this domain is not the death-domain effector in *C. crescentus* and it is notable that the *Caulobacter endophyticus* LypC homologue lacks this domain entirely (Extended Data Fig. 8a).

Unlike most anti-phage defence systems whose encoding genes often cluster into genomic 'defence islands', the *C. crescentus lypABC*

operon is not associated with any known or predicted immunity genes (Extended Data Fig. 8b). Furthermore, CARD–NLR systems are rare in bacteria, present in only ~0.35% of sequenced bacterial genomes, as surveyed by DefenseFinder[42]. Indeed, *lypABC* is not conserved even in *Caulobacter* species; we could detect complete *lypABC* operons in only 4 of the 11 *Caulobacter* species predicted to also encode GTAs: *C. crescentus*, *Caulobacter radicis*, *Caulobacter zeae* and *C. endophyticus* (Extended Data Fig. 8c). Together, these results suggest that *C. crescentus* uses immunity-like components to facilitate the release of GTA particles.

## Misregulation of *lypABC* is highly toxic to *C. crescentus*
Classical abortive infection systems must be tightly regulated to avoid self-inflicted cell death through untimely activation—that is, autoimmunity[52]. To investigate whether misregulation of *lypABC* expression

could result in such toxicity, we inserted a strong cumate-inducible promoter[53] immediately upstream of the *lypABC* operon, creating P*cumate*-*lypABC* strains in either the wild-type or the Δ*rogA* background (Fig. 5a). Addition of cumate had no effect on the viability of wild-type cells nor did it cause an elevated proportion of ghost cells in the Δ*rogA* mutant (Fig. 5b). In the Δ*rogA* P*cumate*-*lypABC* strain, there were 25.1 ± 8.9% ghost cells in the absence of cumate induction—probably due to transcriptional bleed-through from the weaker upstream native P*lypABC* promoter. Notably, cumate induction in this same strain led to a dramatic increase in ghost cells (89.0 ± 1.8%). A similarly high level of cell death (88.8 ± 5.5%) occurred following cumate induction of P*cumate*-*lypABC* in the wild-type background (Fig. 5b). We next constructed a Δ*lypC* P*cumate*-*lypAB* strain in which only *lypA* and *lypB* were overexpressed. No ghost cells were detected following cumate induction, indicating that all three proteins are required for toxicity (Fig. 5b). To test whether GTA core cluster components are required for cell lysis, we deleted the core GTA gene cluster in both P*cumate*-*lypABC* and Δ*rogA* P*cumate*-*lypABC* backgrounds. Cumate-induced toxicity was not suppressed in either strain, resulting in high proportions of ghost cells (93.3 ± 4.9% and 90.0 ± 5.0% ghost cells, respectively; Fig. 5b). These findings show that overexpression, and thus misregulation, of *lypABC* is highly toxic to both GTA-producing and non-producing cells, highlighting the need for the system to be tightly regulated.

## The DNA-binding protein CdxB represses *lypABC* expression

Given that *lypABC* overexpression results in severe cell death (Fig. 5b), we hypothesized that *lypABC* expression must normally be tightly repressed to prevent untimely lysis in *C. crescentus*. To find potential repressors, we searched our Tn-seq dataset for genes whose disruption further reduces the fitness of Δ*rogA* cells. We identified one such candidate, *CCNA_02755*, encoding a predicted DNA-binding transcriptional regulator, which had fewer transposon insertions in Δ*rogA* compared with the wild-type strain (Extended Data Fig. 2a). *CCNA_02755* encodes a 132-amino-acid protein named CdxB and contains a predicted N-terminal helix-turn-helix DNA-binding domain (aa 15–72) and C-terminal putative dimerization domain (aa 82–132) (Fig. 6a). CdxB has recently been identified as an XRE-family transcriptional regulator—a protein family with multiple roles including modulation of adhesin development and susceptibility to φCbK phage infection in *C. crescentus*[54].

To identify CdxB-binding sites across the genome, we performed anti-FLAG chromatin immunoprecipitation followed by high-throughput sequencing (ChIP–seq) using a *flag*-tagged allele of *cdxB*, which was expressed ectopically from the *xylX* locus in a Δ*cdxB* background (Fig. 6b). Compared with an untagged CdxB control, CdxB–FLAG had numerous binding sites genome-wide (115 peaks with a fold enrichment ≥2.0), including strong enrichment at the promoter of *lypABC* (Fig. 6b, Extended Data Fig. 9a and Supplementary Table 2).

MEME-ChIP analysis revealed a consensus binding sequence containing two half sites, one of which is poorly conserved (Extended Data Fig. 9d). To validate the ChIP–seq data, we purified CdxB and tested its binding to P*lypABC* DNA using surface plasmon resonance (SPR). By dividing the *lypABC* promoter region into nine overlapping DNA fragments, we confirmed binding of CdxB to DNA fragment 6 that lies directly underneath the CdxB ChIP–seq peak (dissociation constant ($K_d$) = 28.4 nM; Fig. 6c and Extended Data Fig. 9b). Next, to determine whether CdxB activates or represses *lypABC* transcription, we performed RNA sequencing (RNA-seq) comparing Δ*cdxB* with wild-type cells. We observed that, while *lypABC* was transcribed at a low level in wild-type cells, *lypABC* was significantly upregulated in the Δ*cdxB* mutant by 9.8-fold, 6.5-fold and 8.6-fold, respectively, confirming that CdxB acts as a transcriptional repressor of *lypABC* (Fig. 6d and Supplementary Table 3; see also Supplementary Fig. 2 for heat maps and GO analysis).

In the ghost-cell-producing Δ*rogA* strain, CdxB is still bound to P*lypABC*; thus, we hypothesized that deletion of *cdxB* would further activate *lypABC*, exacerbating lysis and ghost cell formation. Indeed, while a *cdxB* deletion alone did not produce ghost cells, the Δ*rogA*Δ*cdxB* double deletion showed a substantially higher proportion of ghost cells (78.2 ± 5.8%) than the single Δ*rogA* mutant (Fig. 6e). Complementation of Δ*rogA*Δ*cdxB* with *cdxB* expressed from its native promoter restored ghost cell levels to that of the Δ*rogA* mutant (41.7 ± 11.5%) (Fig. 6e). Meanwhile, expression of *cdxB* from a stronger P*xyl* promoter completely suppressed ghost cell formation (Fig. 6e), probably owing to further CdxB-mediated repression of *lypABC*. Altogether, these findings show that CdxB is a transcriptional repressor that directly binds the *lypABC* promoter region to regulate GTA-mediated host lysis.

## CdxB represses *gafYZ* expression to regulate GTA activation

Further examination of ChIP–seq data revealed that CdxB also binds the promoter of GTA-activating genes *gafYZ* (P*gafYZ*) (Extended Data Fig. 10a). In the Δ*rogA* background, CdxB binding at P*gafYZ* was further enriched by 2.6-fold, suggesting potential competitive binding between the two repressors at this promoter region (Extended Data Fig. 10a and Supplementary Table 4). To validate the ChIP–seq data, we again performed SPR using purified CdxB protein and confirmed its binding at P*gafYZ* in vitro ($K_d$ = 8.8 nM; Extended Data Fig. 9c and Extended Data Fig. 10b). To determine whether CdxB also represses *gafYZ* transcription, we conducted RNA-seq in a genetic background containing both a Δ*rogA* deletion to de-repress *gafYZ* and a deletion of the entire core GTA gene cluster (Δ*GTA*) to prevent the confounding loss of RNA via cell lysis. Comparing this Δ*rogA*Δ*GTA* strain to the Δ*cdxB*Δ*rogA*Δ*GTA* triple mutant revealed upregulation of *gafYZ* in the absence of *cdxB* by 3.5-fold and 4.0-fold, respectively (Extended Data Fig. 10c, Supplementary Fig. 2b and Supplementary Table 5). As expected, and consistent with previous findings, *lypABC* were also upregulated in the Δ*cdxB*Δ*rogA*Δ*GTA* strain

**Fig. 6 | The DNA-binding protein CdxB represses *lypABC* expression. a**, Left: schematic of predicted CdxB domains including a helix-turn-helix (HTH) motif. Right: AlphaFold3-predicted structure of CdxB. **b**, ChIP–seq profiles showing CdxB binding to the *lypABC* promoter (P*lypABC*). A *cdxB-flag* (blue line) or untagged *cdxB* (black line) allele was expressed from the xylose promoter (P*xyl*) in a Δ*cdxB* mutant. Two independent repeats were performed and representative profiles are shown. **c**, SPR analysis of purified CdxB binding to P*lypABC*, which was divided into nine overlapping fragments. Binding was assessed at 100 nM and 500 nM concentrations. Data show the mean ± s.d. from two independent experiments. **d**, RNA-seq volcano plot showing the log$_2$-transformed fold change for Δ*cdxB* compared with the WT. Grey lines indicate a two-fold change threshold. *lypA*, *lypB* and *lypC*, which are upregulated when *cdxB* is deleted, are annotated. DESeq2 was used for differential expression analysis with default settings (Wald test followed by *P* value adjustment for multiple comparisons using the Benjamini–Hochberg method). Three independent repeats were performed. Top differentially expressed genes are shown in Supplementary Fig. 2a. **e**, Left: phase-contrast microscopy of *C. crescentus* strains. The Δ*rogA*Δ*cdxB* mutant was complemented with *cdxB* expressed from its native promoter (P*cdxB*) or P*xyl*. Images are representative of three independent repeats. Red arrows indicate ghost cells. Scale bars, 2 μm. Right: quantification of ghost cells as a proportion of the total population. Data represent the mean ± s.d. from three independent experiments (*n* = 400 cells analysed per experiment). **f**, Model of GTA activation and LypABC function. Top: GTA-off WT strain. RogA and CdxB repress transcription from P*gafYZ*. IHF binding at P*gafYZ* and P*gtaT* is insufficient for GTA activation. *lypABC* is basally transcribed, producing LypABC in a probably autoinhibited state. GTAs are not produced and cells do not lyse. Bottom: GTA-on Δ*rogA* strain. GafYZ activate GTA transcription (1). *lypABC* remain basally transcribed. GTA particles are produced (2) and may trigger the activation of LypABC (3 and 4), which may then activate downstream lysis factor(s) to mediate particle release (5). Black wavy lines indicate mRNA transcription. Solid arrows indicate defined steps; dashed arrows indicate unknown mechanisms.

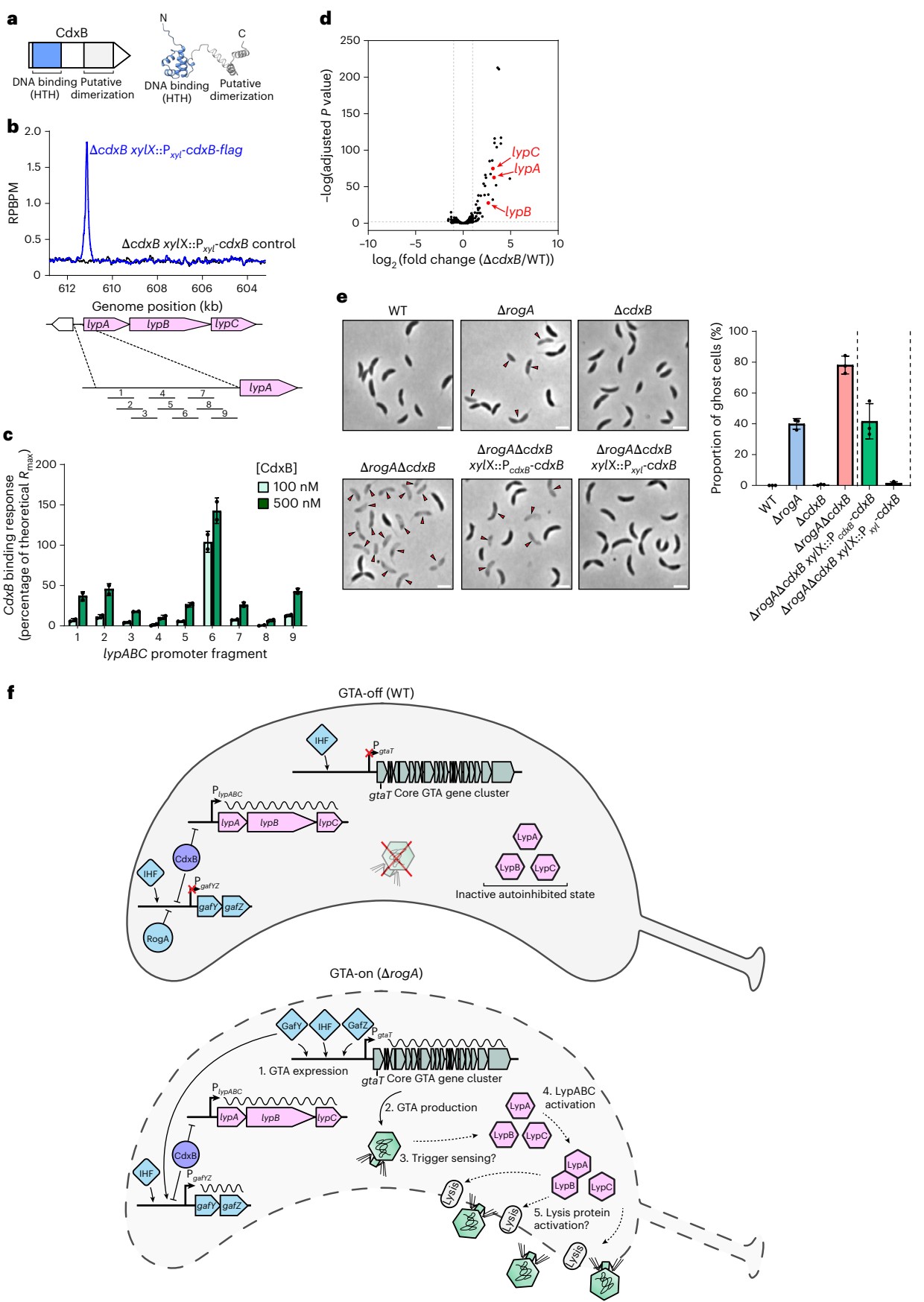

(Extended Data Fig. 10c). Finally, immunoblotting with an anti-GafY polyclonal antibody[35] confirmed that GafY protein was more abundant in the absence of *cdxB* (Extended Data Fig. 10d). Overall, our data show that CdxB directly represses the transcription of GTA-activating genes *gafYZ*. CdxB therefore acts as a dual repressor, coupling two different GTA life stages: transcriptional activation via GafYZ and host cell lysis via LypABC.

## Discussion

Our work uncovers a dedicated lysis control hub, LypABC, which uses domains associated with anti-phage immunity to trigger cell lysis and GTA particle release. *Rhodobacter capsulatus*, by contrast, drives GTA-mediated cell lysis through a classical phage-like holin–endolysin mechanism[22]. *R. capsulatus* and *C. crescentus* GTA systems appear to have evolved from a common α-proteobacterial prophage ancestor[55] and retain several shared features including homology within core cluster proteins and a similar activation factor (GafA in *R. capsulatus* being homologous to GafYZ in *C. crescentus*). Given these similarities, it is intriguing that *C. crescentus* and *R. capsulatus* have evolved diverging solutions for GTA particle release.

The precise mechanism by which LypABC (and CARD–NLR systems generally) are activated and lead to cell death is currently unknown. As LypABC are produced in GTA-off wild-type cells, we reason that the system remains autoinhibited until GTA activation occurs to prevent untimely and unproductive host cell lysis (Fig. 6f). As many anti-phage defence systems are activated by direct interaction with phage structural proteins[47,56–60], it is possible that LypABC may be activated by GTA structural components. Sensing of phage infection can also occur indirectly through defence system guarding of host proteins and processes[61–63]; therefore, it is equally possible that LypABC activation occurs indirectly via sensing of host cell perturbations such as GTA-specific host DNA damage or other stress signals.

Overexpression of *lypABC* caused severe levels of cell lysis including in non-GTA-producing wild-type cells (Fig. 5). Whereas expression of the core GTA cluster is required for cell lysis in the Δ*rogA* strain (Fig. 1b), cumate-induced overexpression of *lypABC* bypasses this requirement, resulting in extensive lysis even in the absence of the core cluster (Fig. 5b). In plants, overexpression of NLR proteins is known to increase the proportion of active-state NLRs, causing a hypersensitive response that leads to cell death even in the absence of pathogens[64–66]. By analogy, elevated levels of LypABC may similarly promote a constitutively active state, removing the requirement for a GTA cluster-associated trigger and initiating cell death autonomously.

In CARD–NLR anti-phage systems, cell death is directly caused by death-domain effectors such as gasdermins, phospholipases or endonucleases[27,28]. As LypABC does not contain a gasdermin and LypC endonuclease activity is dispensable, LypABC do not appear to execute lysis directly. Instead, we propose a model in which LypABC indirectly sense GTA production, transitioning from a resting autoinhibited state to an active state (Fig. 6f). Once activated, LypABC likely trigger the activity of downstream lytic effector(s) (whose identity is currently unknown), resulting in cell lysis.

Host cell lysis is not the only GTA life stage under tight regulatory control; GTA activation in *C. crescentus* is completely inhibited under laboratory conditions via RogA-mediated repression of *gafYZ* expression[25]. Although the natural environmental stimuli that trigger GTA activation remain elusive, our study here identifies another regulatory layer wherein CdxB binding to the *gafYZ* promoter further modulates GTA expression. Deletion of *cdxB* alone is insufficient for GTA activation (Fig. 6e), indicating that RNA polymerase is unable to overcome RogA-mediated repression. However, deletion of both *rogA* and *cdxB* together resulted in many more ghost cells, suggesting that CdxB reinforces *gafYZ* repression. Given that not all cells lyse in the Δ*rogA*Δ*cdxB* double mutant, it is likely that additional repressors

control the key *gafYZ* promoter. Indeed, previous work identified additional XRE-family proteins such as RtrA, RtrB and CdxA, which have multiple binding sites genome-wide and can repress *gafYZ* transcription when overexpressed[54]. Our findings support a model in which CdxB functions as a dual repressor of both GTA activation (via *gafYZ*) and GTA-mediated lysis (via *lypABC*), ensuring that the two critical processes are coupled (Fig. 6f).

LypABC-mediated cell lysis shares some similarity with anti-viral immune systems that confer overall population immunity via death of infected individual cells. While we cannot fully exclude a role for LypABC in anti-phage defence, it is possible that the immunity-like components of LypABC may have been repurposed to facilitate the release of GTA particles. Overall, our work highlights the plasticity of bacterial domains and raises the possibility that immunity-like modules may be adapted to promote horizontal gene transfer.

## Methods

### Strains, media and growth conditions

*Escherichia coli* was routinely grown in LB media at 37 °C, and *C. crescentus* strain NA1000 was grown in PYE media at 30 °C. To culture *C. crescentus* cells to stationary phase for GTA production, 10 ml of PYE liquid was inoculated with a single colony and grown at 30 °C for 20 h with shaking at 250 rpm. A small volume (calculated to set a starting $OD_{600}$ of 0.1) was then transferred from this initial culture into 10 ml of fresh PYE media and grown for a further 20 h. If required, *C. crescentus* growth media were supplemented with antibiotics at the following concentrations: kanamycin, 5 µg ml$^{-1}$ in liquid medium and 25 µg ml$^{-1}$ in plates; oxytetracycline, 1 µg ml$^{-1}$ in liquid medium and 2 µg ml$^{-1}$ in plates; spectinomycin, 25 µg ml$^{-1}$ in liquid medium and 100 µg ml$^{-1}$ in plates; and carbenicillin, not applicable in liquid medium and 50 µg ml$^{-1}$ in plates. Concentrations used for *E. coli* growth media were as follows: kanamycin, 50 µg ml$^{-1}$ in liquid medium and 50 µg ml$^{-1}$ in plates; spectinomycin, 50 µg ml$^{-1}$ in liquid medium and 50 µg ml$^{-1}$ in plates; chloramphenicol, 20 µg ml$^{-1}$ in liquid medium and 30 µg ml$^{-1}$ in plates; and carbenicillin, 50 µg ml$^{-1}$ in liquid medium and 100 µg ml$^{-1}$ in plates. To induce or repress gene expression from the *C. crescentus* P$_{xyl}$ promoter, liquid PYE media were supplemented with either 0.3% xylose (final concentration) or 0.2% glucose (final concentration), respectively. To induce gene expression from the *C. crescentus* P$_{cumate}$ promoter, 100 µM cumate (final concentration) was added to liquid PYE media. Transformations and transductions into *C. crescentus* were performed as previously described[67].

### Plasmid and strain construction

Strains, plasmids and oligonucleotides constructed and used in this study are listed in Supplementary Tables 6, 7 and 8, respectively. The start codon of *C. crescentus* NA1000 *lypC* was corrected to reflect the true coding sequence, which begins at the genomic position 606198, not 606009.

### Construction of pET-21b::*cdxB-6xhis*

The codon-optimized wild-type *cdxB* gene with a *6xhis* tag at the C-terminus was ordered as a double-stranded gBlock gene fragment (IDT) and cloned into NdeI + HindIII-cut pET-21b vector using 2x Gibson assembly master mix. Gibson reactions were introduced into chemically competent *E. coli* DH5α cells. A plasmid containing the correct insert was verified by Sanger sequencing and used to transform chemically competent *E. coli* Rosetta (DE3) cells, generating the strain Rosetta (DE3) pET-21b::*cdxB-6xhis*.

### Construction of pXCHYC-2::*rogA*

The *rogA* gene was amplified from wild-type *C. crescentus* genomic DNA using EJB_005 + EJB_006 primers. PCR products were gel purified and cloned into NdeI + NheI-cut pXCHYC-2 integrative vector using 2x Gibson assembly master mix. Gibson reactions were introduced into

chemically competent *E. coli* DH5α cells. A plasmid containing the correct insert was verified by Sanger sequencing and used to transform electrocompetent *C. crescentus* NA1000 Δ*rogA* cells to generate the strain Δ*rogA xylX*::P*xyl*-*rogA*.

## Construction of pVCHYC-1::P*gtaT*-*mNG*

The promoter region of the core GTA gene cluster (P*gtaT*) gene was amplified from wild-type *C. crescentus* genomic DNA using EJB_001 + EJB_002 primers. The *mNG* gene was amplified from plasmid template using EJB_003 + EJB_004 primers. PCR products were gel purified and cloned into NdeI + NheI-cut pVCHYC-1 integrative vector using 2x Gibson assembly master mix. Gibson reactions were introduced into chemically competent *E. coli* DH5α cells. A plasmid containing the correct insert was verified by Sanger sequencing and used to transform electrocompetent *C. crescentus* NA1000 cells to generate the strain *vanA*::P*gtaT*-*mNG*. Phage ΦCr30 transduction was used to delete *rogA* within this strain, generating the strain Δ*rogA vanA*::P*gtaT*-*mNG*.

## Construction of pNPTS138::Δ*lypABC*

Upstream and downstream regions (each 500 bp) flanking the *lypABC* genomic locus were amplified by PCR using EJB_007 + EJB_008 primers and EJB_009 + EJB_010 primers, respectively. PCR products were gel purified and cloned into BamHI + HindIII-cut pNPTS138 suicide vector using 2x Gibson assembly master mix. Gibson reactions were introduced into chemically competent *E. coli* DH5α cells. A plasmid containing the correct insert was verified by Sanger sequencing and used to transform electrocompetent *C. crescentus* NA1000 wild-type cells and delete *lypABC*. Phage transduction was used to delete *rogA* within this strain, generating the strain Δ*rogA*Δ*lypABC*. To construct the strain Δ*rogA*Δ*lypABC vanA*::P*gtaT*-*mNG*, the plasmid pVCHYC-1::P*gtaT*-*mNG* was first introduced into Δ*lypABC* followed by phage transduction to delete *rogA*.

## Construction of pNPTS138::Δ*lypA*

Upstream and downstream regions (each 500 bp) flanking the *lypA* gene were amplified by PCR using EJB_011 + EJB_012 primers and EJB_013 + EJB_014 primers, respectively. PCR products were gel purified and cloned into BamHI + HindIII-cut pNPTS138 suicide vector using 2x Gibson assembly master mix. Gibson reactions were introduced into chemically competent *E. coli* DH5α cells. A plasmid containing the correct insert was verified by Sanger sequencing and used to transform electrocompetent *C. crescentus* NA1000 wild-type cells and delete *lypA*. Phage transduction was used to delete *rogA* within this strain, generating the strain Δ*rogA*Δ*lypA*.

## Construction of pNPTS138::Δ*lypB*

Upstream and downstream regions (each 500 bp) flanking a region internal to the *lypB* gene (aa 230–335) were amplified by PCR using EJB_015 + EJB_016 primers and EJB_017 + EJB_018 primers, respectively. PCR products were gel purified and cloned into BamHI + HindIII-cut pNPTS138 suicide vector using 2x Gibson assembly master mix. Gibson reactions were introduced into chemically competent *E. coli* DH5α cells. A plasmid containing the correct insert was verified by Sanger sequencing and used to transform electrocompetent *C. crescentus* NA1000 wild-type cells and delete *lypB*. Phage transduction was used to delete *rogA* within this strain, generating the strain Δ*rogA*Δ*lypB*.

## Construction of pNPTS138::Δ*lypC*

Upstream and downstream regions (each 500 bp) flanking the *lypC* gene were amplified by PCR using EJB_019 + EJB_020 primers and EJB_021 + EJB_022 primers, respectively. PCR products were gel purified and cloned into BamHI + HindIII-cut pNPTS138 suicide vector using 2x Gibson assembly master mix. Gibson reactions were introduced into chemically competent *E. coli* DH5α cells. A plasmid containing the

correct insert was verified by Sanger sequencing and used to transform electrocompetent *C. crescentus* NA1000 wild-type cells and delete *lypC*. Phage transduction was used to delete *rogA* within this strain, generating the strain Δ*rogA*Δ*lypC*.

## Construction of pXCHYC-2::*lypA*

The *lypA* gene was amplified from wild-type *C. crescentus* genomic DNA using EJB_023 + EJB_024 primers. PCR products were gel purified and cloned into NdeI + NheI-cut pXCHYC-2 integrative vector using 2x Gibson assembly master mix. Gibson reactions were introduced into chemically competent *E. coli* DH5α cells. A plasmid containing the correct insert was verified by Sanger sequencing and used to transform electrocompetent *C. crescentus* NA1000 Δ*rogA*Δ*lypA* cells to generate the strain Δ*rogA*Δ*lypA xylX*::P*xyl*-*lypA*.

## Construction of pXCHYC-2::*lypB*

The *lypB* gene was amplified from wild-type *C. crescentus* genomic DNA using EJB_025 + EJB_026 primers. PCR products were gel purified and cloned into NdeI + NheI-cut pXCHYC-2 integrative vector using 2x Gibson assembly master mix. Gibson reactions were introduced into chemically competent *E. coli* DH5α cells. A plasmid containing the correct insert was verified by Sanger sequencing and used to transform electrocompetent *C. crescentus* NA1000 Δ*rogA*Δ*lypB* cells to generate the strain Δ*rogA*Δ*lypB xylX*::P*xyl*-*lypB*.

## Construction of pXCHYC-2::*lypC*

The *lypC* gene was amplified from wild-type *C. crescentus* genomic DNA using EJB_027 + EJB_028 primers. PCR products were gel purified and cloned into NdeI + NheI-cut pXCHYC-2 integrative vector using 2x Gibson assembly master mix. Gibson reactions were introduced into chemically competent *E. coli* DH5α cells. A plasmid containing the correct insert was verified by Sanger sequencing and used to transform electrocompetent *C. crescentus* NA1000 Δ*rogA*Δ*lypC* cells to generate the strain Δ*rogA*Δ*lypC xylX*::P*xyl*-*lypC*.

## Construction of pXCHYC-2::*flag*-*lypA* (WT)

The codon-optimized wild-type *lypA* gene with a 1x-*flag* tag at the N-terminus was ordered as a gBlock and cloned into NdeI + NheI-cut pXCHYC-2 integrative vector using 2x Gibson assembly master mix. Gibson reactions were introduced into chemically competent *E. coli* DH5α cells. A plasmid containing the correct insert was verified by Sanger sequencing and used to transform electrocompetent *C. crescentus* NA1000 Δ*rogA*Δ*lypA* cells, generating the strain Δ*rogA*Δ*lypA xylX*::P*xyl*-*flag*-*lypA*.

## Construction of pXCHYC-2::*flag*-*lypA* (ΔCARD)

The codon-optimized *lypA* gene (containing a truncation of the aa 1–80 CARD-like domain) with a 1x-*flag* tag at the N-terminus was ordered as a gBlock and cloned into NdeI + NheI-cut pXCHYC-2 integrative vector using 2x Gibson assembly master mix. Gibson reactions were introduced into chemically competent *E. coli* DH5α cells. A plasmid containing the correct insert was verified by Sanger sequencing and used to transform electrocompetent *C. crescentus* NA1000 Δ*rogA*Δ*lypA* cells, generating the strain Δ*rogA*Δ*lypA xylX*::P*xyl*-*flag*-*lypA* (ΔCARD).

## Construction of pXCHYC-2::*flag*-*lypA* (S262A)

The codon-optimized *lypA* gene (containing a mutation of S262A) with a 1x-*flag* tag at the N-terminus was ordered as a gBlock and cloned into NdeI + NheI-cut pXCHYC-2 integrative vector using 2x Gibson assembly master mix. Gibson reactions were introduced into chemically competent *E. coli* DH5α cells. A plasmid containing the correct insert was verified by Sanger sequencing and used to transform electrocompetent *C. crescentus* NA1000 Δ*rogA*Δ*lypA* cells, generating the strain Δ*rogA*Δ*lypA xylX*::P*xyl*-*flag*-*lypA* (S262A).

## Construction of pXCHYC-2::*flag*-*lypA* (T356A, S360A, F361A)

The codon-optimized *lypA* gene (containing mutations of T356A, S360A and F361A) with a 1x-*flag* tag at the N-terminus was ordered as a gBlock and cloned into NdeI + NheI-cut pXCHYC-2 integrative vector using 2x Gibson assembly master mix. Gibson reactions were introduced into chemically competent *E. coli* DH5α cells. A plasmid containing the correct insert was verified by Sanger sequencing and used to transform electrocompetent *C. crescentus* NA1000 Δ*rogA*Δ*lypA* cells, generating the strain Δ*rogA*Δ*lypA xylX*::P$_{xyl}$-*flag*-*lypA* (T356A, S360A, F361A).

## Construction of pXCHYC-2::*lypB*-*flag* (WT)

The codon-optimized wild-type *lypB* gene with a 1x-*flag* tag at the C-terminus was ordered as a gBlock and cloned into NdeI + NheI-cut pXCHYC-2 integrative vector using 2x Gibson assembly master mix. Gibson reactions were introduced into chemically competent *E. coli* DH5α cells. A plasmid containing the correct insert was verified by Sanger sequencing and used to transform electrocompetent *C. crescentus* NA1000 Δ*rogA*Δ*lypB* cells, generating the strain Δ*rogA*Δ*lypB xylX*::P$_{xyl}$-*lypB*-*flag*.

## Construction of pXCHYC-2::*lypB*-*flag* (ΔNTD)

The codon-optimized *lypB* gene (containing a deletion of the N-terminal domain from aa 1–180) with a 1x-*flag* tag at the C-terminus was ordered as a gBlock and cloned into NdeI + NheI-cut pXCHYC-2 integrative vector using 2x Gibson assembly master mix. Gibson reactions were introduced into chemically competent *E. coli* DH5α cells. A plasmid containing the correct insert was verified by Sanger sequencing and used to transform electrocompetent *C. crescentus* NA1000 Δ*rogA*Δ*lypB* cells, generating the strain Δ*rogA*Δ*lypB xylX*::P$_{xyl}$-*lypB* (ΔNTD)-*flag*.

## Construction of pXCHYC-2::*lypB*-*flag* (K230A)

The codon-optimized *lypB* gene (containing a mutation of K230A) with a 1x-*flag* tag at the C-terminus was ordered as a gBlock and cloned into NdeI + NheI-cut pXCHYC-2 integrative vector using 2x Gibson assembly master mix. Gibson reactions were introduced into chemically competent *E. coli* DH5α cells. A plasmid containing the correct insert was verified by Sanger sequencing and used to transform electrocompetent *C. crescentus* NA1000 Δ*rogA*Δ*lypB* cells, generating the strain Δ*rogA*Δ*lypB xylX*::P$_{xyl}$-*lypB*-*flag* (K230A).

## Construction of pXCHYC-2::*lypB*-*flag* (D335A)

The codon-optimized *lypB* gene (containing a mutation of D335A) with a 1x-*flag* tag at the C-terminus was ordered as a gBlock and cloned into NdeI + NheI-cut pXCHYC-2 integrative vector using 2x Gibson assembly master mix. Gibson reactions were introduced into chemically competent *E. coli* DH5α cells. A plasmid containing the correct insert was verified by Sanger sequencing and used to transform electrocompetent *C. crescentus* NA1000 Δ*rogA*Δ*lypB* cells, generating the strain Δ*rogA*Δ*lypB xylX*::P$_{xyl}$-*lypB*-*flag* (D335A).

## Construction of pXCHYC-2::*lypC*-*flag* (WT)

The codon-optimized wild-type *lypC* gene with a 1x-*flag* tag at the C-terminus was ordered as a gBlock and cloned into NdeI + NheI-cut pXCHYC-2 integrative vector using 2x Gibson assembly master mix. Gibson reactions were introduced into chemically competent *E. coli* DH5α cells. A plasmid containing the correct insert was verified by Sanger sequencing and used to transform electrocompetent *C. crescentus* NA1000 Δ*rogA*Δ*lypC* cells, generating the strain Δ*rogA*Δ*lypC xylX*::P$_{xyl}$-*lypC*-*flag*.

## Construction of pXCHYC-2::*lypC*-*flag* (S296A)

The codon-optimized *lypC* gene (containing a mutation of S296A) with a 1x-*flag* tag at the C-terminus was ordered as a gBlock and cloned into NdeI + NheI-cut pXCHYC-2 integrative vector using 2x Gibson assembly master mix. Gibson reactions were introduced into chemically competent *E. coli* DH5α cells. A plasmid containing the correct insert was verified by Sanger sequencing and used to transform electrocompetent *C. crescentus* NA1000 Δ*rogA*Δ*lypC* cells, generating the strain Δ*rogA*Δ*lypC xylX*::P$_{xyl}$-*lypC*-*flag* (S296A).

## Construction of pXCHYC-2::*lypC*-*flag* (H471A)

The codon-optimized *lypC* gene (containing a mutation of H471A) with a 1x-*flag* tag at the C-terminus was ordered as a gBlock and cloned into NdeI + NheI-cut pXCHYC-2 integrative vector using 2x Gibson assembly master mix. Gibson reactions were introduced into chemically competent *E. coli* DH5α cells. A plasmid containing the correct insert was verified by Sanger sequencing and used to transform electrocompetent *C. crescentus* NA1000 Δ*rogA*Δ*lypC* cells, generating the strain Δ*rogA*Δ*lypC xylX*::P$_{xyl}$-*lypC*-*flag* (H471A).

## Construction of pXCHYC-2::*lypC*-*flag* (N505A)

The codon-optimized *lypC* gene (containing a mutation of N505A) with a 1x-*flag* tag at the C-terminus was ordered as a gBlock and cloned into NdeI + NheI-cut pXCHYC-2 integrative vector using 2x Gibson assembly master mix. Gibson reactions were introduced into chemically competent *E. coli* DH5α cells. A plasmid containing the correct insert was verified by Sanger sequencing and used to transform electrocompetent *C. crescentus* NA1000 Δ*rogA*Δ*lypC* cells, generating the strain Δ*rogA*Δ*lypC xylX*::P$_{xyl}$-*lypC*-*flag* (N505A).

## Construction of pXCHYC-2::*lypC*-*flag* (H471A, N505A)

The codon-optimized *lypC* gene (containing mutations of H471A and N505A) with a 1x-*flag* tag at the C-terminus was ordered as a gBlock and cloned into NdeI + NheI-cut pXCHYC-2 integrative vector using 2x Gibson assembly master mix. Gibson reactions were introduced into chemically competent *E. coli* DH5α cells. A plasmid containing the correct insert was verified by Sanger sequencing and used to transform electrocompetent *C. crescentus* NA1000 Δ*rogA*Δ*lypC* cells, generating the strain Δ*rogA*Δ*lypC xylX*::P$_{xyl}$-*lypC*-*flag* (H471A, N505A).

## Construction of pNPTS138::P$_{cumate}$-*lypABC*

Upstream and downstream regions (each 500 bp) flanking the start codon of the *lypA* gene were amplified by PCR using EJB_029 + EJB_030 primers and EJB_033 + EJB_034 primers, respectively. The cumate promoter sequence was amplified from a plasmid template using primers EJB_031 and EJB_032. PCR products were gel purified and cloned into BamHI + HindIII-cut pNPTS138 suicide vector using 2x Gibson assembly master mix. Gibson reactions were introduced into chemically competent *E. coli* DH5α cells. A plasmid containing the correct insert was verified by Sanger sequencing and used to transform electrocompetent *C. crescentus* NA1000 wild-type cells, generating the strain P$_{cumate}$-*lypABC*. To generate the strain Δ*rogA* P$_{cumate}$-*lypABC*, phage transduction was used to generate a marked deletion of *rogA*. To generate Δ*lypC* P$_{cumate}$-*lypAB*, *lypC* was deleted using the suicide plasmid pNPTS::Δ*lypC*. To construct the strain Δ*GTA* P$_{cumate}$-*lypABC*, the core GTA cluster was deleted using the suicide plasmid pNPTS::ΔGTA. This was followed by phage transduction to delete *rogA* and thus generate the additional strain Δ*rogA*Δ*GTA* P$_{cumate}$-*lypABC*.

## Construction of pNPTS138::Δ*cdxB*

Upstream and downstream regions (each 500 bp) flanking the *cdxB* gene were amplified by PCR using EJB_035 + EJB_036 primers and EJB_037 + EJB_038 primers, respectively. PCR products were gel purified and cloned into BamHI + HindIII-cut pNPTS138 suicide vector using 2x Gibson assembly master mix. Gibson reactions were introduced into chemically competent *E. coli* DH5α cells. A plasmid containing the correct insert was verified by Sanger sequencing and used to transform electrocompetent *C. crescentus* NA1000 wild-type cells and delete

*cdxB*, generating the strain Δ*cdxB*. Phage transduction was used to delete *rogA* within this strain, generating the strain Δ*rogA*Δ*cdxB*.

### Construction of pXCHYC-2::*cdxB-flag*

The *cdxB* gene was amplified from wild-type *C. crescentus* genomic DNA using EJB_039 + EJB_040 primers, which contained a 1x-*flag* tag. PCR products were gel purified and cloned into NdeI + NheI-cut pXCHYC-2 integrative vector using 2x Gibson assembly master mix. Gibson reactions were introduced into chemically competent *E. coli* DH5α cells. A plasmid containing the correct insert was verified by Sanger sequencing and used to transform electrocompetent *C. crescentus* NA1000 Δ*cdxB* cells to generate the strain Δ*cdxB xylX*::P$_{xyl}$-*cdxB-flag*. Phage transduction was used to delete *rogA* within this strain, generating the strain Δ*rogA*Δ*cdxB xylX*::P$_{xyl}$-*cdxB-flag*.

### Construction of pXCHYC-2::*cdxB*

The *cdxB* gene was amplified from wild-type *C. crescentus* genomic DNA using EJB_041 + EJB_042 primers. PCR products were gel purified and cloned into NdeI + NheI-cut pXCHYC-2 integrative vector using 2x Gibson assembly master mix. Gibson reactions were introduced into chemically competent *E. coli* DH5α cells. A plasmid containing the correct insert was verified by Sanger sequencing and used to transform electrocompetent *C. crescentus* NA1000 Δ*cdxB* cells to generate the strain Δ*cdxB xylX*::P$_{xyl}$-*cdxB*. Phage transduction was used to delete *rogA* within this strain, generating the strain Δ*rogA*Δ*cdxB xylX*::P$_{xyl}$-*cdxB*.

### Construction of pXCHYC-2::P$_{cdxB}$-*cdxB*

The *cdxB* gene including its native promoter (P$_{cdxB}$) was amplified from wild-type *C. crescentus* genomic DNA using EJB_043 + EJB_044 primers. PCR products were gel purified and cloned into NdeI + NheI-cut pXCHYC-2 integrative vector using 2x Gibson assembly master mix. Gibson reactions were introduced into chemically competent *E. coli* DH5α cells. A plasmid containing the correct insert was verified by Sanger sequencing and used to transform electrocompetent *C. crescentus* NA1000 Δ*cdxB* cells to generate the strain Δ*cdxB xylX*::P$_{cdxB}$-*cdxB*. Phage transduction was used to delete *rogA* within this strain, generating the strain Δ*rogA*Δ*cdxB xylX*::P$_{cdxB}$-*cdxB*.

### Construction of pNPTS138::Δ*GTA*

Upstream and downstream regions (each 500 bp) flanking the core GTA cluster (CCNA_02861-CCNA_02880) were amplified by PCR using EJB_045 + EJB_046 primers and EJB_047 + EJB_048 primers, respectively. PCR products were gel purified and cloned into BamHI + HindIII-cut pNPTS138 suicide vector using 2x Gibson assembly master mix. Gibson reactions were introduced into chemically competent *E. coli* DH5a cells. A plasmid containing the correct insert was verified by Sanger sequencing and used to transform electrocompetent *C. crescentus* NA1000 wild-type cells and delete the gene cluster, generating the strain Δ*GTA*. Phage transduction was used to delete *rogA* within this strain, generating the strain Δ*rogA*Δ*GTA*. To construct the strain Δ*cdxB*Δ*rogA*Δ*GTA*, *cdxB* was first deleted within the Δ*GTA* strain to make Δ*cdxB*Δ*GTA* and then phage transduction was performed to delete *rogA*.

### Construction of pKT25::*lypA/lypB/lypC* and pUT18C::*lypA/lypB/lypC* (individual domains)

DNA encoding individual domains of LypA, LypB and LypC were chemically synthesized as double-stranded gBlock DNA fragments (IDT). These DNA fragments were assembled into a BamHI–EcoRI-cut pKT25 vector or a BamHI–EcoRI-cut pUT18C vector using 2x Gibson assembly master mix. The resulting plasmids were verified by whole-plasmid sequencing (Plasmidsaurus). To construct bacterial two-hybrid strains, chemically competent BTH101 *E. coli cya⁻* cells were co-transformed with combinations of pKT25 and pUT18C plasmids above.

### Construction of pKT25::*lypA/lypB/lypC* and pUT18C::*lypA/lypB/lypC* (full-length proteins)

DNA encoding LypA (two overlapping segments), LypB (four overlapping segments) and LypC (two overlapping segments) were chemically synthesized as double-stranded gBlock DNA fragments (IDT). These DNA fragments were assembled into a BamHI–EcoRI-cut pKT25 vector or a BamHI–EcoRI-cut pUT18C vector using 2x Gibson assembly master mix. The resulting plasmids were verified by whole-plasmid sequencing (Plasmidsaurus). To construct bacterial two-hybrid strains, chemically competent BTH101 *E. coli cya⁻* cells were co-transformed with combinations of pKT25 and pUT18C plasmids above.

### Construction of pNPTS138::*CCNA_02899-tet$^R$*

A 1-kb region upstream and a 1-kb region downstream of the intergenic space between CCNA_02899 and CCNA_02900 were amplified with the primer pairs oKRG646 + oKRG647 and oKRG680 + oKRG681, respectively. The tetracycline resistance cassette (*tet$^R$*) was amplified with the primer pair oKRG648 and oKRG649. All three PCR products were fused into one product via SOE-PCR using the primer pair oKRG646 and oKRG681. The resulting PCR product was gel purified, digested with AflII and NheI, and ligated into AflII–NheI-cut pNPTS138. To construct the strain CB15 *tet$^R$*, two-step homologous recombination was performed by introduction of pNPTS::CCNA_02899-*tetR* into electrocompetent *C. crescentus* CB15 cells followed by sucrose counter-selection.

### Construction of pBXMCS-6::P$_{xyl}$-*gafYZ*

The *gafYZ* genes were amplified from wild-type *C. crescentus* genomic DNA using oKRG92 + oKRG401 primers. PCR products were digested with NdeI and SacI and ligated into NdeI + SacI-cut pBXMCS-6 high-copy vector. Ligation reactions were introduced into chemically competent *E. coli* DH5α cells. A plasmid containing the correct insert was verified by Plasmidsaurus sequencing and used to transform electrocompetent *C. crescentus* CB15 *tet$^R$* cells to generate the donor strain CB15 *tet$^R$* pBXMCS-6::P$_{xyl}$-*gafYZ*. An empty pBXMCS-6 vector was separately transformed into *C. crescentus* CB15 *tet$^R$* cells to generate the control donor strain CB15 *tet$^R$* pBXMCS-6::P$_{xyl}$-*empty*.

### Genomic DNA extraction

To extract total genomic DNA from *C. crescentus* strains, 1 ml of stationary-phase cells was centrifuged at 17,000*g* for 1 min and the pellet was resuspended in 300 µl of Cell Lysis Solution (PureGene, Qiagen). Resuspended cells were incubated at 50 °C for 15 min, mixed with 1 µl of RNase A (NEB 20 mg ml⁻¹ stock) and incubated at 37 °C for 1 h. Samples were cooled to room temperature, mixed with 200 µl of Protein Precipitation Solution (PureGene, Qiagen) and then centrifuged at 17,000*g* for 10 min. The supernatant was combined with 600 µl of isopropanol and mixed well by inversion. Samples were centrifuged again at 17,000*g* for 1 min, and the supernatant was discarded. Pellets were resuspended in 600 µl of 70% ethanol and centrifuged at 17,000*g* for 1 min. The supernatant was discarded, and a final 1-min centrifugation was performed to remove any remaining liquid. The DNA pellet was resuspended in 100 µl of sterile nuclease-free water and further incubated at 37 °C for 15–30 min. To assay for encapsulation of host DNA into GTA particles, 50 µl of genomic DNA was run on a 1% agarose gel at 150 V for 45–60 min and visualized to determine the presence or absence of an 8.3-kb band.

### Tn-seq

To prepare strains for Tn-seq, 10 ml of *C. crescentus* NA1000 wild-type and Δ*rogA* strains were cultured to stationary phase; simultaneously, the Tn5 transposon delivery plasmid pMCS-6-Tn5-R6kg-kan2 was introduced into an *E. coli* S17-1 conjugative donor strain by heat-shock transformation. The next day, *E. coli* transformants were resuspended in LB liquid and washed three times by centrifugation to remove kanamycin. *C. crescentus* and *E. coli* cultures were adjusted to an OD$_{600}$ of 0.5, and

then 10 ml of each *C. crescentus* strain was mixed with 1 ml of *E. coli* S17-1 harbouring pMCS-6-Tn5-R6kg-kan2. The mixtures were centrifuged at 4,650*g* for 5 min, and the pellet was resuspended in 500 µl of PYE. Ten sterile 0.45 µm nitrocellulose membranes were placed onto PYE agar plates, and then 50 µl of the *C. crescentus*–*E. coli* resuspension was pipetted onto each membrane and incubated at 30 °C for 4 h. Cells were removed from each filter membrane by the addition of 500 µl of PYE followed by vigorous vortexing and then combined. Then, 500 µl of either the wild-type or the Δ*rogA* conjugation mix was spread onto large PYE agar plates supplemented with kanamycin (to select for *C. crescentus* cells with transposon insertions) and carbenicillin (to kill the *E. coli* S17-1 donor) and incubated at 30 °C for 3 days. Resulting colonies were washed from large Petri discs and resuspended in PYE liquid, and the suspension was mixed thoroughly; then, 1 ml was removed for total genomic DNA extraction. Purified genomic DNA was sheared into 200–500-bp fragments, and libraries were constructed for Illumina sequencing as previously described[68] followed by a PCR amplification using a universal P5-ME primer annealing to the transposon and a specific index primer annealing to adaptor-ligated sheared DNA fragments. DNA was sequenced on an Illumina HiSeq2500 platform at the Tufts University Genomics facility. To analyse Tn-seq data, short single-end Illumina reads were mapped back to the *C. crescentus* NA1000 reference genome using HISAT2 (ref. 69) and then associated with genes using HTSEQ-count. The log$_2$-transformed fold change and adjusted *P* values were calculated using DESeq2 (ref. 70).

## RNA-seq
*C. crescentus* strains were cultured to stationary phase, and 5-ml cell cultures were pelleted for RNA extraction. Total RNA was purified using the Direct-zol RNA miniprep kit (Zymo Research), and 10 µg was incubated with 20 units of Turbo DNaseI (Invitrogen) at 37 °C for 1 h to remove any contaminated genomic DNA. DNaseI was subsequently removed using the RNA Clean and Concentrator-25 kit (Zymo Research). Purified RNA samples were shipped to Genewiz where bacterial rRNA depletion was performed with a NEBNext rRNA Depletion Kit (catalogue number E7850X). DNA libraries were prepared by Genewiz and sequenced on an Illumina NovaSeq platform. RNA-seq data, consisting of short, paired-end Illumina reads, were analysed as described previously for Tn-seq data.

## ChIP–seq
*C. crescentus* strains were cultured to stationary phase and fixed with a final concentration of 1% formaldehyde. Cells were incubated on a wheel rotator at room temperature for 30 min; then, the reaction was quenched by the addition of 250 mM glycine for 10 min. Cells were washed three times in 1× PBS and then resuspended in 1 ml of cell lysis buffer 1 (20 mM K-HEPES, pH 7.9; 50 mM KCl; 10% glycerol; and Roche EDTA-free protease inhibitors). Samples were sonicated on ice (11 cycles of 15 s 'on' and 15 s 'off' at an amplitude setting of 8) using a Soniprep 150 probe-type sonicator; then, debris was pelleted by centrifugation at 17,000*g* at 4 °C for 20 min. Supernatant was transferred to a new microcentrifuge tube, and the buffer concentration was adjusted via the addition of 10 µl of 1 M Tris, pH 8 (10 mM final concentration); 20 µl of 5 M NaCl (150 mM final concentration); and 10 µl of 10% NP40 (0.1% final concentration). Then, 50 µl was transferred to a new tube ('input' control) and stored at −20 °C. Storage buffer was removed from anti-Flag antibody M2 agarose beads (Merk) by repeated cycles of centrifugation and resuspension of anti-Flag beads (100 µl) in 1 ml of IPP150 buffer (10 mM Tris-HCl, pH 8; 150 mM NaCl; and 0.1% NP40). ChIP sample supernatant was transferred to the prepared anti-Flag beads and incubated on a wheel rotator at 4 °C overnight. Samples were then washed five times with 1 ml of IPP150 buffer, followed by two washes with 1 ml of 1× TE buffer (10 mM Tris-HCl, pH 8; 1 mM EDTA). A two-step elution of protein–DNA complexes was performed. Briefly, beads were first incubated with 150 µl of elution buffer (50 mM Tris-HCl,

pH 8; 10 mM EDTA; and 1% SDS) at 65 °C for 15 min, then centrifuged at 17,000*g* for 5 min, and the supernatant was transferred to a new tube. Beads were then incubated with 100 µl of 1× TE + 1% SDS at 65 °C for an additional 15 min, then centrifuged at 17,000*g* for 5 min. Supernatant eluates (ChIP samples) were combined and incubated at 65 °C overnight to reverse cross-links. The 'input' control fraction was also incubated at 65 °C overnight with the addition of 200 µl of 1× TE + 1% SDS. DNA from the ChIP and input fractions was purified with a QIAquick PCR purification kit (Qiagen) and eluted in 40 µl of water. Libraries were then prepared for Illumina sequencing using the NEXT Ultra II library preparation kit (NEB). DNA was sequenced on an Illumina Nextseq 550 platform at the Tufts University Genomics facility. To analyse ChIP–seq data, short paired-end Illumina reads were mapped back to the *C. crescentus* NA1000 reference genome using HISAT2. The sequencing coverage for each nucleotide position was determined using bedtools genome-cov[71]. Peaks were identified using MACS2 callpeak[72]. ChIP–seq profiles were plotted using GraphPad Prism, with the *x*-axis representing the genomic nucleotide position and the *y*-axis representing the RPBPM.

## Protein purification
Plasmid pET21b::*cdxB-his6* was introduced into *E. coli* Rosetta BL21 DE3 (pLys) cells by heat-shock transformation. A 20-ml overnight culture was used to inoculate 1 l of LB media supplemented with chloramphenicol and carbenicillin. Cells were cultured at 37 °C with shaking to early exponential phase (OD$_{600}$ = ~0.4), cooled to 18 °C and then supplemented with 1 mM isopropyl-β-ᴅ-thiogalactopyranoside (IPTG). The culture was then incubated at 18 °C with shaking for 20 h to induce protein production. Cells were collected by centrifugation at 4,248*g* for 10 min and resuspended in 25 ml of buffer A (5% glycerol; 100 mM Tris-HCl, pH 8.0; 300 mM NaCl; 10 mM imidazole) supplemented with 1 EDTA-free protease inhibitor tablet and lysozyme. Cells were then lysed by sonication (10 cycles of 15 s, resting on ice for 15 s between cycles). Cell debris was pelleted by centrifugation at 32,000*g* for 35 min, and the supernatant was filtered through a 0.22-µm membrane. The lysate was incubated with 2 ml of HIS-Select Cobalt Affinity Gel Resin (Merck) (pre-washed with 50 ml of buffer A) on a wheel rotator at 4 °C for 1 h. The lysate was then drained and the resin washed with 75 ml of buffer A. Protein was eluted in 2.7 ml of buffer B (5% glycerol; 100 mM Tris-HCl, pH 8.0; 300 mM NaCl; 500 mM imidazole). Finally, protein was desalted using a PD-10 column, checked for purity by SDS–PAGE and stored at −80 °C in storage buffer (5% glycerol; 100 mM Tris-HCl, pH 8.0; 300 mM NaCl).

## SPR
SPR experiments were performed using a Biacore 8K (Cytiva) and the Reusable DNA Capture Technique (ReDCaT)[73] to quantify protein–DNA interactions. Promoter regions of *gafYZ* and *lypABC* were divided into 50-bp single-stranded oligonucleotides overlapping by 20 bp and annealed to a complementary ssDNA fragment containing a ReDCaT-specific adaptor. The adaptor binds a biotinylated linker permanently attached to a Series S Sensor Chip Streptavidin (Cytiva), allowing DNA fragments to be bound and stripped after each experiment. Double-stranded DNA duplexes (1 µM in 1× HBS-EP+ buffer: 150 mM NaCl, 3 mM EDTA, 0.05% Tween-20, 10 mM HEPES, pH 7.4) were loaded onto the chip. Flow cells 1, 3, 5, 7, 9, 11, 13 and 15 served as references (FC_ref), and 2, 4, 6, 8, 10, 12, 14 and 16 as test cells (FC_test). Multi-cycle kinetics were used to determine the binding affinity of CdxB to the promoters of *lypABC* (oligo 6) and *gafYZ* (oligo 10). DNA fragments were injected over FC_test at 10 µl min⁻¹ for 60 s to achieve 85–95 response units, followed by CdxB injection over both FC_ref and FC_test for 60 s and a 420-s dissociation phase. Protein concentrations ranged from 0.98 nM to 500 nM, with buffer-only controls. Chips were regenerated between cycles with 1 M NaCl and 50 mM NaOH for 60 s. FC_ref responses were subtracted from FC_test to obtain the specific

protein–DNA interaction signal, and double referencing was applied using buffer-only controls to correct for bulk refractive index and instrument artefacts. Binding was recorded in response units (RU) and compared to the theoretical maximal response at saturation ($R_{max}$). The $R_{max}$ value was calculated using the equation: $R_{max}$ (RU) = (molecular mass of protein/molecular mass of DNA fragment) × stoichiometry × DNA capture (RU). The strength of binding was expressed as a percentage of $R_{max}$ which was calculated using the equation: $\%R_{max}$ = RU/$R_{max}$ × 100 (ref. [73]). Binding affinities ($K_d$) were calculated using Biacore Insight Evaluation Software (GE Healthcare).

### Immunoblots

Bacterial cells were collected by centrifugation at 17,000g for 10 min, and pellets were resuspended in 300 µl of lysis buffer (20 mM K-HEPES, pH 7.9; 50 mM KCl; 10% glycerol; 1 EDTA-free protease inhibitor tablet). Cells were lysed by sonication (3 cycles of 10 s, resting on ice for 10 s between cycles), and the debris was pelleted by centrifugation at 17,000g for 15 min at 4 °C. If required, extracellular fractions were prepared by filtering supernatant from the collected cultures through a 0.22-µm membrane, followed by spin-concentration in Amicon Ultra-15 centrifugal filter units with a 100-kDa molecular weight cut-off to collect GTA particles. Total protein concentrations were determined using Bradford reagent and used to match the amount of total protein loaded across different samples. Samples were denatured by boiling in SDS–PAGE loading dye containing β-mercaptoethanol and then loaded onto 12% Novex Tris-Glycine WedgeWell gels (Thermo Fisher) alongside a Color Pre-stained Broad Range protein ladder (NEB). Gels were run at 200 V for 30 min, then proteins were transferred onto a PVDF membrane using the Trans-Blot Transfer System (BioRad). The membrane was blocked in 1× TBS buffer (100 mM Tris-HCl, 140 mM NaCl, pH 7.4) + 0.1% Tween-20 + 5% milk powder) for 1 h. For anti-Flag immunoblots, the membrane was then incubated with a 1:5,000 dilution of a monoclonal anti-Flag M2-Peroxidase HRP-conjugated antibody (Merck) for 1 h. The membrane was then washed five times in TBS buffer + 0.1% Tween-20 for 1 min each time. Finally, the membrane was incubated with SuperSignal West Femto Maximum Sensitivity Substrate (Thermo Scientific) and visualized in an Amersham Imager 600 (GE Healthcare). For immunoblots to detect either GafY or GtaL, the membrane was first incubated with either a custom-made polyclonal anti-GafY antiserum (3:1,000 dilution) or polyclonal anti-GtaL antiserum (1:1,000 dilution) for 1 h. Following three 5-min washes in TBS buffer + 0.1% Tween-20, membranes were incubated with a 1:10,000 dilution of HRP-conjugated goat anti-rabbit antibody (catalogue number ab6721, Abcam) for 1 h. Membranes were washed three times for 5 min and imaged as described above. Protein loading controls were run on 12% SDS–PAGE gels and stained with an InstantBlue Coomassie protein stain (Abcam).

### Light microscopy and image analysis

Bacteria were immobilized on 1% agarose pads and visualized under a Zeiss Axio Observer Z.1 inverted epifluorescence microscope equipped with a Zeiss Plan Apochromat 100x/NA 1.4 Ph3 objective lens, an sCMOS camera (Hamamatsu Orca FLASH 4) and a Zeiss Colibri 7 LED light source. For propidium iodide staining, cells were incubated with a final concentration of 15 µM propidium iodide for 15 min in the dark and then immediately imaged on agarose pads. The following filter sets were used when required: GFP (excitation, 450–488 nm; emission, 499–549 nm) and propidium iodide (excitation, 450–488 nm; emission, 599–659 nm). Images were acquired in Zeiss Zen Blue software then processed and analysed in Fiji[74]. The MicrobeJ plug-in[75] for Fiji was used for quantitative analysis. All bacteria detected in MicrobeJ (n = 400 per biological repeat) were inspected to ensure that they were detected correctly. For ghost cell quantification, the mean intensity value of the phase channel was automatically measured in each individual cell and a threshold was set to distinguish between phase-light ghost cells

(lower intensity value) and phase-dark cells (higher intensity value). To determine the proportion of ghost cells, the number of cells with a mean phase intensity value beneath the threshold was calculated and expressed as a percentage of the total cell population. To quantify the proportion of mNG-fluorescent or propidium iodide-fluorescent cells, the mean intensity value of each fluorescence channel was automatically measured in each individual cell and a threshold was set to distinguish between fluorescent and non-fluorescent cells. To determine the proportion of fluorescent cells, the number of cells with a mean fluorescence intensity value above the threshold was calculated and expressed as a percentage of the total cell population.

### Time-lapse microscopy

To prepare strains for time-lapse microscopy, a 10-ml culture of *C. crescentus* was grown for 16 h at 30 °C with shaking. A 5-ml aliquot from this culture was then centrifuged at 4,650g for 10 min, and the supernatant was filtered through a 0.22-µm membrane to collect early stationary-phase spent PYE culture media. Spent PYE media was then mixed with melted agarose and pipetted into a GeneFrame (Thermo Scientific) attached to a glass slide to generate a 1.2% PYE agarose pad. Where required, the agarose pad was supplemented with propidium iodide at a final concentration of 0.1 µM. Once set, 1 µl of *C. crescentus* culture was pipetted onto the agarose pad, which was then covered with a cover slip and firmly sealed. Time-lapse experiments were performed at 30 °C inside a temperature-controlled incubation chamber fitted to the Zeiss Axio Observer Z.1 microscope. Time-lapse images were acquired every 10 min across 10 different positions. Data were acquired in Zeiss Zen Blue software then processed and analysed in Fiji. Where required, channels were registered to correct for drift using the HyperStackReg plug-in for Fiji.

### Cryo-electron tomography and processing

*C. crescentus* strains were grown for 20 h, diluted in fresh media to an $OD_{600}$ of 0.1 and then incubated for a further 20 h. Cultures were mixed in a ratio of 1:2 with 6 nm BSA-conjugated gold fiducials (Aurion), which were buffer transferred to PYE. The mixture (3.8 µl) was applied to 200-mesh R2/1 copper grids (Quantifoil) and vitrified in liquid ethane (blot time of 7 s, force 0, wait time of 2 s) using a Vitrobot Mark IV (Thermo Fisher Scientific). The sample was blotted using only the back vitrobot pad, with the front pad covered in parafilm as reported previously[76]. Grids were either imaged at 200 kV using a Glacios TEM equipped with a Falcon4 detector (University of York) or imaged at 300 kV using a Krios G3i TEM equipped with a K3 camera (SLAC-Stanford). Tilt series were acquired using a dose symmetric scheme (SLAC-Stanford data) or bidirectional scheme starting from −21° (University of York data), with a 3° increment and 54°/−54° span. The dose per tilt image was 2.9 (SLAC-Stanford data) and 3.2 e⁻ Å⁻² (University of York data). Tilt images were gain corrected, aligned to stacks and dose weighted using the IMOD alignframes command[77]. Stacks were imported to EMAN2 (ref. [78]) in which tomograms were reconstructed. Tomograms were visualized using IMOD. Tomogram segmentation was performed in EMAN2 using the tomoseg convolutional neural network-based semi-automated cellular tomogram annotation protocol[79], and segmented volumes were visualized using ChimeraX[80].

### Bacterial two-hybrid assays

Bacterial two-hybrid assays were performed exactly as described in the Euromedex Bacterial Adenylate Cyclase Two-Hybrid System Kit manual (catalogue number EUK001). Briefly, *E. coli* BTH101 cells were co-transformed with a pair of plasmids by electroporation, then spread onto LB plates containing carbenicillin and kanamycin and incubated at 30 °C for 20 h. Three colonies from each plasmid combination were cultured in LB with carbenicillin and kanamycin for 16 h, then 5 µl was spotted on McConkey agar plates and incubated at 30 °C for 24 h.

## Gene transfer assay

Donor strains (containing both the replicative plasmid pBXMCS-6::$P_{xyl}$-gafYZ and a tetracycline resistance marker integrated at the 3.0-Mbp locus) and a recipient strain (containing both a kanamycin resistance marker integrated at the *hfaB* locus and a chloramphenicol resistance marker integrated at the *xylX* locus) were grown to an $OD_{600}$ = ~1.3 in 25 ml PYE containing 0.2% glucose and chloramphenicol in 250-ml flasks. Then, 6 ml of each donor and recipient was mixed in a 1:1 ratio in 125-ml flasks and induced with 0.3% xylose for 16 h. Subsequently, 1 ml of each culture was pelleted at 5,000*g* for 2 min and directly plated onto PYE plates containing 0.3% xylose and both kanamycin and tetracycline (at half-strength concentration) to select for doubly resistant transductants. To determine the total number of recipient cells, serial dilutions from the mixed cultures were spotted onto PYE plates containing kanamycin only. Colonies were counted and imaged after 3 days of growth at 30 °C. Gene transfer rates were calculated by dividing transductant CFU ml$^{-1}$ by the total recipient CFU ml$^{-1}$.

## GO enrichment analysis

GO enrichment was performed using the GOseq package[81] for R (R version 4.5.0), and the results were visualized with ggplot2 (ref. [82]) as bubble plots showing the top 10 enriched GO terms. Genes without GO annotations were excluded.

## Phylogenetic analysis

CARD–NLR protein sequence accessions were obtained from the DefenseFinder server[83], bNACHT sequence accessions were sourced from ref. [45], and MalT and SWACOS STAND ATPase sequence accessions were retrieved from ref. [84]. A total of 83 sequences were aligned using MAFFT[85], and the ATPase domain alignment was trimmed with ClipKit[86]. The phylogenetic tree was constructed with IQ-TREE 3 (ref. [87]) using the maximum likelihood method and the best-fit substitution model (Q.PFAM+F+R4) with 1,000 bootstrap replicates to assess node support. The final tree was visualized and annotated using iTOL[88].

## Bioinformatics analysis and structural predictions

Sequence similarity searches for LypABC protein homologues were carried out in BLASTP[89] with default parameters (nr database, expect threshold 0.05, BLOSUM62 matrix, most recent searches: May 2025). Structural homology searches were performed with Foldseek against the databases PDB100 20240101 and AlphaFold/UniProt50 v6 using the 3Di/AA mode[43]. InterProScan[90] and HHPred[91] (using the HHPred PDB_mmCIF70_25_May database) were used to annotate protein domains (most recent searches: May 2025). AlphaFold3 (ref. [92]) was used to generate predicted protein structures. Models were visualized in PyMOL v.2.5.3 (Schrödinger) and prepared for presentation using UCSF ChimeraX v.1.9 (ref. [80]).

## Reporting summary

Further information on research design is available in the Nature Portfolio Reporting Summary linked to this article.

## Data availability

The ChIP–seq, RNA-seq and Tn-seq raw data generated in this work have been deposited in the Gene Expression Omnibus (GEO) database under accession codes GSE295577, GSE295580 and GSE295581, respectively. All plasmids and strains constructed in this study are available upon request. Source data are provided with this paper.

## Code availability

For Tn-seq and RNA-seq analyses, DESeq2 output files were annotated using a custom Python pipeline available via Zenodo at https://doi.org/10.5281/zenodo.18781500 (ref. [93]). GO analyses, including plotting, were performed using custom R scripts available via Zenodo at https://doi.org/10.5281/zenodo.18781450 (ref. [94]).

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

## Acknowledgements

We thank members of our laboratory, M. Laub, M. Buttner, S. Kamoun, J. Jones and S. Schlimpert, for helpful discussion and feedback on this paper. We thank R. Lo for preliminary strain construction and both J. Kaljević and M. Jordan for guidance with RNA-seq and phylogenetic analyses, respectively. Imaging was done at the University of York cryo-EM facility (supported by the Wellcome Trust (206161/Z/17/Z)), and at the Stanford-SLAC Cryo-EM Center (S2C2), which is supported by the National Institute of General Medical Sciences (1R24GM154186). The content is solely the responsibility of the authors and does not necessarily represent the official views of the National Institutes of Health. This work is supported by a Lister Institute Fellowship and Wellcome Trust Investigator grant 221776/Z/2/Z (to T.B.K.L.), a Royal Commission for the Exhibition of 1851 Fellowship (to E.J.B.), a Sir Henry Wellcome Fellowship (224067/Z/21/Z to P.B.), an independent fellowship from the Rowland Institute (to K.G.) and the BBSRC-funded Harnessing Biosynthesis for Sustainable Food and Health (HBio) Institute Strategic Programme BB/X01097X/1 (to the John Innes Centre).

## Author contributions

Conceptualization: E.J.B. and T.B.K.L. Methodology: E.J.B., P.B., N.T.T., P.M.N., K.G., B.S., A.M. and T.B.K.L. Investigation: E.J.B., P.B., N.T.T., P.M.N., K.G., B.S., A.M. and T.B.K.L. Visualization: E.J.B., P.B. and P.M.N. Funding acquisition: E.J.B., P.B., K.G. and T.B.K.L. Project administration: E.J.B. and T.B.K.L. Supervision: T.B.K.L. Writing—original draft: E.J.B., P.B., N.T.T., P.M.N., A.M. and T.B.K.L. Writing—review and editing: E.J.B. and T.B.K.L.

## Competing interests

The authors declare no competing interests.

## Additional information

**Extended data** is available for this paper at https://doi.org/10.1038/s41564-026-02316-4.

**Correspondence and requests for materials** should be addressed to Emma J. Banks or Tung B. K. Le.

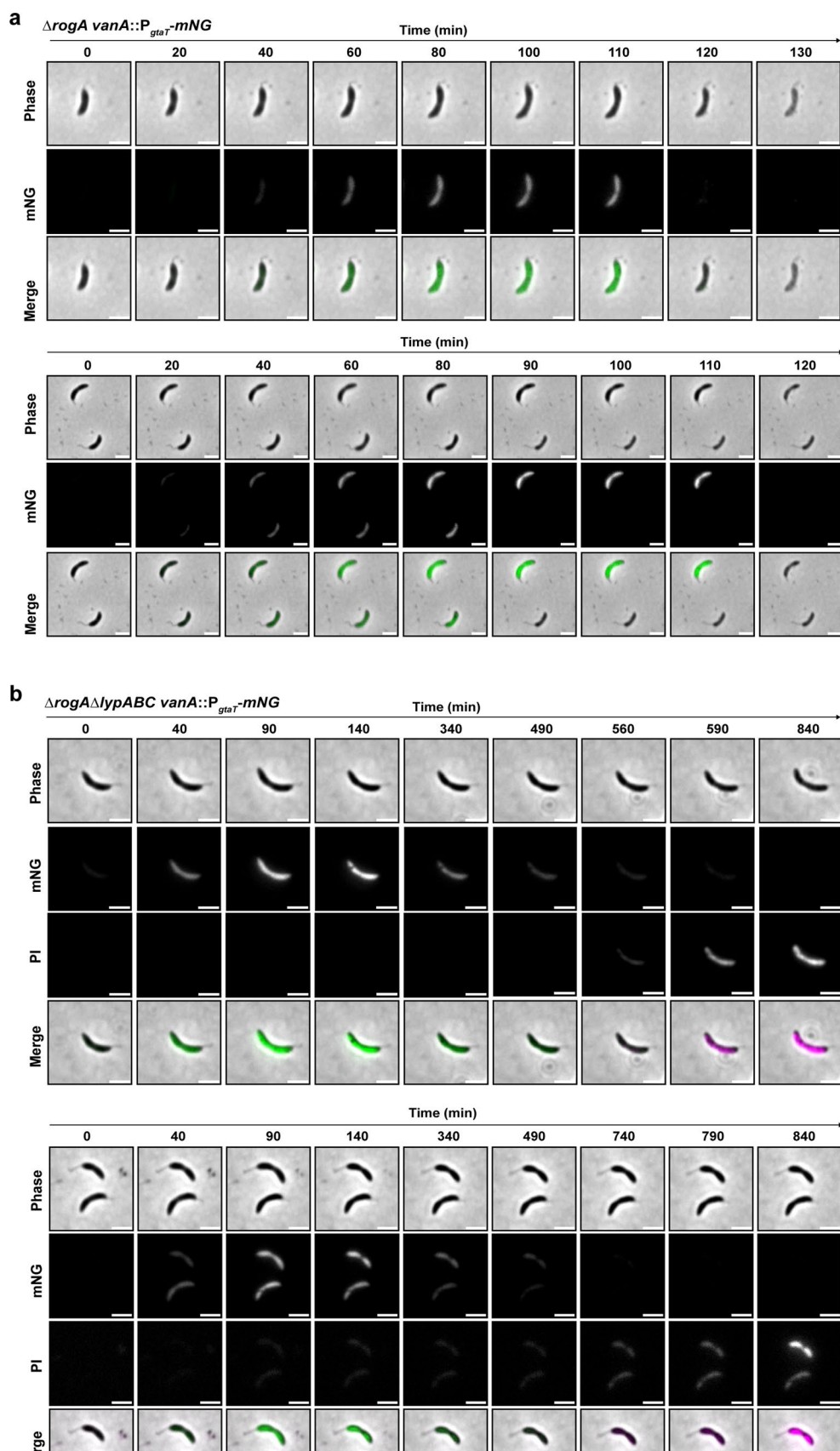

**Extended Data Fig. 1 | See next page for caption.**

**Extended Data Fig. 1 | Time-lapse fluorescence microscopy of *Caulobacter crescentus* GTA production and cell death. a**. Two additional examples of fluorescence microscopy images from time-lapse experiments performed with the Δ*rogA vanA*::P*gtaT*-*mNG* strain showing GTA activation (mNG signal) followed by cell lysis (transition to phase-light ghost cell). Time-lapses can be viewed in **Supplementary Video 2** and **Supplementary Video 3**. **b**. Two examples of fluorescence microscopy images from time-lapse experiments performed with the Δ*rogA*Δ*lypABC vanA*::P*gtaT*-*mNG* strain showing GTA activation (mNG signal), followed by cell death (PI signal) but no lysis or transition to a ghost cell. Time-lapses can be viewed in **Supplementary Video 4** and **Supplementary Video 5**. All data are representative of at least three independent repeats. Scale bar: 2 μm.

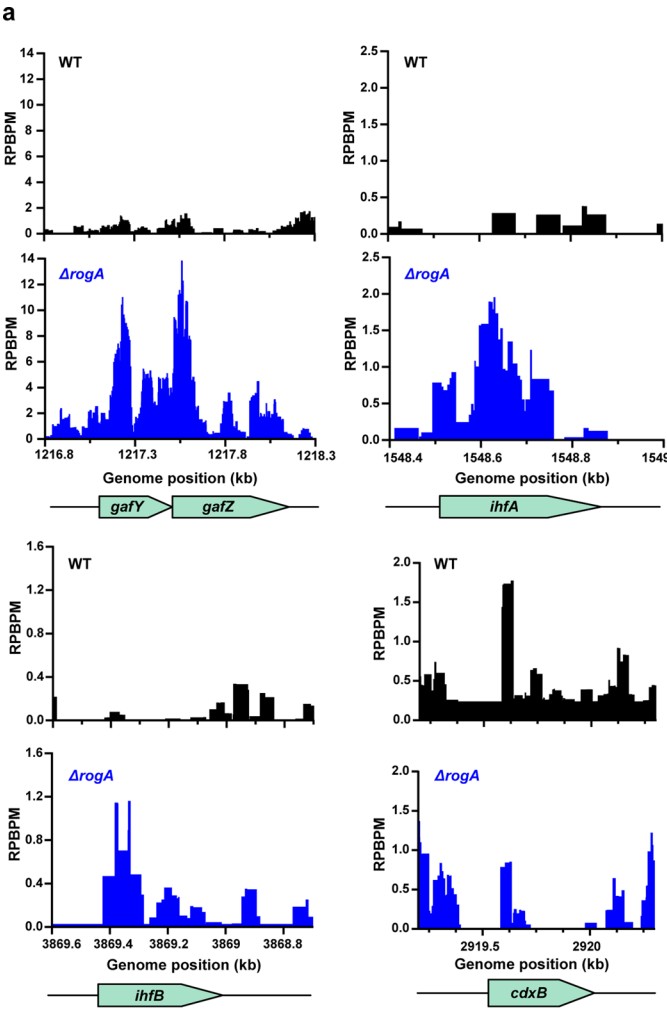

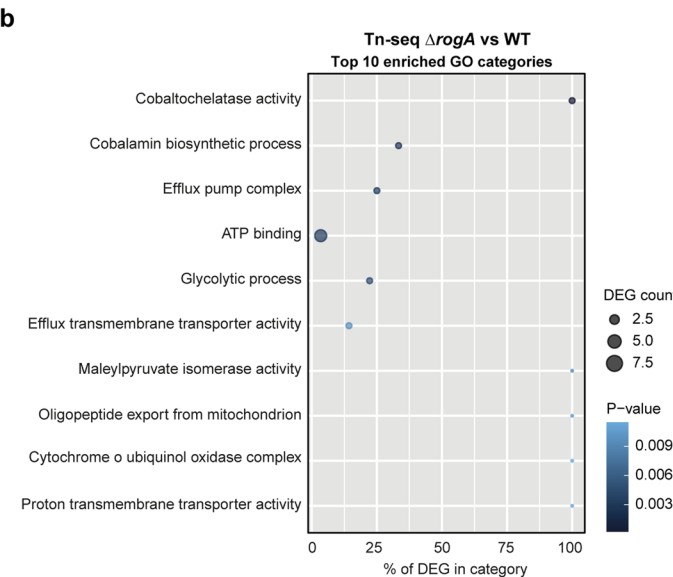

**Extended Data Fig. 2 | Tn-seq transposon insertion frequency and gene ontology enrichment analyses. a.** Plots from Tn-seq (Fig. 2b) showing the reads per base pair per million reads (RPBPM) at the genomic loci of *gafYZ* (top left), *ihfA* (top right), *ihfB* (bottom left) and *cdxB* (bottom right), indicating the frequency of transposon insertions in these genes within the Δ*rogA* strain (bottom, blue) compared to the wildtype strain (top, black). Three independent repeats were performed. **b.** Top ten gene ontology (GO) categories enriched in Tn-seq data comparing Δ*rogA* to wildtype. DESeq2 was used for differential expression analysis with default settings (Wald test followed by P-value adjustment for multiple comparisons using the Benjamini-Hochberg method). Data are presented as bubble plots in which the size of the bubble indicates the number of differentially expressed genes (DEG) within each GO category, and the bubble colour indicates the P-value (dark blue: more significant; light blue: less significant).

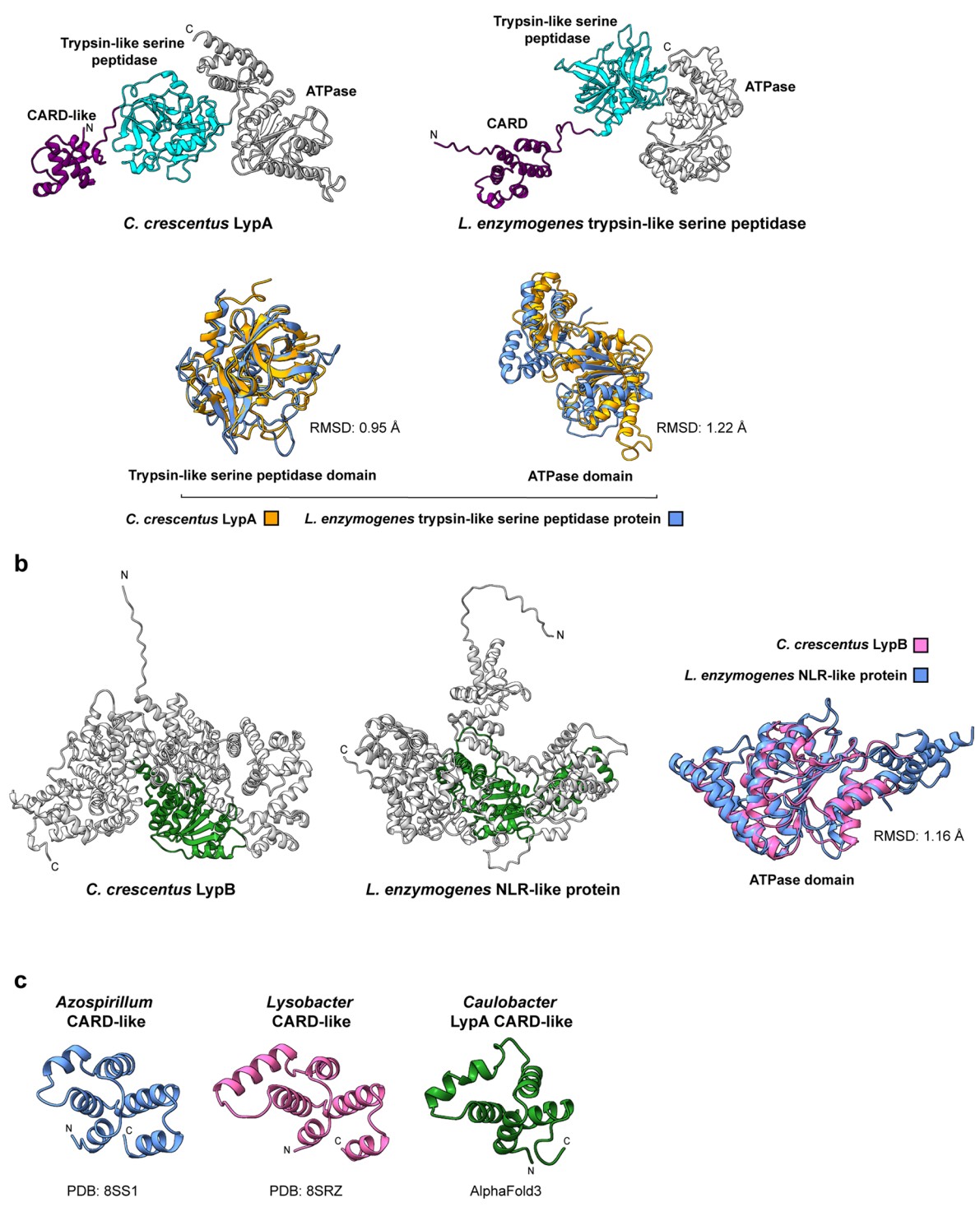

**Extended Data Fig. 3 | AlphaFold3 models of *C. crescentus* LypA and LypB resemble CARD-NLR homologues. a**. Top: AlphaFold3 models of *C. crescentus* LypA (top left) and the *L. enzymogenes* CARD-NLR system trypsin-like serine peptidase protein (top right) coloured according to domain organisation. Bottom: AlphaFold3 model alignments of the trypsin-like serine peptidase domains (bottom left) and ATPase domains (bottom right) in isolation. Orange: *C. crescentus* LypA; blue: *L. enzymogenes* trypsin-like serine peptidase. Trypsin-like peptidase RMSD: 0.95 Å, ATPase RMSD: 1.22 Å. **b**. AlphaFold3 models of *C. crescentus* LypB (left) and the *L. enzymogenes* CARD-NLR system NLR-like protein (middle) coloured according to domain organisation. Right: AlphaFold3 model alignment of the ATPase domains in isolation. Magenta: *C. crescentus* LypB; blue: *L. enzymogenes* NLR-like protein. RMSD: 1.16 Å. **c**. Comparison of crystal structures of the CARD-like domains of proteins encoded by the CARD-NLR anti-phage defence systems of *Azospirillum* sp. TSO35-2 (left) and *L. enzymogenes* (centre) with the AlphaFold3-predicted structure of the N-terminal CARD-like domain (residues 1-80) of LypA from *C. crescentus* (right). PDB identifiers are shown beneath structures.

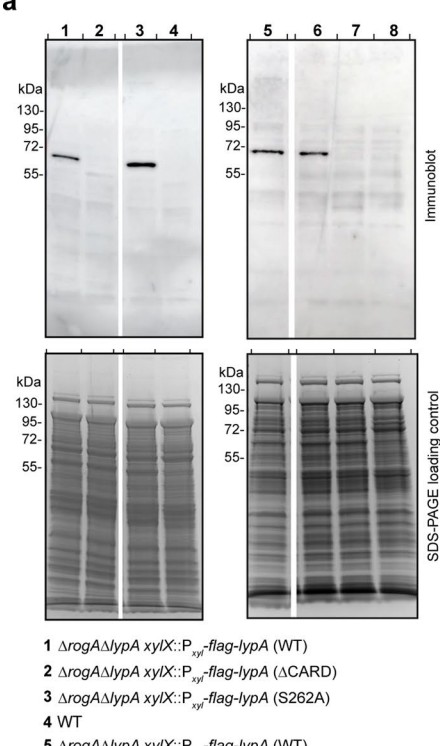

**a**

1　  ΔrogAΔlypA xylX::P$_{xyl}$-flag-lypA (WT)
2　  ΔrogAΔlypA xylX::P$_{xyl}$-flag-lypA (ΔCARD)
3　  ΔrogAΔlypA xylX::P$_{xyl}$-flag-lypA (S262A)
4　 WT
5　  ΔrogAΔlypA xylX::P$_{xyl}$-flag-lypA (WT)
6　  ΔrogAΔlypA xylX::P$_{xyl}$-flag-lypA (T356A, S360A, F361A)
7　  ΔrogAΔlypA xylX::P$_{xyl}$-flag-lypA (ΔATPase)
8　 WT

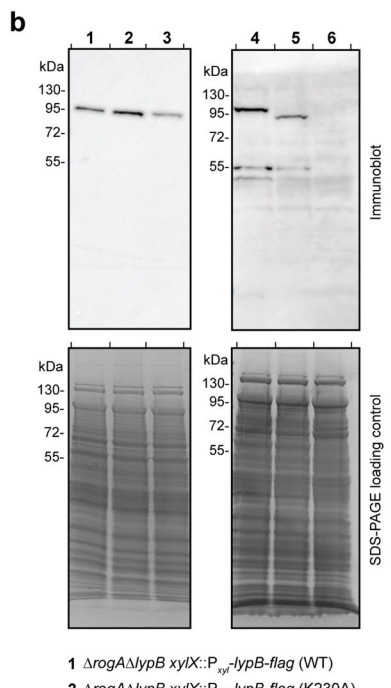

**b**

1　  ΔrogAΔlypB xylX::P$_{xyl}$-lypB-flag (WT)
2　  ΔrogAΔlypB xylX::P$_{xyl}$-lypB-flag (K230A)
3　  ΔrogAΔlypB xylX::P$_{xyl}$-lypB-flag (D335A)
4　  ΔrogAΔlypB xylX::P$_{xyl}$-lypB-flag (WT)
5　  ΔrogAΔlypB xylX::P$_{xyl}$-lypB-flag (ΔNTD)
6　 WT

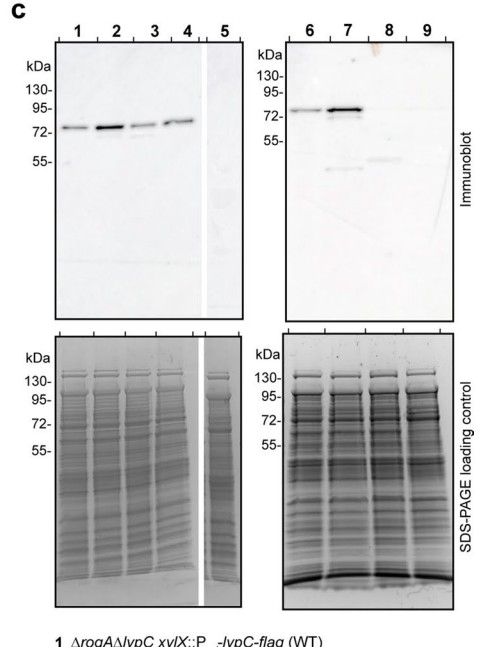

**c**

1　  ΔrogAΔlypC xylX::P$_{xyl}$-lypC-flag (WT)
2　  ΔrogAΔlypC xylX::P$_{xyl}$-lypC-flag (S296A)
3　  ΔrogAΔlypC xylX::P$_{xyl}$-lypC-flag (H471A)
4　  ΔrogAΔlypC xylX::P$_{xyl}$-lypC-flag (N505A)
5　 WT
6　  ΔrogAΔlypC xylX::P$_{xyl}$-lypC-flag (WT)
7　  ΔrogAΔlypC xylX::P$_{xyl}$-lypC-flag (H471A, N505A)
8　  ΔrogAΔlypC xylX::P$_{xyl}$-lypC-flag (Δendonuclease)
9　 WT

**Extended Data Fig. 4 | Production of LypABC protein variants in vivo.**
Immunoblots of total cell lysates from *C. crescentus* using a monoclonal anti-FLAG antibody showing production of either LypA protein variants (**a**), LypB protein variants (**b**), or LypC protein variants (**c**) tested in Fig. 4. The expected sizes for LypA, LypB and LypC are 65, 112, and 73 kDa, respectively. Separate Coomassie-stained SDS-PAGE gels were loaded with equal sample volumes to serve as loading controls. Immunoblots are representative of at least two independent experiments.

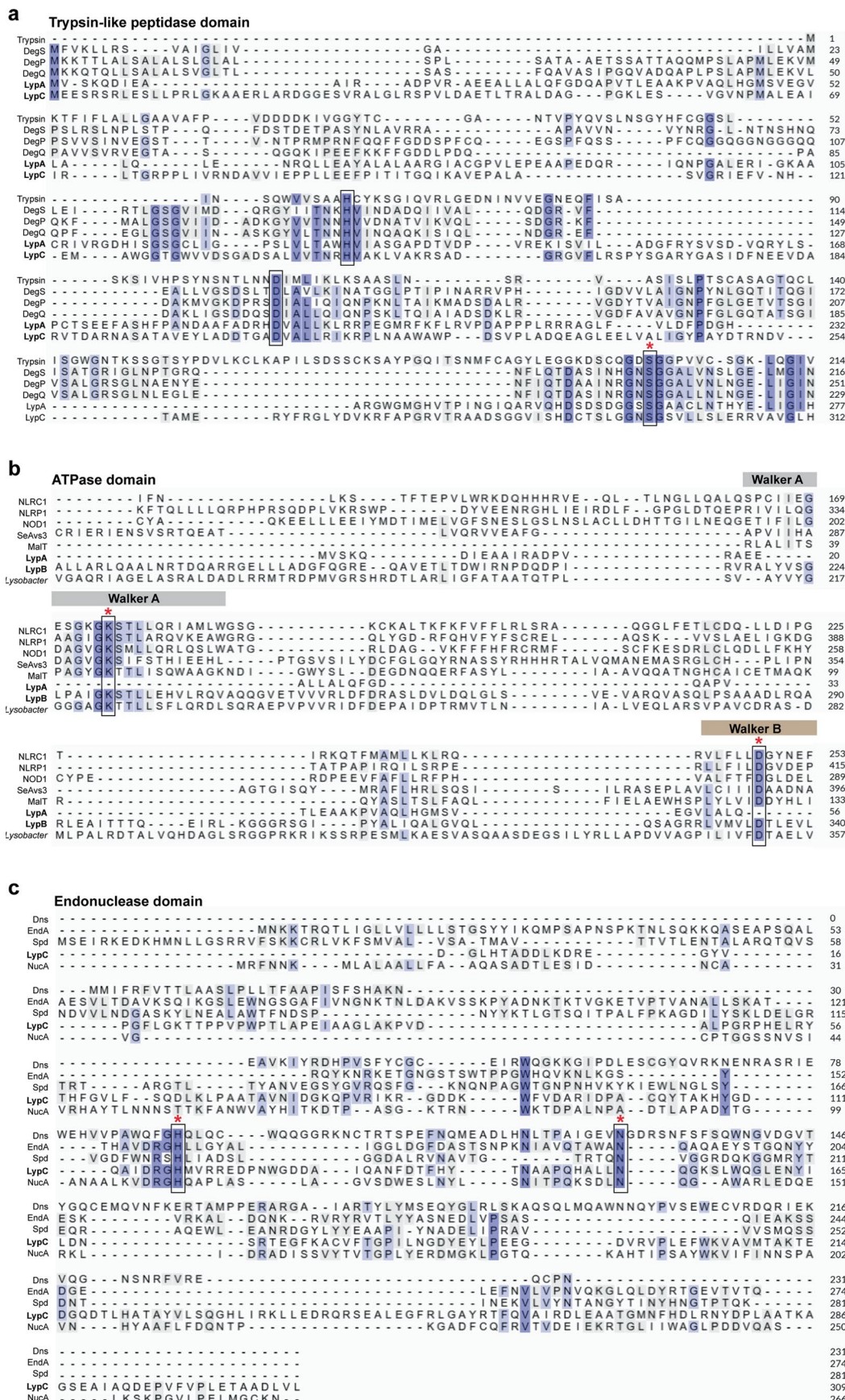

**Extended Data Fig. 5 | See next page for caption.**

**Extended Data Fig. 5 | Multiple sequence alignments of LypABC protein domains. a**. Sequence alignment of trypsin-like peptidase domains from Trypsin (*B. taurus*, NCBI accession: AAI34798.1), DegS (*E. coli*, NCBI accession: AAC44006.1), DegP (*E. coli*, NCBI accession: AAC73272.1), DegQ (*E. coli*, NCBI accession: AAC43992.1), LypA (*C. crescentus*, NCBI accession: AGJ94600.2), and LypC (*C. crescentus*, NCBI accession: ACL94044.1). **b**. Sequence alignment of nucleotide binding domains from NLRC1 (*H. sapiens*, NCBI accession: AAH31555), NLRP1 (*H. sapiens*, NCBI accession: AAG30288), NOD1 (*H. sapiens*, NCBI accession: AAD29125), SeAvs3 (S*almonella*, NCBI accession: WP_126523998.1), MalT (*E. coli*, NCBI accession: AAA8388.1), LypA (*C. crescentus*, NCBI accession: AGJ94600.2), LypB (*C. crescentus*, NCBI accession: ACL94045.1), and *Lysobacter* (*L. enzymogenes*, NCBI accession: ALN60087.1). **c**. Sequence alignment of the endonuclease domains from Dns (*V. cholerae*, NCBI accession: AAA27516.1), EndA (*S. pneumoniae*, NCBI accession: AAL00582.1), Spd (*S. pyogenes*, NCBI accession: ABF32934.1), LypC (*C. crescentus*, NCBI accession: ACL94044.1), and NucA (*S. marcescens*, NCBI accession: AAA26560.1). In all alignments, conserved catalytic residues are boxed and residues mutated in this work are denoted with a red asterisk.

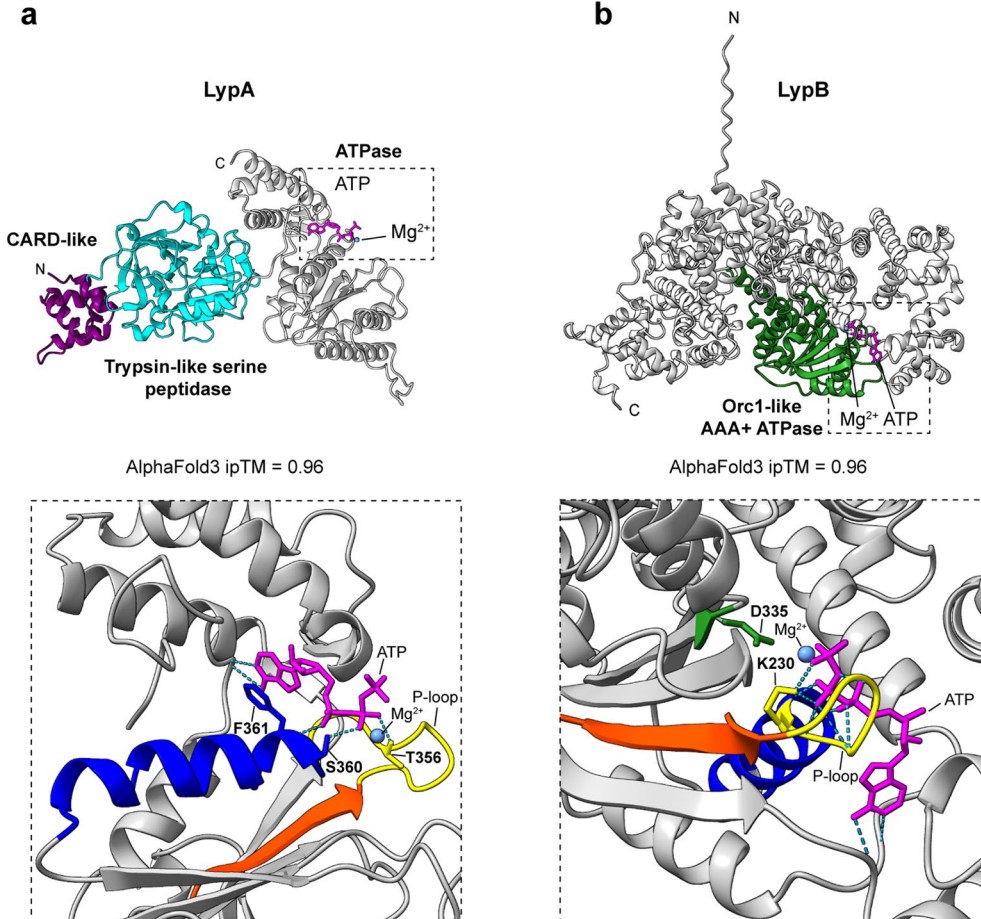

**Extended Data Fig. 6 | Structural predictions suggest LypA and LypB may bind ATP.** Top: AlphaFold3 structural predictions of either LypA (**a**) or LypB (**b**) in complex with ATP (stick-format, magenta) and a magnesium Mg²⁺ ion (blue). The high ipTM score (0.96 for both) suggests that both LypA and LypB may bind ATP. Domains are coloured according to structures presented in Fig. 4. Bottom: Magnified regions of the putative ATP-binding pockets for either LypA or LypB.

Magenta: ATP in stick-format, light blue: Mg²⁺ ion, blue helix: α-helix preceding the P-loop, yellow: P-loop, orange: β-strand following the loop, green: Walker B motif. Possible hydrogen bond contacts are shown as pale blue dashed lines between residues. Amino acids targeted for mutation are labelled in bold and shown in stick-format.

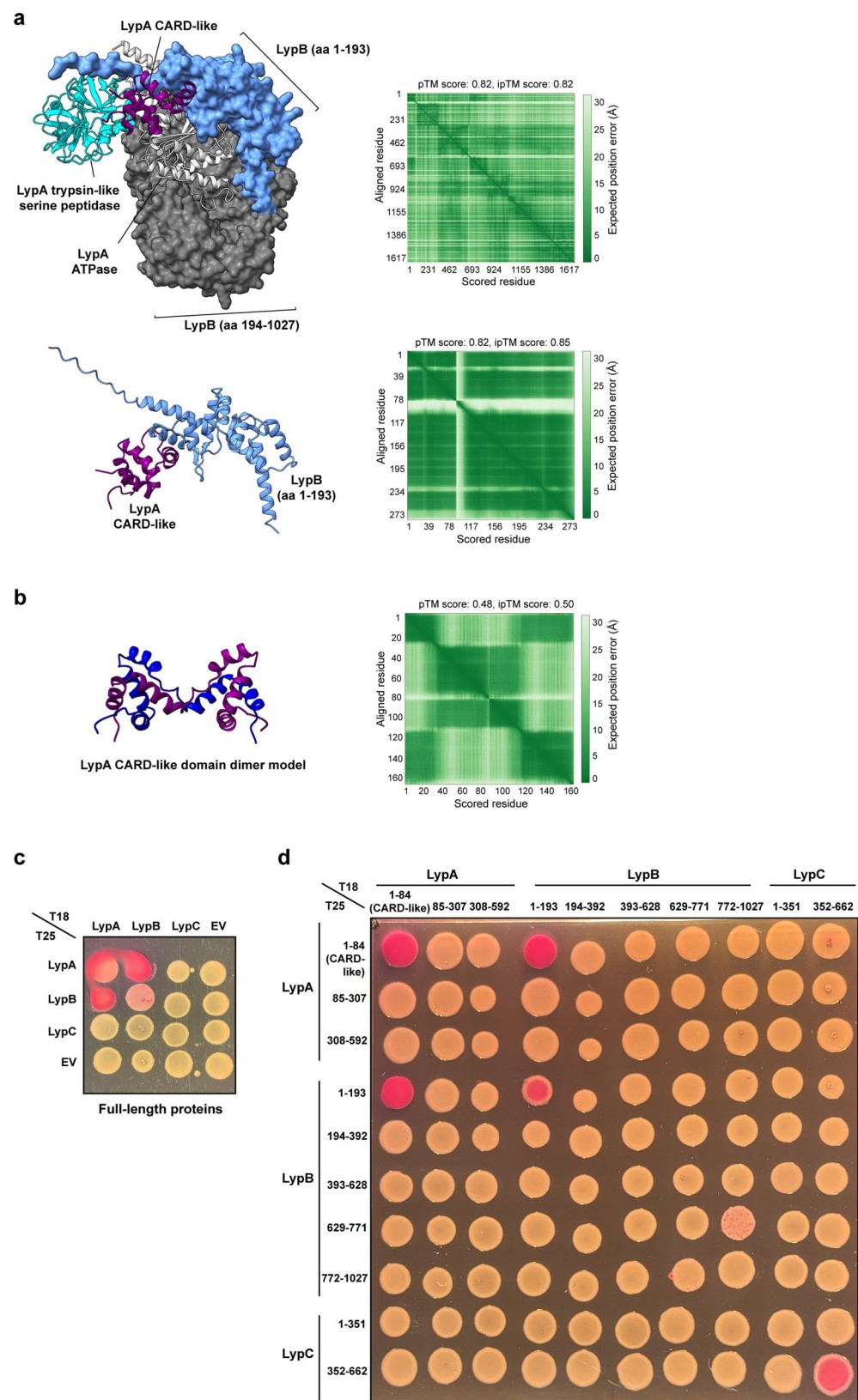

**Extended Data Fig. 7 | See next page for caption.**

**Extended Data Fig. 7 | LypA and LypB interact via the CARD-like domain of LypA. a**. AlphaFold3-Multimer models of LypA and LypB (top) and the CARD-like domain of LypA with the N-terminal domain of LypB (aa 1-193) (bottom). Predicted aligned error plots are shown to the right of each model with model confidence scores stated above each error plot. The CARD-like domain, trypsin-like serine peptidase domain and putative ATPase domains of LypA are coloured purple, cyan, and light grey, respectively and depicted in ribbon format. LypB is either depicted in surface solid format (top) or in ribbon format (bottom). LypB aa 1-193: blue; aa 194-1027: dark grey. **b**. AlphaFold3-Multimer model of the LypA CARD-like domain as a dimer (left) with the predicted aligned error plot and model confidence scores (right). **c**. Bacterial two-hybrid experiment testing interactions between full-length LypA, LypB, and LypC proteins in either T25 or T18 vectors. **d**. Bacterial two-hybrid assay showing interactions (red) or non-interactions (white) between every possible domain combination encoded by the proteins LypABC. Amino acid ranges for each domain are listed and domains were tested in both T18 and T25 vectors. For all two-hybrid experiments, three independent repeats were performed and representative images are shown.

**a**

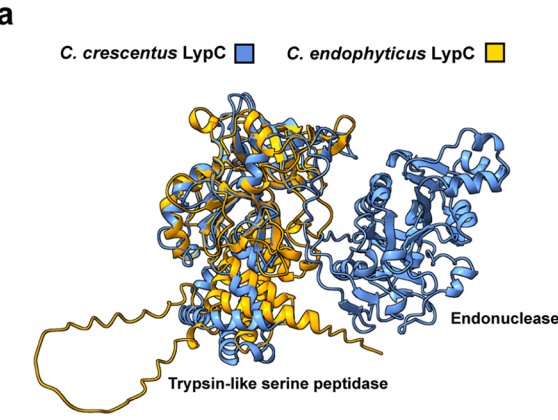

*C. crescentus* LypC ▢   *C. endophyticus* LypC ▢

Endonuclease

Trypsin-like serine peptidase

**b**

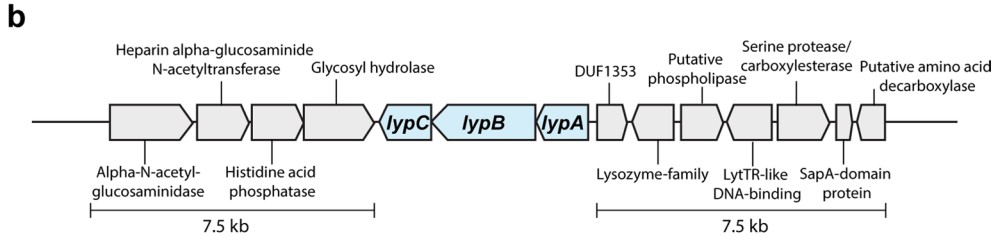

**c**

| *Caulobacter* species | GTA system | *lypABC* system |
|---|---|---|
| *C. crescentus* | ■ | ■ |
| *C. radicis* | ■ | ■ |
| *C. zeae* | ■ | ■ |
| *C. endophyticus* | ■ | ■ |
| *C. segnis* | ■ | ▢ |
| *C. henricii* | ■ | ▢ |
| *C. rhizosphaerae* | ■ | ▢ |
| *C. hibisci* | ■ | ▢ |
| *C. soli* | ■ | ▢ |
| *C. flavus* | ■ | ▢ |
| *C. mirabilis* | ■ | ▢ |

**Extended Data Fig. 8 | Genomic context and conservation of LypABC.**
**a**. AlphaFold3 structural predictions for LypC from either *C. crescentus* (blue) or *C. endophyticus* (orange), highlighting the absence of a C-terminal endonuclease domain in *C. endophyticus*. **b**. Schematic of the genomic context of *lypABC* within *C. crescentus* strain NA1000. The genes are not encoded within a defence island like many anti-phage defence systems. The putative functions of genes flanking *lypABC* are annotated. **c**. Predicted conservation of a GTA system and a complete *lypABC* system within different *Caulobacter* species (black box: present; white box: absent). Only four species encode both systems.

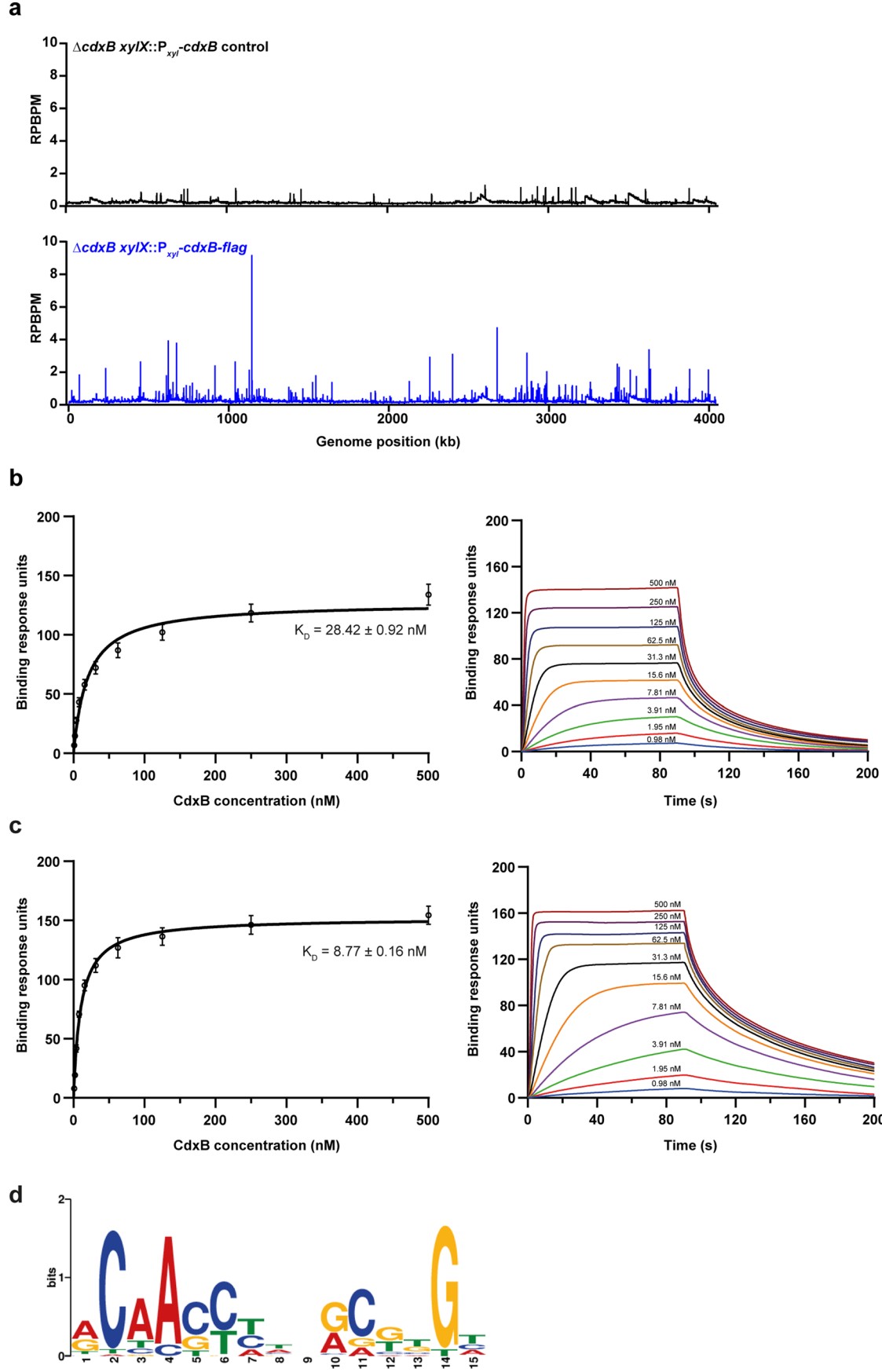

**Extended Data Fig. 9 | See next page for caption.**

**Extended Data Fig. 9 | CdxB genomic binding sites and binding affinities to the *lypABC* and *gafYZ* promoters. a**. ChIP-seq profiles showing CdxB binding sites across the *C. crescentus* NA1000 genome. A *cdxB-flag* (bottom profile, blue line) or untagged *cdxB* (top profile, black line) allele was expressed from the xylose promoter ($P_{xyl}$) in a $\Delta cdxB$ mutant. X-axis: genomic position (kb); y-axis: reads per base pair per million reads (RPBPM). Two independent repeats were performed and representative profiles are shown. **b-c**. Left: response curves of CdxB binding to either (**b**) $P_{lypABC}$ or (**c**) $P_{gafYZ}$, with $K_D$ values of 28.4 nM and 8.8 nM, respectively. Data show the mean ± s.d from four independent experiments. Right: sensorgrams used to generate the binding curves and calculate $K_D$ values. The different concentrations of CdxB protein are labelled above each trace. Data are representative of three independent experiments. **d**. CdxB DNA sequence consensus binding motif identified with MEME-ChIP using the top 100 most-enriched CdxB ChIP-seq peaks.

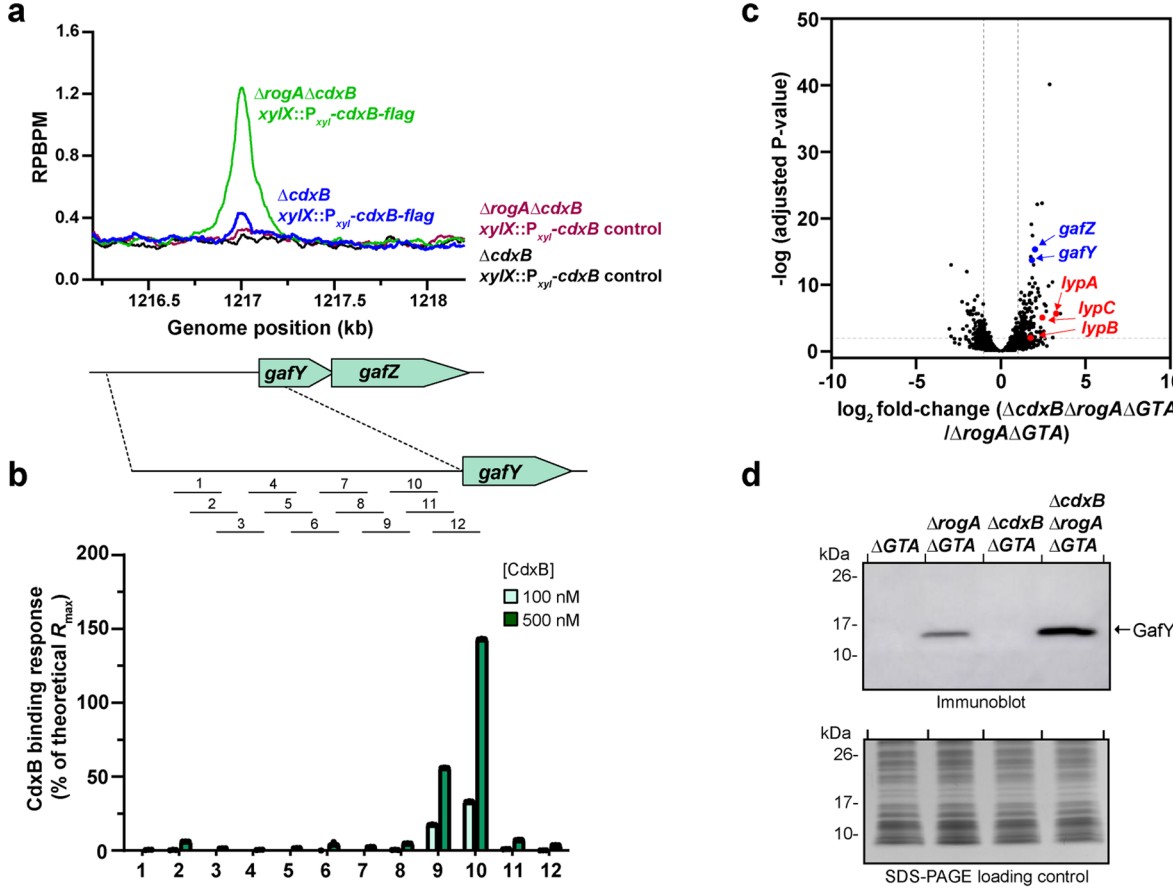

**Extended Data Fig. 10 | CdxB represses *gafYZ* expression to regulate GTA activation. a**. ChIP-seq profiles showing binding of CdxB to the *gafYZ* promoter. A *cdxB-flag* allele was expressed from the xylose promoter ($P_{xyl}$) in either a single Δ*cdxB* mutant (blue line) or a double Δ*rogA*Δ*cdxB* mutant (green line). Untagged *cdxB* controls are shown in black and maroon for the single and double mutant backgrounds, respectively. Two independent repeats were performed and representative profiles are shown. **b**. Surface plasmon resonance (SPR) showing binding of purified CdxB protein to the *gafYZ* promoter which was divided into twelve overlapping DNA fragments. Two different concentrations of CdxB protein were used: 100 nM (light green bars) and 500 nM (dark green bars). Data show the mean ± s.d from two independent experiments. **c**. RNA-seq volcano plot showing the log₂ fold-change for the Δ*cdxB*Δ*rogA*Δ*GTA* strain compared

to the Δ*rogA*Δ*GTA* strain. Grey lines: value of 2.0. *lypA*, *lypB*, *lypC*, *gafY*, and *gafZ* which are upregulated when *cdxB* is deleted, are annotated. DESeq2 was used for differential expression analysis with default settings (Wald test followed by P-value adjustment for multiple comparisons using the Benjamini-Hochberg method). Three independent repeats were performed. Top differentially expressed genes are presented as a heatmap in Supplementary Fig. 2b. **d**. Immunoblot (top) of total cell lysates from *C. crescentus* using a polyclonal anti-GafY antibody to show the levels of GafY protein in the indicated mutants. A separate Coomassie-stained SDS-PAGE gel was loaded with equal sample volumes to serve as a loading control (bottom). The immunoblot is representative of at least two independent experiments.

# Reporting Summary

## Statistics

For all statistical analyses, confirm that the following items are present in the figure legend, table legend, main text, or Methods section.

| n/a | Confirmed | |
|---|---|---|
| ☐ | ☒ | The exact sample size (*n*) for each experimental group/condition, given as a discrete number and unit of measurement |
| ☐ | ☒ | A statement on whether measurements were taken from distinct samples or whether the same sample was measured repeatedly |
| ☐ | ☒ | The statistical test(s) used AND whether they are one- or two-sided *Only common tests should be described solely by name; describe more complex techniques in the Methods section.* |
| ☒ | ☐ | A description of all covariates tested |
| ☐ | ☒ | A description of any assumptions or corrections, such as tests of normality and adjustment for multiple comparisons |
| ☐ | ☒ | A full description of the statistical parameters including central tendency (e.g. means) or other basic estimates (e.g. regression coefficient) AND variation (e.g. standard deviation) or associated estimates of uncertainty (e.g. confidence intervals) |
| ☐ | ☒ | For null hypothesis testing, the test statistic (e.g. *F*, *t*, *r*) with confidence intervals, effect sizes, degrees of freedom and *P* value noted *Give P values as exact values whenever suitable.* |
| ☒ | ☐ | For Bayesian analysis, information on the choice of priors and Markov chain Monte Carlo settings |
| ☒ | ☐ | For hierarchical and complex designs, identification of the appropriate level for tests and full reporting of outcomes |
| ☒ | ☐ | Estimates of effect sizes (e.g. Cohen's *d*, Pearson's *r*), indicating how they were calculated |

*Our web collection on statistics for biologists contains articles on many of the points above.*

## Software and code

Policy information about availability of computer code

| | |
|---|---|
| Data collection | Zeiss ZenBlue Version 2.3 microscope software, Amersham Imager 600 (GE Healthcare) western blot software |
| Data analysis | Adobe Illustrator v. 28 (Adobe), Excel 365 (Microsoft), Galaxy EU server, PyMOL v. 2.5.2, GraphPad Prism v. 10, UCSF Chimera X v. 1.9, ImageJ v. 1.53t, MicrobeJ v.13I, AlphaFold3 (https://alphafoldserver.com/). Custom code used to analyse Tn-seq and RNA-seq data has been deposited in GitHub and Zenodo repositories (https://doi.org/10.5281/zenodo.18781500 and https://doi.org/10.5281/zenodo.18781450). All information is included in the code availability section of the manuscript. |

For manuscripts utilizing custom algorithms or software that are central to the research but not yet described in published literature, software must be made available to editors and reviewers. We strongly encourage code deposition in a community repository (e.g. GitHub). See the Nature Portfolio guidelines for submitting code & software for further information.

## Data

Policy information about availability of data

All manuscripts must include a data availability statement. This statement should provide the following information, where applicable:
- Accession codes, unique identifiers, or web links for publicly available datasets
- A description of any restrictions on data availability
- For clinical datasets or third party data, please ensure that the statement adheres to our policy

> A data availability statement is included in the manuscript. All sequencing data generated in this study have been deposited in the GEO database under the accession codes: GSE295577 (ChIP-seq), GSE295580 (RNA-seq), and GSE295581 (Tn-seq).

## Research involving human participants, their data, or biological material

Policy information about studies with human participants or human data. See also policy information about sex, gender (identity/presentation), and sexual orientation and race, ethnicity and racism.

| | |
|---|---|
| Reporting on sex and gender | NA |
| Reporting on race, ethnicity, or other socially relevant groupings | *Please specify the socially constructed or socially relevant categorization variable(s) used in your manuscript and explain why they were used. Please note that such variables should not be used as proxies for other socially constructed/relevant variables (for example, race or ethnicity should not be used as a proxy for socioeconomic status).*<br>*Provide clear definitions of the relevant terms used, how they were provided (by the participants/respondents, the researchers, or third parties), and the method(s) used to classify people into the different categories (e.g. self-report, census or administrative data, social media data, etc.)*<br>*Please provide details about how you controlled for confounding variables in your analyses.* |
| Population characteristics | NA |
| Recruitment | NA |
| Ethics oversight | NA |

Note that full information on the approval of the study protocol must also be provided in the manuscript.

# Field-specific reporting

Please select the one below that is the best fit for your research. If you are not sure, read the appropriate sections before making your selection.

☒ Life sciences  ☐ Behavioural & social sciences  ☐ Ecological, evolutionary & environmental sciences

For a reference copy of the document with all sections, see nature.com/documents/nr-reporting-summary-flat.pdf

# Life sciences study design

All studies must disclose on these points even when the disclosure is negative.

| | |
|---|---|
| Sample size | No statistical test was used to determine sample size. These were based on well-established protocols that provide reliable and representative data. All sequencing experimental datasets represent the population average of millions of bacteria. Microscopy quantification was carried out with n=400 cells were biological repeat to ensure that a large number of cells were analysed equally across biological replicates. |
| Data exclusions | No data were excluded from the analyses |
| Replication | All experiments were performed at least twice to ensure reproducibility, and similar results were obtained throughout. |
| Randomization | Not relevant to this study. All strains were selected randomly for inoculation from plate and grown under the same conditions. Observable differences are due to the difference in the genotype of each strain. |
| Blinding | Not relevant to this study. Observed differences are due to the genotype of the analysed strain. |

# Reporting for specific materials, systems and methods

We require information from authors about some types of materials, experimental systems and methods used in many studies. Here, indicate whether each material, system or method listed is relevant to your study. If you are not sure if a list item applies to your research, read the appropriate section before selecting a response.

## Materials & experimental systems

| n/a | Involved in the study |
|---|---|
| ☐ | ☒ Antibodies |
| ☒ | ☐ Eukaryotic cell lines |
| ☒ | ☐ Palaeontology and archaeology |
| ☒ | ☐ Animals and other organisms |
| ☒ | ☐ Clinical data |
| ☒ | ☐ Dual use research of concern |
| ☒ | ☐ Plants |

## Methods

| n/a | Involved in the study |
|---|---|
| ☐ | ☒ ChIP-seq |
| ☒ | ☐ Flow cytometry |
| ☒ | ☐ MRI-based neuroimaging |

# Antibodies

| Antibodies used | Monoclonal α-FLAG M2-Peroxidase HRP-conjugated antibody (Merck, 1:5000 dilution) cat number A8592. Polyclonal antibody against GtaL (1:1000 dilution of anti-serum, custom synthesis by Biosynth Laboratories, UK). Polyclonal antibody against GafY (3:1000 dilution of anti-serum, custom synthesis by Biosynth Laboratories, UK). Secondary antibody: HRP-conjugated goat anti-rabbit (Abcam, 1:10,000 dilution), cat number 6721. |
|---|---|
| Validation | The specificity of all antibodies used in this study was verified against lysates from deletion mutant strains. Both polyclonal antibodies have been used and reported in previous publications (Tran & Le, Nature Communications, 2024). Validation of the commercial anti-FLAG antibody was based on the technical data sheets from the manufacturer. |

# Plants

| Seed stocks | Not relevant |
|---|---|
| Novel plant genotypes | Not relevant |
| Authentication | Not relevant |

# ChIP-seq

## Data deposition

☒ Confirm that both raw and final processed data have been deposited in a public database such as GEO.

☒ Confirm that you have deposited or provided access to graph files (e.g. BED files) for the called peaks.

| Data access links *May remain private before publication.* | Reviewer token edszuogwttovdgf (for the GSE295577 ChIP-seq data: https://www.ncbi.nlm.nih.gov/geo/query/acc.cgi?acc=GSE295577) |
|---|---|
| Files in database submission | Raw fastq sequencing files and processed files (MACS2 callpeak data) are deposited in the GEO database. ChIP-seq file list: TLE11: delta cdxB xylX::Pxyl-cdxB negative control rep 1 TLE12: delta cdxB xylX::Pxyl-cdxB negative control rep 2 TLE13: delta cdxB xylX::Pxyl-cdxB-flag rep 1 TLE14: delta cdxB xylX::Pxyl-cdxB-flag rep 2 TLE15: delta rogA delta cdxB xylX::Pxyl-cdxB negative control rep 1 TLE16: delta rogA delta cdxB xylX::Pxyl-cdxBnegative control rep 2 TLE17: delta rogA delta cdxB xylX::Pxyl-cdxB-flag rep 1 TLE18: delta rogA delta cdxB xylX::Pxyl-cdxB-flag rep 2 |
| Genome browser session (e.g. UCSC) | Not applicable because there is no UCSC browser for the reference genome of the bacterium Caulobacter crescentus NA1000. However, all processed data have been uploaded to GEO and are available to the public and ChIP-seq profiles are show in the manuscript. |

## Methodology

| Replicates | Two biological replicates were performed for ChIP-seq experiments. Peaks described in the manuscript were further validated by biochemical experiments (surface plasmon resonance) |
|---|---|

| Sequencing depth | Reads were short (50-100 bp) and paired-end with high FastQC scores. The mean total number of reads per sample was 5.9 million of which 4.8 million mapped uniquely |
| --- | --- |
| Antibodies | ANTI-FLAG M2 Affinity Gel (Merck, cat number A2220) https://www.sigmaaldrich.com/GB/en/product/sigma/a2220 |
| Peak calling parameters | All analysis was done using the Galaxy platform (https://usegalaxy.eu/). Reads were mapped with Hisat2 and peaks were called using MACS2 callpeak (comparing test sample to negative control (non-flag-tagged gene) with a significance cut-off q-value of <0.01) |
| Data quality | On average 5.9 million reads were mapped per sample and quality of reads was checked with FastQC (Galaxy platform). The minimum FDR (q-value) for peak detection used was <0.01. We report the number of peaks <2-fold enrichment. All peaks are listed in Table S2 and Table S4. All peaks described in the manuscript were inspected visually in both biological replicate experiments. |
| Software | All data processing was done using the Galaxy platform (https://usegalaxy.eu/). For analysis of ChIP-seq data, Illumina short reads were mapped back to the Caulobacter crescentus NA1000 genome with Hisat2. The coverage at each nucleotide position was computed using bedtools genome-cov. MACS2 callpeak was used to call peaks. ChIP-seq profiles were plotted using GraphPad Prism software. |

