## [Peer Review File · Nature Microbiology]

A bacterial CARD-NLR-like immune system controls the release of gene transfer agents

Corresponding Author: Professor Tung Le

Version 0:

Reviewer comments:

Reviewer #1

(Remarks to the Author)

The manuscript by Banks et al. describes a mechanism which *Caulobacter crescentus* uses to initiate cell lysis and release Gene Transfer Agents (GTAs). GTAs are domesticated prophages that facilitate horizontal gene transfer. GTAs have been coopted from phages and here, the authors show that the cellular lysis initiation mechanisms have been coopted from antiphage systems. This work is an exciting next chapter in our understanding of how phage (and antiphage) genes have impacted the biology of bacteria.

The authors discover the LypABC system, characterize its function primarily using genetics, and provide additional insights into transcriptional regulation. There are two major issues identified that should be addressed to improve the support for the conclusions, detailed below.

Major Comments

1. The connection of LypABC to CARD-NLR systems is provocative and a major point of emphasis in the manuscript. The sequence/structure comparisons should be more firmly supported. The authors draw the conclusion that LypABC evolved divergently from CARD-NLR systems. This is a challenging argument to make based on tools like DefenseFinder and FoldSeek. The question is whether the NTPase + CARD domains of LypB are more significantly related to the NTPase+CARD domains in CARD-NLR systems than to other NTPase or CARD domains found in non-antiphage systems—or even found in different ORFs. DefenseFinder does not index non-immune proteins and simply returns the immune pathway most related to the query, leading to confirmation bias. One could imagine a scenario where LypB is actually the parent of the CARD-NLR system, and that antiphage activity emerged as an adaptation later. Alternatively, perhaps LypB is a novel reformulation of a distantly related NTPase and a distantly related CARD that have, by chance, combined into a new formulation. At issue is whether this is convergent evolution of architecture or divergent evolution of sequence. The authors should (1) analyze the sequence relationships of LypB to CARD-NLR systems in detail and (2) discuss the limitations of the speculation on relationship to CARD-NLR. See also Minor Point 1 about nomenclature.
2. The authors should show a role for LypABC in a GTA functional assay. While LypABC is clearly required for lysis/cell death, and the microscopy/tomography supports a role for GTA particle dissemination, there is no experiment in the manuscript that demonstrates GTA-mediated gene transfer is impaired. Could the authors place a selectable marker on the chromosome and measure GTA-mediated generalized transduction of that marker into a naïve recipient? This should be greatly impaired by mutations in LypABC and could, in theory, show a phenotype for the enigmatic nuclease domain of LypC.

Minor Comments

1. Typically, the term NLR is reserved for proteins that encode “nucleotide-binding and leucine-rich repeats”. Here, the LRR is lacking and should be clarified in the text. A useful term that helps with this differentiation is NLR-related. Another point to consider is that the LypB system appears highly similar to mammalian APAF-1. This protein is a STAND NTPase (AP-ATPase) that is fused to a CARD domain. Because this protein lacks LRRs, it is not classified as an NLR. However, it appears more grossly similar to LypB due to its non-immune role in programmed cell death. This could also be included in the discussion.

Aaron Whiteley

Reviewer #2

(Remarks to the Author)

In this manuscript, the authors showed that a phage defence system composed of 3 proteins (LypABC) has been repurposed for

cell lysis and ghost cell formation upon GTA activation in *Caulobacter crescentus*. Using bioinformatic tools and AlphaFold predictions, they found that LypABC belong to the CARD-NLR protein family. By using targeted mutagenesis, they inactivated the catalytic activity of the domains of the 3 Lyp proteins and tested their importance in ghost cell formation. Finally, they found a second transcriptional regulator (RogB) that represses expression of lypABC but which also has multiple other target genes in vivo.

The discovery of the LypABC system as inducing the lysis of *Caulobacter* cells is novel and very interesting from an evolutionary point of view. However, the characterization of the Lyp proteins remains somewhat limited here and there are still many open questions about this system. On the other hand, the description of RogB appears a little off topic in this manuscript. Of course not entirely since it represses the expression of lypABC but the transition from the incomplete characterisation of LypABC to the discovery of RogB is a little frustrating and also let many open questions about this repressor. In addition, CCNA_02755, called RogB by the authors, has already been characterized in another publication cited by the authors (McLaughlin et al., 2023), in which the XRE transcription regulator was called CdxB. The CdxB-dependent regulon has already been described with ChIP-seq and RNA-seq data in this publication. Even a link between GTA activation and CdxB is described in this manuscript.

Thus, for all these reasons, I would suggest to the authors to further characterise LypABC and focus only on this repurposed lytic system leaving CdxB (RogB) out of this story.

I have several questions and suggestions for the LypABC system.

- Why the role of the ATPase domain of LypA hasn't been addressed? At least a truncated variant lacking this domain could be used in the complementation assay?

- Deleting the CarD domain makes LypA unstable and useless. But based on the AlphaFold3-Multimer model of the LypA_CarD:LypB complex, wouldn't it be possible to predict pairs of residues at the interface important for the interaction? Based on that, loss of interaction mutants could be first tested by BTH and then in vivo for cell lysis. Same for the homodimerization of the CarD domain. Mutations disrupting CarD homodimer formation should be tested. This would bring important information about the role played by the CarD domain in this process.

- It would be good to show multiple sequence alignments for each domain of the 3 proteins, not only nucleotide binding domain of LypB, to highlight the residues targeted as catalytic mutants. There is also a problem with the length of LypC since it is annotated as a 599 aa protein in NA1000 and as a 695 aa protein in GB15 while the authors talk about a protein of 662 aa. These numbers should be clarified.

- I agree with the conclusion that the endonuclease activity of LypC is not required for the lysis since the catalytic mutants H471A and N505A still led to lysis. But there are many examples of enzymatic domains repurposed for new functions. In that case, this domain could be important but independently of the endonuclease activity. A LypC variant truncated for the endonuclease domain should be tested.

- Based on the following statement, considering that the trypsin-like serine peptidase protein and NLR-like protein of the L. enzymogenes anti-phage system interact, the authors tested the interaction between LypA, which harbours a trypsin-like serine peptidase, and LypB, which is a NLR-like protein harboring a ATPase domain. However LypC also harbours a trypsin-like serine peptidase domain and LypA has a predicted ATPase domain. The potential interactions between LypC and LypA or LypB should also be tested. The truncated variants as well as the catalytic mutants should be tested as well. This will allow to better understand the formation of the complex and to check whether the catalytic activities (ATPase and peptidase) are important for the complex formation or for the cell lysis (or both).

- Did the authors try to overexpress individually (with Pcumate) each of the lyp genes and compare the toxicity to the lypABC overexpression?

- LypABC is required for ghost cell formation (cell lysis) but not for DNA packaging. Hence Δ rogA Δ lypABC cells die without lysing likely due to host genome digestion, suggesting that LypABC is critical for disseminating the capsid full of DNA. To firmly conclude that, the authors should measure the capacity of Δ lypABC mutant strains to transfer genetic material from donor to recipient in a way similar to the one described in Gozzi et al., 2022.

- There are other gene candidates than lypABC identified in the Tn-seq screen (at least 41) but the authors do not describe them. It would be nice to have a few words about these other candidates. Is there any COG enrichment or genes belonging to the same pathway?

- The presence of two trypsin-like serine peptidase domains, one in LypA and one in LypC is not discussed while both of them are required for ghost cell formation. This should be discussed.

Then, I have a several question about the GTA mechanism.

- Why GTA is required for ghost cells formation in Δ rogA?

- If GTA is required for ghost cell formation, why LypABC is toxic and leads to in WT cells where GTA is repressed? Did the authors test the overexpression of lypABC in Δ GTA Δ rogA? Is there any difference when compared to Δ rogA single mutant?

- The authors state (L329-332): Since LypABC are produced in GTA-off wildtype cells, we reason that the system remains

autoinhibited until GTA activation occurs to prevent untimely and unproductive host cell lysis.

On which data is based this statement that LypABC are produced in WT where RogB represses its expression?

Related to the previous questions, how LypABC can be toxic in WT if GTA is off?

Finally I have a few questions/comments on the transcriptional regulation

- To avoid confusion, RogB should be called CdxB as in McLaughlin et al., 2023. A deep comparison with the transcriptomic data described in McLaughlin et al., 2023 should be done to keep at best only the original information.

- There are several transcriptomic data (RNA-seq and ChIP-seq) about the RogB repressor. However, except lypABC and gafYZ, the authors do not describe at all the RogB regulon.

Is there any COG enrichment in this regulon that might help understanding in which conditions RogB is inactivated? The authors could provide some further analysis of this regulon.

- We easily understand why GTA has to be tightly repressed given the toxicity of GTA and Lyp proteins. However, the system has to be de-repressed in some conditions. Based on previous publication, entry to stationary phase increased GTA production and DNA transfer.

The authors could therefore check whether the protein levels of RogA and/or RogB decrease at this growth stage. If so, a proteolytic mechanism could be investigated. Alternatively, RogB could be inactivated by protein:protein interactions, so the authors could look for protein interacting with RogB.

Are there other conditions known to induce GTA? If so, similar experiments should be done in the same conditions.

Reviewer #3

(Remarks to the Author)

Recently discovered gene transfer agent in *Caulobacter crescentus* (hereafter, CcGTA) is not yet as well understood as its counterpart in another alphaproteobacterium, *Rhodobacter capsulatus* (RcGTA). In this manuscript, Banks et al. identify and characterize genes and proteins that are required for the release of CcGTA particles. Their findings are exciting for two reasons. First, the CcGTA release mechanism is very different from that of RcGTA (which uses holin-endolysin system that is typically employed by bona fide phages.) Second, the CcGTA system appears to be co-opted from a phage defense system, which is an unexpected and novel finding. Thus, I think the manuscript would be of interest not only to "GTA enthusiasts", but to a broader audience of microbiologists and evolutionary biologists. I support publication of this work in *Nature Microbiology*.

Overall, the manuscript is well-written and easy to follow, with an informative introduction and the results that are logically described and augmented by well-organized figures. Although, I think the authors could broaden and strengthen their Discussion by highlighting the importance of their findings to more general (i.e., beyond GTA particle release) understanding of how GTAs evolved from their viral ancestors. Because the CcGTA core GTA cluster is indisputably homologous to RcGTA head-tail cluster (first observed in Shakya et al., 2017; PMID 29250433), they must have shared a common viral ancestor. But, clearly, *Rhodobacter* and *Caulobacter* diverged in the "solution" for GTA particle release. I think this is worth mentioning in the Discussion (but I leave this for the authors to decide).

Additional requests for a few minor modifications:

Lines 97-99: please list % of the cells, to be consistent with the previous sentence that list the data. It is not possible to extract the % from Figure 1B itself due to small numbers.

Line 99: there should be a comma before "i.e."

Lines 196-197: "While structural modelling of LypA with ATP and Mg²⁺ by AlphaFold3 suggested potential ATP-binding ability (Figure S6A), canonical Walker A/B motifs are absent". Could this putative ATPase domain contain ATP-grasp fold (the only other known ATP-binding fold) instead of P-loop? I believe this is possible to check bioinformatically.

Lines 218-219: The reference to Figure 8B is incorrect here, not sure which figure the authors wanted to refer to here.

Line 227: I believe "CARDS" should be "CARDs".

Lines 603-608: This section should provide more details about parameters and databases used in searches. Specifically, a) Please list database(s) against which BLASTP searches were carried out. List version of the BLAST program used. If the search was carried out via web, date of search should be listed, as databases change over time. Default parameters of the BLAST programs change over time, so it would be preferable for reproducibility if the authors would list the parameters such as E-value cutoff, substitution matrix and low complexity filtering. Insert reference for the BLAST program (PMID: 9254694).

b) For FoldSeek, please list databases and search mode used

c) For InterProScan, please provide dates of searches as their databases change over time.

d) For HHPred, please list database(s) and parameters used.

Supplementary Figures and Tables:

a) Supplementary Figure S12 should be referenced in the text (it is currently not).

b) Supplementary Figure S13 is only referenced in the legends of the Figures 6 and 7. Ideally, it should be referenced in the text.

c) Supplementary Materials and Methods should be referenced in the text (they are currently not).

d) Tables S1-S5 should have table caption as the top line in the Excel spreadsheet, to improve readability.

Reviewer #4

(Remarks to the Author)

The paper by Banks et al., addresses the mechanism of regulation of LypABC, to study how bacterial defence system are involved in the release of GTAs. The authors used genetic screenings to identify bacterial immunity operons involved in GTA-mediated host cell lysis in *C. crescentus* and validated lypABC as a crucial operon regulating the GTA particle release. LypABC

is a homologue of caspase recruitment domain-nucleotide binding leucine-rich repeat (CARD-NLR) anti-phage defence systems that trigger abortive infection. They also showed how the overexpression of LypABC is highly toxic which together with other observations was interpreted as a putative immunity-related activity.

This is a solid and well written manuscript that deconstructs different aspects of the regulation of of GTAs. The main criticism I have is that lypABC was not validated as a defence system and while a homologue to other CARD-NLR, lypABC is actually non-canonical with an effector that seems non-essential for the phenotype; without experimental validation the work feels lacking. Given the functional variety and plasticity of these mobile operons it could still be plausible that lypABC is just another functional cog in the GTA machine...

Main comments:

The FoldSeek searches that are shown also involve AF predictions, I was curious to know what were the results against the PDB?

Page 6: Given that the active site is non-canonical it would be helpful to provide in Fig. 4 of S6 a detailed view of the predicted ATP binding mode and its coordination, how are the base and 3P coordinated and how does it compares with known binding modes, are there obvious catalytic residues?

As part of the characterization of LypC, the protease domain was inactivated by replacing a catalytic Ser which correlates with the loss of lytic function, however when this was tried for the endonuclease domain the single point substitutions were not sufficient to change the phenotype. For some enzymes active site residues are redundant and some single substitutions are not enough to switch off the activity. So would it be possible to validate the effect of the mutations in vitro? Perhaps combine several? In the same way, some systems repurpose enzymatic domains as regulatory domains, given that in *C. endophyticus* the endonuclease is not present, could the authors evaluate the effect of a truncate. These are necessary controls that need to be provided to support the claim that the cell lysis is endonuclease independent.

One of the most interesting findings of the paper is the discovery that RogB is a transcriptional regulator of the system that also controls the gafYZ operon. The authors use SPR as a qualitative method to show the specific binding to the lypABC promoter, however very little is then done or pursued in terms of the actual characterization of the interaction of RogB and DNA. 1) Is there a reason why the Kd for DNA could not be measured with SPR? 2) What type of DNA sequence is RogB recognizing (inverted repeats, palindromes)? 3) Is there a sequence motif shared between the promoter of lypABC and gafYZ? And if different, given the dual repressor hypothesis suggested in the manuscript, are such differences translated in different affinities? Given that RogB can be produced, even simple EMSA experiments can provide significant insights to the regulatory process.

Minor issues:

Ref 28 can now be updated, the work was published in May 2025.

In page 3 at the end of the introduction the authors state: "AlphaFold-Multimer and bacterial two-hybrid assays indicate that LypA and LypB form a co-complex mediated by the CARD-like domain of LypA". This should be rephrased, AF-multimer is not an experimental method and should not be used as an indication of a complex-formation no matter how high the cofolding scores are.

The same can be say about the final sentence of the introduction: "In summary, we present evidence that immune systems are versatile and can perform new biological functions that are beneficial, rather than antagonistic, to MGEs." This is a very general and strong statement, however the authors didn't actually show that lypABC is involved in defence, so without experimental validation this needs to be tone down, the paper has other significant achievements that are supported by experiments...

Reviewer #5

(Remarks to the Author)

Decision Letter:

20th July 2025

Dear Professor Le,

Thank you for your patience while your manuscript "A bacterial CARD-NLR immune system controls the release of gene transfer agents" was under peer-review at Nature Microbiology. It has now been seen by 5 referees, whose expertise and comments you will find at the end of this email. Although they find your work of some potential interest, they have raised a number of concerns that will need to be addressed before we can consider publication of the work in Nature Microbiology.

In particular, the referees noted that more work is needed to demonstrate that LypABC is a phage defence system and one referee suggested that a little more work might be needed to confirm whether it is an NLR-CARD system or perhaps another type of system as they suggest in their comments. The referees also note that it's important to demonstrate that LypABC activity is promoting transfer of the GTA as well. Then there are a number of comments that focus around building the evidence for LypABC activity through experiments that further analyse which domains are doing what, further explore the function of RogB and when it's repression might be alleviated. One of the referees suggested removing the analysis of RogB function. We have discussed this editorially, and would suggest that you keep this aspect of the work (since it gives further insight into the regulation of LypABC and the GTA) but use the previously published name for this factor for consistency with the literature. These are the critical points which need to be addressed in a revised manuscript. There are a few other points among the reviewer reports which should be straightforward and feasible to address.

Should further experimental data allow you to address these criticisms, we would be happy to look at a revised manuscript.

Please include a data availability statement as a separate section after Methods but before references, under the heading "Data Availability". This section should inform readers about the availability of the data used to support the conclusions of your study. This information includes accession codes to public repositories (data banks for protein, DNA or RNA sequences, microarray, proteomics data etc...), references to source data published alongside the paper, unique identifiers such as URLs to data repository entries, or data set DOIs, and any other statement about data availability. At a minimum, you should include the following statement: "The data that support the findings of this study are available from the corresponding author upon request", mentioning any restrictions on availability. If DOIs are provided, we also strongly encourage including these in the Reference list (authors, title, publisher (repository name), identifier, year). For more guidance on how to write this section please see: <http://www.nature.com/authors/policies/data/data-availability-statements-data-citations.pdf>

* If you have not done so already we suggest that you begin to revise your manuscript so that it conforms to our Article format instructions at <http://www.nature.com/nmicrobiol/info/final-submission>. Refer also to any guidelines provided in this letter.

When submitting the revised version of your manuscript, please pay close attention to our [href="https://www.nature.com/nature-portfolio/editorial-policies/image-integrity">Digital Image Integrity Guidelines.](https://www.nature.com/nature-portfolio/editorial-policies/image-integrity) and to the following points below:

EXTENDED DATA FIGURES

Link Redacted

Note: This url links to your confidential homepage and associated information about manuscripts you may have submitted or be reviewing for us. If you wish to forward this e-mail to co-authors, please delete this link to your homepage first.

Nature Microbiology is committed to improving transparency in authorship. As part of our efforts in this direction, we are now requesting that all authors identified as 'corresponding author' on published papers create and link their Open Researcher and Contributor Identifier (ORCID) with their account on the Manuscript Tracking System (MTS), prior to acceptance. This applies to primary research papers only. ORCID helps the scientific community achieve unambiguous attribution of all scholarly contributions. You can create and link your ORCID from the home page of the MTS by clicking on 'Modify my Springer Nature account'. For more information please visit www.springernature.com/orcid.

If you wish to submit a suitably revised manuscript we would hope to receive it within 6 months. If you cannot send it within this time, please let us know. We will be happy to consider your revision, even if a similar study has been accepted for publication at

Nature Microbiology or published elsewhere (up to a maximum of 6 months).

Yours sincerely,

Reviewer Expertise:

Referee #1: phage defence systems
Referee #2: Caulobacter
Referee #3: GTAs and horizontal gene transfer
Referee #4: phage defence systems

Reviewer Comments:

Reviewer #1 (Remarks to the Author):

The manuscript by Banks et al. describes a mechanism which *Caulobacter crescentus* uses to to initiate cell lysis and release Gene Transfer Agents (GTAs). GTAs are domesticated prophages that facilitate horizontal gene transfer. GTAs have been coopted from phages and here, the authors show that the cellular lysis initiation mechanisms have been coopted from antiphage systems. This work is an exciting next chapter in our understanding of how phage (and antiphage) genes have impacted the biology of bacteria.

The authors discover the LypABC system, characterize its function primarily using genetics, and provide additional insights into transcriptional regulation. There are two major issues identified that should be addressed to improve the support for the conclusions, detailed below.

Major Comments

1. The connection of LypABC to CARD-NLR systems is provocative and a major point of emphasis in the manuscript. The sequence/structure comparisons should be more firmly supported. The authors draw the conclusion that LypABC evolved divergently from CARD-NLR systems. This is a challenging argument to make based on tools like DefenseFinder and FoldSeek. The question is whether the NTPase + CARD domains of LypB are more significantly related to the NTPase+CARD domains in CARD-NLR systems than to other NTPase or CARD domains found in non-antiphage systems—or even found in different ORFs. DefenseFinder does not index non-immune proteins and simply returns the immune pathway most related to the query, leading to confirmation bias. One could imagine a scenario where LypB is actually the parent of the CARD-NLR system, and that antiphage activity emerged as an adaptation later. Alternatively, perhaps LypB is a novel reformulation of a distantly related NTPase and a distantly related CARD that have, by chance, combined into a new formulation. At issue is whether this is convergent evolution of architecture or divergent evolution of sequence. The authors should (1) analyze the sequence relationships of LypB to CARD-NLR systems in detail and (2) discuss the limitations of the speculation on relationship to CARD-NLR. See also Minor Point 1 about nomenclature.
2. The authors should show a role for LypABC in a GTA functional assay. While LypABC is clearly required for lysis/cell death, and the microscopy/tomography supports a role for GTA particle dissemination, there is no experiment in the manuscript that demonstrates GTA-mediated gene transfer is impaired. Could the authors place a selectable marker on the chromosome and measure GTA-mediated generalized transduction of that marker into a naïve recipient? This should be greatly impaired by mutations in LypABC and could, in theory, show a phenotype for the enigmatic nuclease domain of LypC.

Minor Comments

1. Typically, the term NLR is reserved for proteins that encode “nucleotide-binding and leucine-rich repeats”. Here, the LRR is lacking and should be clarified in the text. A useful term that helps with this differentiation is NLR-related. Another point to consider is that the LypB system appears highly similar to mammalian APAF-1. This protein is a STAND NTPase (AP-ATPase) that is fused to a CARD domain. Because this protein lacks LRRs, it is not classified as an NLR. However, it appears more grossly similar to LypB due to its non-immune role in programmed cell death. This could also be included in the discussion.

Aaron Whiteley

Reviewer #2 (Remarks to the Author):

In this manuscript, the authors showed that a phage defence system composed of 3 proteins (LypABC) has been repurposed for cell lysis and ghost cell formation upon GTA activation in *Caulobacter crescentus*. Using bioinformatic tools and AlphaFold predictions, they found that LypABC belong to the CARD-NLR protein family. By using targeted mutagenesis, they inactivated the catalytic activity of the domains of the 3 Lyp proteins and tested their importance in ghost cell formation. Finally, they found a second transcriptional regulator (RogB) that represses expression of lypABC but which also has multiple other target genes in vivo.

The discovery of the LypABC system as inducing the lysis of Caulobacter cells is novel and very interesting from an evolutionary point of view. However, the characterization of the Lyp proteins remains somewhat limited here and there are still many open questions about this system. On the other hand, the description of RogB appears a little off topic in this manuscript. Of course not entirely since it represses the expression of lypABC but the transition from the incomplete characterisation of LypABC to the discovery of RogB is a little frustrating and also let many open questions about this repressor. In addition, CCNA_02755, called RogB by the authors, has already been characterized in another publication cited by the authors (McLaughlin et al., 2023), in which the XRE transcription regulator was called CdxB. The CdxB-dependent regulon has already been described with ChIP-seq and RNA-seq data in this publication. Even a link between GTA activation and CdxB is described in this manuscript.

Thus, for all these reasons, I would suggest to the authors to further characterise LypABC and focus only on this repurposed lytic system leaving CdxB (RogB) out of this story.

I have several questions and suggestions for the LypABC system.

- Why the role of the ATPase domain of LypA hasn't been addressed? At least a truncated variant lacking this domain could be used in the complementation assay?

- Deleting the CarD domain makes LypA unstable and useless. But based on the AlphaFold3-Multimer model of the LypA_CarD:LypB complex, wouldn't it be possible to predict pairs of residues at the interface important for the interaction? Based on that, loss of interaction mutants could be first tested by BTH and then in vivo for cell lysis. Same for the homodimerization of the CarD domain. Mutations disrupting CarD homodimer formation should be tested. This would bring important information about the role played by the CarD domain in this process.

- It would be good to show multiple sequence alignments for each domain of the 3 proteins, not only nucleotide binding domain of LypB, to highlight the residues targeted as catalytic mutants. There is also a problem with the length of LypC since it is annotated as a 599 aa protein in NA1000 and as a 695 aa protein in CB15 while the authors talk about a protein of 662 aa. These numbers should be clarified.

- I agree with the conclusion that the endonuclease activity of LypC is not required for the lysis since the catalytic mutants H471A and N505A still led to lysis. But there are many examples of enzymatic domains repurposed for new functions. In that case, this domain could be important but independently of the endonuclease activity. A LypC variant truncated for the endonuclease domain should be tested.

- Based on the following statement, considering that the trypsin-like serine peptidase protein and NLR-like protein of the L. enzymogenes anti-phage system interact, the authors tested the interaction between LypA, which harbours a trypsin-like serine peptidase, and LypB, which is a NLR-like protein harboring a ATPase domain. However LypC also harbours a trypsin-like serine peptidase domain and LypA has a predicted ATPase domain. The potential interactions between LypC and LypA or LypB should also be tested. The truncated variants as well as the catalytic mutants should be tested as well. This will allow to better understand the formation of the complex and to check whether the catalytic activities (ATPase and peptidase) are important for the complex formation or for the cell lysis (or both).

- Did the authors try to overexpress individually (with Pcumate) each of the lyp genes and compare the toxicity to the lypABC overexpression?

- LypABC is required for ghost cell formation (cell lysis) but not for DNA packaging. Hence $\Delta\text{rogA}\Delta\text{lypABC}$ cells die without lysing likely due to host genome digestion, suggesting that LypABC is critical for disseminating the capsid full of DNA. To firmly conclude that, the authors should measure the capacity of ΔlypABC mutant strains to transfer genetic material from donor to recipient in a way similar to the one described in Gozzi et al., 2022.

- There are other gene candidates than lypABC identified in the Tn-seq screen (at least 41) but the authors do not describe them. It would be nice to have a few words about these other candidates. Is there any COG enrichment or genes belonging to the same pathway?

- The presence of two trypsin-like serine peptidase domains, one in LypA and one in LypC is not discussed while both of them are required for ghost cell formation. This should be discussed.

Then, I have a several question about the GTA mechanism.

- Why GTA is required for ghost cells formation in ΔrogA ?

- If GTA is required for ghost cell formation, why LypABC is toxic and leads to in WT cells where GTA is repressed? Did the authors test the overexpression of lypABC in $\Delta\text{GTA}\Delta\text{rogA}$? Is there any difference when compared to ΔrogA single mutant?

- The authors state (L329-332): Since LypABC are produced in GTA-off wildtype cells, we reason that the system remains autoinhibited until GTA activation occurs to prevent untimely and unproductive host cell lysis. On which data is based this statement that LypABC are produced in WT where RogB represses its expression? Related to the previous questions, how LypABC can be toxic in WT if GTA is off?

Finally I have a few questions/comments on the transcriptional regulation

- To avoid confusion, RogB should be called CdxB as in McLaughlin et al., 2023. A deep comparison with the transcriptomic data described in McLaughlin et al., 2023 should be done to keep at best only the original information.

- There are several transcriptomic data (RNA-seq and ChIP-seq) about the RogB repressor. However, except lypABC and gafYZ, the authors do not describe at all the RogB regulon. Is there any COG enrichment in this regulon that might help understanding in which conditions RogB is inactivated? The authors could provide some further analysis of this regulon.

- We easily understand why GTA has to be tightly repressed given the toxicity of GTA and Lyp proteins. However, the system has to be de-repressed in some conditions. Based on previous publication, entry to stationary phase increased GTA production and DNA transfer.

The authors could therefore check whether the protein levels of RogA and/or RogB decrease at this growth stage. If so, a proteolytic mechanism could be investigated. Alternatively, RogB could be inactivated by protein:protein interactions, so the authors could look for protein interacting with RogB.

Are there other conditions known to induce GTA? If so, similar experiments should be done in the same conditions.

Reviewer #3 (Remarks to the Author):

Recently discovered gene transfer agent in *Caulobacter crescentus* (hereafter, CcGTA) is not yet as well understood as its counterpart in another alphaproteobacterium, *Rhodobacter capsulatus* (RcGTA). In this manuscript, Banks et al. identify and characterize genes and proteins that are required for the release of CcGTA particles. Their findings are exciting for two reasons. First, the CcGTA release mechanism is very different from that of RcGTA (which uses holin-endolysin system that is typically employed by bona fide phages.) Second, the CcGTA system appears to be co-opted from a phage defense system, which is an unexpected and novel finding. Thus, I think the manuscript would be of interest not only to "GTA enthusiasts", but to a broader audience of microbiologists and evolutionary biologists. I support publication of this work in *Nature Microbiology*.

Overall, the manuscript is well-written and easy to follow, with an informative introduction and the results that are logically described and augmented by well-organized figures. Although, I think the authors could broaden and strengthen their Discussion by highlighting the importance of their findings to more general (i.e., beyond GTA particle release) understanding of how GTAs evolved from their viral ancestors. Because the CcGTA core GTA cluster is indisputably homologous to RcGTA head-tail cluster (first observed in Shakya et al., 2017; PMID 29250433), they must have shared a common viral ancestor. But, clearly, *Rhodobacter* and *Caulobacter* diverged in the "solution" for GTA particle release. I think this is worth mentioning in the Discussion (but I leave this for the authors to decide).

Additional requests for a few minor modifications:

Lines 97-99: please list % of the cells, to be consistent with the previous sentence that list the data. It is not possible to extract the % from Figure 1B itself due to small numbers.

Line 99: there should be a comma before "i.e."

Lines 196-197: "While structural modelling of LypA with ATP and Mg²⁺ by AlphaFold3 suggested potential ATP-binding ability (Figure S6A), canonical Walker A/B motifs are absent". Could this putative ATPase domain contain ATP-grasp fold (the only other known ATP-binding fold) instead of P-loop? I believe this is possible to check bioinformatically.

Lines 218-219: The reference to Figure 8B is incorrect here, not sure which figure the authors wanted to refer to here.

Line 227: I believe "CARDS" should be "CARDS".

Lines 603-608: This section should provide more details about parameters and databases used in searches. Specifically, a) Please list database(s) against which BLASTP searches were carried out. List version of the BLAST program used. If the search was carried out via web, date of search should be listed, as databases change over time. Default parameters of the BLAST programs change over time, so it would be preferable for reproducibility if the authors would list the parameters such as E-value cutoff, substitution matrix and low complexity filtering. Insert reference for the BLAST program (PMID: 9254694).

b) For FoldSeek, please list databases and search mode used

c) For InterProScan, please provide dates of searches as their databases change over time.

d) For HHPred, please list database(s) and parameters used.

Supplementary Figures and Tables:

a) Supplementary Figure S12 should be referenced in the text (it is currently not).

b) Supplementary Figure S13 is only referenced in the legends of the Figures 6 and 7. Ideally, it should be referenced in the text.

c) Supplementary Materials and Methods should be referenced in the text (they are currently not).

d) Tables S1-S5 should have table caption as the top line in the Excel spreadsheet, to improve readability.

Reviewer #4 (Remarks to the Author):

The paper by Banks et al., addresses the mechanism of regulation of LypABC, to study how bacterial defence system are involved in the release of GTAs. The authors used genetic screenings to identify bacterial immunity operons involved in GTA-mediated host cell lysis in *C. crescentus* and validated lypABC as a crucial operon regulating the GTA particle release. LypABC is a homologue of caspase recruitment domain-nucleotide binding leucine-rich repeat (CARD-NLR) anti-phage defence systems that trigger abortive infection. They also showed how the overexpression of LypABC is highly toxic which together with other observations was interpreted as a putative immunity-related activity.

This is a solid and well written manuscript that deconstructs different aspects of the regulation of of GTAs. The main criticism I have is that lypABC was not validated as a defence system and while a homologue to other CARD-NLR, lypABC is actually

non-canonical with an effector that seems non-essential for the phenotype; without experimental validation the work feels lacking. Given the functional variety and plasticity of these mobile operons it could still be plausible that lypABC is just another functional cog in the GTA machine...

Main comments:

The FoldSeek searches that are shown also involve AF predictions, I was curious to know what were the results against the PDB?

Page 6: Given that the active site is non-canonical it would be helpful to provide in Fig. 4 of S6 a detailed view of the predicted ATP binding mode and its coordination, how are the base and 3P coordinated and how does it compare with known binding modes, are there obvious catalytic residues?

As part of the characterization of LypC, the protease domain was inactivated by replacing a catalytic Ser which correlates with the loss of lytic function, however when this was tried for the endonuclease domain the single point substitutions were not sufficient to change the phenotype. For some enzymes active site residues are redundant and some single substitutions are not enough to switch off the activity. So would it be possible to validate the effect of the mutations in vitro? Perhaps combine several? In the same way, some systems repurpose enzymatic domains as regulatory domains, given that in *C. endophyticus* the endonuclease is not present, could the authors evaluate the effect of a truncate. These are necessary controls that need to be provided to support the claim that the cell lysis is endonuclease independent.

One of the most interesting findings of the paper is the discovery that RogB is a transcriptional regulator of the system that also controls the gafYZ operon. The authors use SPR as a qualitative method to show the specific binding to the lypABC promoter, however very little is then done or pursued in terms of the actual characterization of the interaction of RogB and DNA. 1) Is there a reason why the K_d for DNA could not be measured with SPR? 2) What type of DNA sequence is RogB recognizing (inverted repeats, palindromes)? 3) Is there a sequence motif shared between the promoter of lypABC and gafYZ? And if different, given the dual repressor hypothesis suggested in the manuscript, are such differences translated in different affinities? Given that RogB can be produced, even simple EMSA experiments can provide significant insights to the regulatory process.

Minor issues:

Ref 28 can now be updated, the work was published in May 2025.

In page 3 at the end of the introduction the authors state: "AlphaFold-Multimer and bacterial two-hybrid assays indicate that LypA and LypB form a co-complex mediated by the CARD-like domain of LypA". This should be rephrased, AF-multimer is not an experimental method and should not be used as an indication of a complex-formation no matter how high the cofolding scores are.

The same can be said about the final sentence of the introduction: "In summary, we present evidence that immune systems are versatile and can perform new biological functions that are beneficial, rather than antagonistic, to MGEs." This is a very general and strong statement, however the authors didn't actually show that lypABC is involved in defence, so without experimental validation this needs to be toned down, the paper has other significant achievements that are supported by experiments...

Version 1:

Reviewer comments:

Reviewer #1

(Remarks to the Author)

The authors have satisfied all of my concerns and I congratulate them on their work.

Reviewer #2

(Remarks to the Author)

All the points raised by the reviewers have been addressed and new data requested by the reviewers have been incorporated in the revised manuscript, further supporting the main conclusions. In particular, the new gene transfer experiments further support the critical role played by LypABC in GTA dissemination.

In the present form, I do not have any other concern or request for this ms.

Reviewer #3

(Remarks to the Author)

The authors satisfactorily addressed my concerns.

I also appreciate the inclusion of a phylogenetic tree of LypB and its homologs (Fig 4c), which was done in response to another reviewer but is an excellent additional information regarding LypB relationship to other CARD-NLR-like systems.

Reviewer #4

(Remarks to the Author)

As noted in my previous report, this is an interesting and generally well-constructed study, and the authors have addressed most of my comments in the revised manuscript. However, the question of a putative endonuclease activity for LypC remains unresolved.

In the revised version, substitution of several residues in the endonuclease domain predicted to be catalytic, results in no detectable phenotype and fully phenocopies the wild-type protein. Moreover, the truncated endonuclease LypC version is unstable, precluding meaningful functional or biochemical interpretation. As a consequence, there is no experimental evidence

supporting the presence of endonuclease activity, nor is such activity directly tested in the manuscript. It is well established that nuclease-like domains are frequently repurposed during evolution as non-catalytic modules involved in nucleic acid binding, protein recruitment, or allosteric regulation. This provides an alternative plausible explanation for the conservation of this domain within the LypC family without implying enzymatic activity. Accordingly, the authors should refrain from assigning a nuclease function to LypC unless catalytic activity is experimentally demonstrated (as they do with the lack of gasdermin-like activity). The functional annotation and interpretation of this domain should therefore be revised in the final version of the manuscript.

Decision Letter:

Our ref: NMICROBIOL-25051825A

9th February 2026

Dear Dr. Le,

Thank you for submitting your revised manuscript "A bacterial CARD-NLR immune system controls the release of gene transfer agents" (NMICROBIOL-25051825A). It has now been seen by the original referees and their comments are below. The reviewers find that the paper has improved in revision, and therefore we'll be happy in principle to publish it in Nature Microbiology, pending minor revisions to satisfy the referees' final requests and to comply with our editorial and formatting guidelines.

Thank you again for your interest in Nature Microbiology Please do not hesitate to contact me if you have any questions.

Sincerely,

Reviewer #1 (Remarks to the Author):

The authors have satisfied all of my concerns and I congratulate them on their work.

Reviewer #2 (Remarks to the Author):

All the points raised by the reviewers have been addressed and new data requested by the reviewers have been incorporated in the revised manuscript, further supporting the main conclusions. In particular, the new gene transfer experiments further support the critical role played by LypABC in GTA dissemination. In the present form, I do not have any other concern or request for this ms.

Reviewer #3 (Remarks to the Author):

The authors satisfactorily addressed my concerns. I also appreciate the inclusion of a phylogenetic tree of LypB and its homologs (Fig 4c), which was done in response to another reviewer but is an excellent additional information regarding LypB relationship to other CARD-NLR-like systems.

Reviewer #4 (Remarks to the Author):

As noted in my previous report, this is an interesting and generally well-constructed study, and the authors have addressed most of my comments in the revised manuscript. However, the question of a putative endonuclease activity for LypC remains unresolved.

In the revised version, substitution of several residues in the endonuclease domain predicted to be catalytic, results in no detectable phenotype and fully phenocopies the wild-type protein. Moreover, the truncated endonuclease LypC version is unstable, precluding meaningful functional or biochemical interpretation. As a consequence, there is no experimental evidence supporting the presence of endonuclease activity, nor is such activity directly tested in the manuscript. It is well established that nuclease-like domains are frequently repurposed during evolution as non-catalytic modules involved in nucleic acid binding, protein recruitment, or allosteric regulation. This provides an alternative plausible explanation for the conservation of this domain within the LypC family without implying enzymatic activity. Accordingly, the authors should refrain from assigning a nuclease function to LypC unless catalytic activity is experimentally

demonstrated (as they do with the lack of gasdermin-like activity). The functional annotation and interpretation of this domain should therefore be revised in the final version of the manuscript.

Version 2:

Decision Letter:

5th March 2026

Dear Tung,

I am very pleased to accept your Article "A bacterial CARD-NLR-like immune system controls the release of gene transfer agents" for publication in Nature Microbiology. Thank you for having chosen to submit your work to us and many congratulations.

Authors may need to take specific actions to achieve compliance with funder and institutional open access mandates. If your research is supported by a funder that requires immediate open access (e.g. according to [a href="https://www.springernature.com/gp/open-science/plan-s-compliance"> Plan S principles](https://www.springernature.com/gp/open-science/plan-s-compliance) or the [a href="https://www.springernature.com/gp/open-science/us-federal-agency-compliance"> NIH public access policy](https://www.springernature.com/gp/open-science/us-federal-agency-compliance)) then you should select the gold OA route, and we will direct you to the compliant route where possible. Because authors warrant under our subscription licensing terms that they haven't committed to licensing any version of their article under a licence inconsistent with the terms of our agreement – including the applicable embargo period – publication under the subscription model isn't suitable for authors whose funders require no embargo.

An online order form for reprints of your paper is available at [a href="https://www.nature.com/reprints/author-reprints.html">https://www.nature.com/reprints/author-reprints.html](https://www.nature.com/reprints/author-reprints.html). All co-authors, authors' institutions and authors' funding agencies can order reprints using the form appropriate to their geographical region.

With kind regards,

P.S. Click on the following link if you would like to recommend Nature Microbiology to your librarian
<http://www.nature.com/subscriptions/recommend.html#forms>

** Visit the Springer Nature Editorial and Publishing website at http://editorial-jobs.springernature.com?utm_source=ejP_NMicro_email&utm_medium=ejP_NMicro_email&utm_campaign=ejp_NMicro for more information about our career opportunities. If you have any questions please click [here](mailto:editorial.publishing.jobs@springernature.com). **

Open Access This Peer Review File is licensed under a Creative Commons Attribution 4.0 International License, which permits use, sharing, adaptation, distribution and reproduction in any medium or format, as long as you give appropriate credit to the original author(s) and the source, provide a link to the Creative Commons license, and indicate if changes were made. In cases where reviewers are anonymous, credit should be given to 'Anonymous Referee' and the source. The images or other third party material in this Peer Review File are included in the article's Creative Commons license, unless indicated otherwise in a credit line to the material. If material is not included in the article's Creative Commons license and your intended use is not permitted by statutory regulation or exceeds the permitted use, you will need to obtain permission directly from the copyright holder.

We thank the editor and the reviewers for their supportive and constructive comments on our manuscript. We have now performed additional experiments to address concerns that were raised by the reviewers and have revised the manuscript accordingly. All source data, their replicates, and uncropped gel images have been uploaded; the total number of main figures and extended data figures are six and ten, respectively, according to guidelines from *Nature Microbiology*. An article file with revisions made in Track Changes has also been uploaded. Our responses to specific comments raised by the reviewers are detailed below.

EDITOR COMMENTS

Thank you for your patience while your manuscript "A bacterial CARD-NLR immune system controls the release of gene transfer agents" was under peer-review at Nature Microbiology. It has now been seen by 5 referees, whose expertise and comments you will find at the end of this email. Although they find your work of some potential interest, they have raised a number of concerns that will need to be addressed before we can consider publication of the work in Nature Microbiology.

In particular, the referees noted that more work is needed to demonstrate that LypABC is a phage defence system and one referee suggested that a little more work might be needed to confirm whether it is an NLR-CARD system or perhaps another type of system as they suggest in their comments. The referees also note that it's important to demonstrate that LypABC activity is promoting transfer of the GTA as well. Then there are a number of comments that focus around building the evidence for LypABC activity through experiments that further analyse which domains are doing what, further explore the function of RogB and when it's repression might be alleviated. One of the referees suggested removing the analysis of RogB function. We have discussed this editorially, and would suggest that you keep this aspect of the work (since it gives further insight into the regulation of LypABC and the GTA) but use the previously published name for this factor for consistency with the literature. These are the critical points which need to be addressed in a revised manuscript. There are a few other points among the reviewer reports which should be straightforward and feasible to address.

Should further experimental data allow you to address these criticisms, we would be happy to look at a revised manuscript.

REVIEWER COMMENTS

Reviewer #1 (Remarks to the Author):

The manuscript by Banks et al. describes a mechanism which *Caulobacter crescentus* uses to initiate cell lysis and release Gene Transfer Agents (GTAs). GTAs are domesticated prophages that facilitate horizontal gene transfer. GTAs have been coopted from phages and here, the authors show that the cellular lysis initiation mechanisms have been coopted from antiphage systems. This work is an exciting next chapter in our understanding of how phage (and antiphage) genes have impacted the biology of bacteria.

Thank you very much for the supportive comments and excellent suggestions.

The authors discover the LypABC system, characterize its function primarily using genetics, and provide additional insights into transcriptional regulation. There are two major issues identified that should be addressed to improve the support for the conclusions, detailed below.

Major Comments

1. The connection of LypABC to CARD-NLR systems is provocative and a major point of emphasis in the manuscript. The sequence/structure comparisons should be more firmly supported. The authors draw the conclusion that LypABC evolved divergently from CARD-NLR systems. This is a challenging argument to make based on tools like DefenseFinder and FoldSeek. The question is whether the NTPase + CARD domains of LypB are more significantly related to the NTPase+CARD

domains in CARD-NLR systems than to other NTPase or CARD domains found in non-antiphage systems—or even found in different ORFs. DefenseFinder does not index non-immune proteins and simply realigns the immune pathway most related to the query, leading to confirmation bias. One could imagine a scenario where LypB is actually the parent of the CARD-NLR system, and that antiphage activity emerged as an adaptation later. Alternatively, perhaps LypB is a novel reformulation of a distantly related NTPase and a distantly related CARD that have, by chance, combined into a new formulation. At issue is whether this is convergent evolution of architecture or divergent evolution of sequence. The authors should (1) analyze the sequence relationships of LypB to CARD-NLR systems in detail and (2) discuss the limitations of the speculation on relationship to CARD-NLR. See also Minor Point 1 about nomenclature.

The reviewer raises a great question regarding the evolution and potential repurposing of the LypABC system. While we do claim that LypABC resembles a CARD-NLR anti-phage system (and have now performed further experiments and analysis to support this), we were careful not to make any phylogenetic/evolutionary claims that are unsupported by evidence. Moreover, in this revised manuscript, we have ensured that any speculation that LypABC may have been repurposed for GTA-mediated lysis is confined to the last part of the discussion only. We agree that the evolutionary history of LypABC is an interesting question and therefore, as the reviewer suggested, we have analysed the sequence relationships of LypB to CARD-NLR systems with a phylogenetic analysis. To do this, we built a maximum likelihood phylogenetic tree using LypB, each NLR ATPase predicted to be encoded as part of a CARD-NLR defence system by DefenseFinder, NLR proteins from bNACHT anti-phage defence systems, and NLR proteins that are not involved in anti-phage defence (representatives of the MalT and SWACOS families). Each of these NLR groups formed distinct clades within the tree, with LypB nested within the CARD-NLR clade (**Fig. 4c**). This indicates that LypB indeed belongs to the CARD-NLR family, however considering the current scarcity of experimentally validated CARD-NLR systems, we hesitate to speculate further regarding the question of a convergent/divergent evolutionary origin. We added the following text to the manuscript: “Moreover, AlphaFold3-generated models of LypA and LypB superimpose closely to their *L. enzymogenes* homologues (**Extended Data Fig. 3a-b**), and a phylogenetic tree built from the ATPase domains of CARD-NLR, bNACHT NLR, SWACOS, and MalT-family proteins placed LypB within a distinct CARD-NLR clade (**Fig. 4c**). These data suggest that LypABC belong to the CARD-NLR protein family.”

2. The authors should show a role for LypABC in a GTA functional assay. While LypABC is clearly required for lysis/cell death, and the microscopy/tomography supports a role for GTA particle dissemination, there is no experiment in the manuscript that demonstrates GTA-mediated gene transfer is impaired. Could the authors place a selectable marker on the chromosome and measure GTA-mediated generalized transduction of that marker into a naïve recipient? This should be greatly impaired by mutations in LypABC and could, in theory, show a phenotype for the enigmatic nuclease domain of LypC.

We have now performed the experiments to show that gene transfer activity is abolished in the absence of the *lypABC* operon (**Fig. 2e and Response to reviewers Fig. 1**). For this experiment, we used a xylose-inducible *gafYZ* overproducer strain (pBXMCS-6::P_{xyI}-*gafYZ*) as the donor since (1) this strain produces far more GTA-lysing cells (~70-80%) than a Δ *rogA* mutant and (2) GTA transduction has been validated for a *gafYZ* overproducer strain previously (Gozzi *et al.*, 2022). For the pBXMCS-6::P_{xyI}-*gafYZ* (GTA-on) strain, we observed transduction of a chromosomally encoded tetracycline resistance marker to the kanamycin-resistant recipient strain (mean transfer rate of 2.35×10^{-6} /cell). In contrast, neither the pBXMCS-6::P_{xyI}-empty vector control (GTA-off) nor the Δ *lypABC* pBXMCS-6::P_{xyI}-*gafYZ* (GTA-on, no lysis) donor strains were capable of DNA transfer to the recipient strain.

The results section has subsequently been expanded to include the following text: “To test whether the absence of *lypABC* - and thus cell lysis - abrogates GTA-mediated gene transfer, we performed a gene transfer assay, measuring transduction of a tetracycline resistance marker from different

donor strains to a kanamycin-resistant recipient strain. For this experiment, we used a xylose-inducible *gafYZ* overproducer strain (pBXMCS-6::P_{xyI}-*gafYZ*) as the donor since 1) this strain produces far more GTA-lysing cells (~70-80%) than a Δ rogA mutant and 2) GTA transduction has been measured for a *gafYZ* overproducer strain previously²⁵. In contrast to the lysis-competent P_{xyI}-*gafYZ* donor strain which had a transduction rate of $\sim 2.35 \times 10^{-6}$ /cell, the lysis-incompetent Δ lypABC P_{xyI}-*gafYZ* donor was incapable of gene transfer, equivalent to the GTA-off P_{xyI}-empty vector negative control donor strain (Fig. 2e). These data demonstrate that all three *lyp* genes are required for GTA-mediated host cell lysis and consequent transfer of DNA to recipient cells”.

We deleted the LypC endonuclease domain entirely, however, this protein truncation was unstable, hinting at a potential role for this domain in protein folding/stability (Extended Data Fig. 4c). We also combined individual H471A and N505A point mutations within a single strain, however, this double mutant phenocopied the single mutants, i.e. there was no effect on LypABC-mediated cell lysis (Fig. 4d). We agree that the function of the LypC endonuclease domain is enigmatic and warrants further investigation in a future manuscript.

Response to reviewers Fig. 1.

Gene transfer assay measuring GTA transduction of a chromosomally encoded tetracycline resistance cassette from *C. crescentus* donor strains into a kanamycin-resistant recipient strain. Donor strains contain the replicative plasmid pBXMCS-6 encoding either *gafYZ* (to activate GTA expression) or nothing (empty vector control). The transfer rate was calculated by dividing doubly antibiotic-resistant CFU/ml by the total recipient CFU/ml. Data show the mean \pm s.d from three independent experiments.

Minor Comments

1. Typically, the term NLR is reserved for proteins that encode “nucleotide-binding and leucine-rich repeats”. Here, the LRR is lacking and should be clarified in the text. A useful term that helps with this differentiation is NLR-related. Another point to consider is that the LypB system appears highly similar to mammalian APAF-1. This protein is a STAND NTPase (AP-ATPase) that is fused to a CARD domain. Because this protein lacks LRRs, it is not classified as an NLR. However, it appears more grossly similar to LypB due to its non-immune role in programmed cell death. This could also be included in the discussion.

We have clarified the fact that LypB is an NLR-like protein by adding the following sentence to the results section: “*This domain architecture is reminiscent of NLR-related proteins (such as APAF1⁴⁵) which may substitute the leucine-rich repeats of canonical NLRs for alternatives such as TPR or WD40 repeats⁴⁶.*”

Aaron Whiteley

Reviewer #2 (Remarks to the Author):

In this manuscript, the authors showed that a phage defence system composed of 3 proteins (LypABC) has been repurposed for cell lysis and ghost cell formation upon GTA activation in *Caulobacter crescentus*. Using bioinformatic tools and AlphaFold predictions, they found that LypABC belong to the CARD-NLR protein family. By using targeted mutagenesis, they inactivated the catalytic activity of the domains of the 3 Lyp proteins and tested their importance in ghost cell formation. Finally, they found a second transcriptional regulator (RogB) that represses expression of lypABC but which also has multiple other target genes *in vivo*.

The discovery of the LypABC system as inducing the lysis of *Caulobacter* cells is novel and very interesting from an evolutionary point of view. However, the characterization of the Lyp proteins remains somewhat limited here and there are still many open questions about this system. On the other hand, the description of RogB appears a little off topic in this manuscript. Of course not entirely since it represses the expression of lypABC but the transition from the incomplete characterisation of LypABC to the discovery of RogB is a little frustrating and also let many open questions about this repressor. In addition, CCNA_02755, called RogB by the authors, has already been characterized in another publication cited by the authors (McLaughlin *et al.*, 2023), in which the XRE transcription regulator was called CdxB. The CdxB-dependent regulon has already been described with ChIP-seq and RNA-seq data in this publication. Even a link between GTA activation and CdxB is described in this manuscript.

We thank the reviewer for the constructive comments. RogB/CdxB was discovered independently by both the Crosson group and ours. However, we chose not to co-publish our RogB/CdxB story because we wished to rigorously analyse the LypABC-RogB/CdxB connection. Our transcriptomic and ChIP-seq analyses are fully consistent with those reported in McLaughlin *et al.* Moreover, through many additional *in vitro* works, we solidified the finding that RogB regulates the expression lypABC, thereby regulating GTA-mediated host lysis. Our findings complement the McLaughlin *et al.*, work which primarily focuses on the link of CdxB to adhesin development and ϕ CbK infection.

Thus, for all these reasons, I would suggest to the authors to further characterise LypABC and focus only on this repurposed lytic system leaving CdxB (RogB) out of this story.

For reasons stated above and following the suggestion from the editor, we decided to retain the RogB-related aspects of our work. However, to maintain consistency with the literature, we have now renamed RogB to CdxB throughout this manuscript.

I have several questions and suggestions for the LypABC system.

- Why the role of the ATPase domain of LypA hasn't been addressed? At least a truncated variant lacking this domain could be used in the complementation assay?

As suggested by the reviewer, we have now constructed an additional complementation strain: $\Delta rogA \Delta lypA$ $xyIX::P_{xyI}-lypA$ (Δ ATPase domain)-FLAG, however this truncated variant was unstable as indicated by the anti-FLAG immunoblot (**Extended Data Fig. 4a**).

Notwithstanding, we examined the LypA ATPase domain more closely to identify potential ATP-contacting residues for mutagenesis. We identified T356, S360, and F361, which likely contact ATP (please see our response to Reviewer 3 for more detail) and substituted all three residues for alanine to create $xyIX::P_{xyI}-lypA$ (T356A, S360A, F361A)-FLAG. This variant was stably produced *in vivo* and microscopy analysis revealed that it failed to complement a $\Delta rogA \Delta lypA$ deletion mutant i.e. no cell lysis (**Fig. 4d**). Our results suggest that despite not encoding canonical Walker residues, the ATPase domain of LypA might still bind ATP and that this activity is required for LypABC-mediated cell lysis.

We have now expanded the Results section as follows: “*Deletion of the ATPase domain resulted in an unstable truncated protein (Extended Data Fig. 4a), therefore we examined the predicted ATP-binding pocket which contains a potential β -strand-loop- α -helix motif, characteristic of P-loop AAA+ ATPases. We identified two residues, T356 and S360, which are predicted to form hydrogen bond contacts with ATP phosphates, and F361, which is positioned parallel to the adenine base (Extended Data Fig. 6a). To test the importance of this putative pocket, we generated a lypA triple mutant (T356A, S360A, F361A) that was stably produced *in vivo* (Extended Data Fig. 4a) but failed to restore lysis in the $\Delta rogA \Delta lypA$ mutant (Fig. 4d). These data suggest that the divergent ATPase domain of LypA might bind ATP *in vivo*, and this activity is essential for LypABC-mediated cell lysis.*”

- Deleting the CarD domain makes LypA unstable and useless. But based on the AlphaFold3-Multimer model of the LypA_CarD:LypB complex, wouldn't it be possible to predict pairs of residues at the interface important for the interaction? Based on that, loss of interaction mutants could be first tested by BTH and then *in vivo* for cell lysis. Same for the homodimerization of the CarD domain. Mutations disrupting CarD homodimer formation should be tested. This would bring important information about the role played by the CarD domain in this process.

As suggested by the reviewer, we used AlphaFold3 to predict potential amino acid residues that mediate CARD-CARD and CARD-LypB interactions. We identified two residues within the LypA CARD domain, I7 and H45, that might form an interface with an opposing CARD domain or the N-terminal domain of LypB. We mutated each residue to an alanine (either individually or together), introduced these mutated alleles of *lypA* into a $\Delta rogA \Delta lypA$ deletion mutant, and tested for complementation. LypA (H45A)-FLAG was stably produced *in vivo* and $xyIX::P_{xyI}-lypA$ (H45A)-FLAG complemented the $\Delta rogA \Delta lypA$ deletion mutant, restoring cell lysis (**Response to reviewers Fig. 2**). In contrast, the $xyIX::P_{xyI}-lypA$ (I7A)-FLAG mutant and the $xyIX::P_{xyI}-lypA$ (I7A, H45A)-FLAG double mutant did not restore cell lysis, but neither of these protein variants was stably produced *in vivo* (**Response to reviewers Fig. 2**). Despite our effort here, it remains difficult to reliably assess the role of the CARD domain *in vivo*. Notwithstanding, we constructed an additional mutant to probe the CARD-LypB interaction. The N-terminal domain (NTD) of LypB, which was shown to interact with the CARD domain by bacterial two-hybrid experiments (**Extended Data Fig. 7d**), was truncated, and this allele was introduced into the $\Delta rogA \Delta lypB$ background. LypB (Δ NTD)-FLAG was stably produced *in vivo* yet $xyIX::P_{xyI}-lypB$ (Δ NTD)-FLAG failed to restore cell lysis, indicating that the NTD and potentially the NTD-CARD interaction is important for LypB function, and ultimately cell lysis (**Fig. 4d**). We have now expanded the Results section to report these data.

Response to reviewers Fig. 2.

a. Quantification of ghost cells within the population of $\Delta rogA \Delta lypA$ or $\Delta rogA \Delta lypC$ mutants complemented with either the wildtype *lyp* gene or a *lyp* gene mutant. Means \pm standard deviations of three independent repeats are shown. $n=400$ cells analysed per repeat. **b.** Anti-FLAG immunoblots of total cell lysates from indicated *C. crescentus* strains to assess levels of FLAG-tagged LypA variants. Coomassie-stained SDS-PAGE gels serve as loading controls. Immunoblots are representative of three independent experiments.

- It would be good to show multiple sequence alignments for each domain of the 3 proteins, not only nucleotide binding domain of LypB, to highlight the residues targeted as catalytic mutants.

This has been done and incorporated into Extended Data Fig. 5.

There is also a problem with the length of LypC since it is annotated as a 599 aa protein in NA1000 and as a 695 aa protein in CB15 while the authors talk about a protein of 662 aa. These numbers should be clarified.

Our initial attempts to complement the $\Delta rogA \Delta lypC$ deletion mutant with the 599 aa version of *lypC* failed. This prompted a closer inspection of the *lypC* reading frame, which later revealed a potential alternative start codon 189 bp upstream of the annotated one. Complementation using this longer 662 aa version of *lypC* was successful, validating the correct start codon. While we have not worked

with *lypABC* in CB15, a similar annotation issue likely explains the different numbering here also. We have now added a note to the Materials and Methods to bring this to the attention of readers.

- I agree with the conclusion that the endonuclease activity of LypC is not required for the lysis since the catalytic mutants H471A and N505A still led to lysis. But there are many examples of enzymatic domains repurposed for new functions. In that case, this domain could be important but independently of the endonuclease activity. A LypC variant truncated for the endonuclease domain should be tested.

We agree that the endonuclease domain could serve an alternative function. As also discussed in the response to Reviewer 1, we additionally constructed a *xyiX::P_{xyi}-lypC* (Δ endonuclease)-FLAG strain, however the truncated protein variant was not stably produced *in vivo* (**Extended Data Fig. 4c**), preventing us from drawing a firm conclusion regarding the function of this domain. Nevertheless, the instability of the LypC Δ endonuclease variant points at its possible contribution to protein folding/stability/structural integrity.

We have now expanded the Results section as follows: “*We also deleted the endonuclease domain entirely, however the resulting truncated variant was unstable in vivo which prevented functional assessment yet suggests a possible contribution towards LypC folding or stability (Extended Data Fig. 4c).*”

- Based on the following statement, considering that the trypsin-like serine peptidase protein and NLR-like protein of the L. enzymogenes anti-phage system interact, the authors tested the interaction between LypA, which harbours a trypsin-like serine peptidase, and LypB, which is a NLR-like protein harboring a ATPase domain. However LypC also harbours a trypsin-like serine peptidase domain and LypA has a predicted ATPase domain. The potential interactions between LypC and LypA or LypB should also be tested.

We agree, and we have now expanded our bacterial two-hybrid analysis to test all possible interactions between LypABC. Using full-length LypABC, we found that LypA and LypB each can self-interact, and that LypA and LypB interact with each other but not with LypC (**Extended Data Fig. 7c**). For a higher resolution analysis, we additionally performed bacterial two-hybrid assays for each individual domain of LypABC against every other domain (100 combinations in total) (**Extended Data Fig. 7d**). We again observed self-interaction for the LypA CARD domain and a direct interaction between the LypA CARD domain and the N-terminal domain of LypB. We further uncovered self-interaction for the N-terminal domain of LypB and a weaker self-interaction for the LypC endonuclease domain.

We have now expanded the Results section as follows: “*Bacterial two-hybrid assays confirmed that full-length LypA and LypB interact exclusively with each other and not with LypC (Extended Data Fig. 7c). Systematic pairwise testing of all possible LypA, LypB, and LypC domain combinations revealed weak self-interactions for the LypB N-terminal domain and LypC endonuclease domain and validated that the LypA-LypB interaction occurs specifically between the LypA CARD and the LypB N-terminal domain (Extended Data Fig. 7d).*”

The truncated variants as well as the catalytic mutants should be tested as well. This will allow to better understand the formation of the complex and to check whether the catalytic activities (ATPase and peptidase) are important for the complex formation or for the cell lysis (or both).

We reason that this question has been answered adequately by bacterial two-hybrid assays carried out with all individual domain combinations of LypABC, as described above. Furthermore, all catalytic residues are far from the predicted protein-protein interfaces i.e. unlikely to contribute to protein complex formation. Overall, the additional data support the conclusion that LypA and LypB can

interact together via an interface between the CARD domain of LypA and the N-terminal domain of LypB, and that LypC does not directly interact with LypA or LypB.

- Did the authors try to overexpress individually (with P_{cumate}) each of the *lyp* genes and compare the toxicity to the *lypABC* overexpression?

To address this, we constructed an additional strain, $P_{cumate}\text{-lypAB } (\Delta\text{lypC})$, which contains a markerless deletion of the *lypC* gene. No toxicity/cell lysis was observed upon cumate-induction (**Fig. 5b**), indicating that all three *lypABC* genes are required for cell lysis. This is consistent with our observation that deletion of any individual *lyp* gene is sufficient to abrogate GTA-mediated cell lysis (**Fig. 2**). We have added the following text to the manuscript: “*We next constructed a $\Delta\text{lypC } P_{cumate}\text{-lypAB}$ strain in which only *lypA* and *lypB* were overexpressed. No ghost cells were detected upon cumate-induction indicating that all three proteins are required for toxicity (Fig. 5b).*”

- LypABC is required for ghost cell formation (cell lysis) but not for DNA packaging. Hence $\Delta\text{rogA}\Delta\text{lypABC}$ cells die without lysing likely due to host genome digestion, suggesting that LypABC is critical for disseminating the capsid full of DNA. To firmly conclude that, the authors should measure the capacity of ΔlypABC mutant strains to transfer genetic material from donor to recipient in a way similar to the one described in Gozzi et al., 2022.

As detailed in the response to Reviewer 1, we have now performed gene transfer experiments which revealed that the ΔlypABC mutant was incapable of transferring DNA from donor to recipient cells (**Fig. 2e**).

- There are other gene candidates than *lypABC* identified in the Tn-seq screen (at least 41) but the authors do not describe them. It would be nice to have a few words about these other candidates. Is there any COG enrichment or genes belonging to the same pathway?

We have now performed clusters of orthologous groups (COG) analysis and also gene ontology (GO) analysis on the Tn-seq data. In the COG analysis, the top three most-enriched categories within the GTA-on ΔrogA mutant were: ‘ABC transporter family’, ‘post-translational modification/protein turnover/chaperones’, and ‘signal transduction mechanisms’. In the GO analysis, the top three categories were: ‘cobaltochelataase activity’, ‘cobalamin biosynthetic process’, and ‘efflux pump complex’. To avoid disrupting the flow of the manuscript, we have not added extensive text to the manuscript but instead incorporated the GO analysis into **Extended Data Fig. 2b** which is referenced within the manuscript.

- The presence of two trypsin-like serine peptidase domains, one in LypA and one in LypC is not discussed while both of them are required for ghost cell formation. This should be discussed.

In our current manuscript, we propose that LypABC remain inactive until triggered by a signal associated with GTA production. Beyond this, we further speculate that LypB may act as the primary sensor of this signal since NLRs and NLR-like proteins often function as sensor proteins. LypB may undergo a conformational change and bind to LypA, triggering the trypsin-like peptidase activity of LypA which might cleave and activate LypC. The separate trypsin-like peptidase activity of LypC might then proteolytically activate a downstream lysis pathway. Since this speculative model currently lacks sufficient experimental evidence, we refrained from including it in the current manuscript and intend to test this hypothesis more rigorously in future work. We hope the reviewer agrees.

Then, I have a several question about the GTA mechanism.

- Why GTA is required for ghost cells formation in ΔrogA ?

This is a great question. As discussed in the manuscript, we have not yet solved the complete mechanism by which LypABC become activated and trigger cell lysis. We propose that component(s) encoded within the core GTA cluster may be required to trigger LypABC activation - either directly, e.g. sensing of GTA particle production, or indirectly, e.g. sensing of GTA-induced cellular stresses. This might explain the absence of cell lysis in the $\Delta rogA \Delta GTA$ mutant.

- If GTA is required for ghost cell formation, why LypABC is toxic and leads to in WT cells where GTA is repressed?

Did the authors test the overexpression of lypABC in $\Delta GTA \Delta rogA$? Is there any difference when compared to $\Delta rogA$ single mutant?

Overexpression of *lypABC* from a very strong P_{cumate} promoter likely results in autoactivation of the LypABC system. This has been reported for plant NLRs in which overexpression of the NLR or mutations within the MHD motif of the NLR protein increase the proportion of active-state NLR proteins. We reason that overexpression of *lypABC*, and thus an elevated level of auto-active LypABC, bypassed the requirement for an upstream LypABC-activating trigger. Auto-active LypABC could thus cause cell lysis independently of the production of GTA particles.

As suggested by the reviewer, we further investigated this by constructing two additional strains: (1) ΔGTA cluster P_{cumate} -*lypABC* and (2) $\Delta rogA \Delta GTA$ cluster P_{cumate} -*lypABC*. Upon cumate-induction, we observed a similar proportion of ghost cells in both strains compared to the P_{cumate} -*lypABC* strain (in an otherwise wild-type background) (**Fig. 5b**). This provides further evidence that a potentially auto-active LypABC trigger cell lysis in the absence of stimuli that involves GTA particle components. We have now expanded the Results section as follows “*To test whether GTA core cluster components are required for cell lysis, we deleted the core GTA gene cluster in both P_{cumate} -*lypABC* and $\Delta rogA$ P_{cumate} -*lypABC* backgrounds. Cumate-induced toxicity was not suppressed in either strain, resulting in high proportions of ghost cells ($93.3 \pm 4.9\%$ and $90.0 \pm 5.0\%$ ghost cells, respectively, **Fig. 5b**).”*

- The authors state (L329-332): Since LypABC are produced in GTA-off wildtype cells, we reason that the system remains autoinhibited until GTA activation occurs to prevent untimely and unproductive host cell lysis. On which data is based this statement that LypABC are produced in WT where RogB represses its expression?

Our RNA-seq data indicate that *lypABC* are transcribed at a low basal level even in wild-type cells and are upregulated in the $\Delta rogB$ mutant (by 9.8-fold, 6.5-fold, and 8.6-fold, respectively). The normalised counts for *lypABC* are shown below:

Gene	wildtype			$\Delta rogB$		
	Repeat 1	Repeat 2	Repeat 3	Repeat 1	Repeat 2	Repeat 3
lypA	37.3	50.9	56.3	496.0	450.1	452.5
lypB	32.3	29.5	47.1	203.2	247.2	241.1
lypC	88.8	65.6	81.4	648.9	743.0	681.7

Related to the previous questions, how LypABC can be toxic in WT if GTA is off?

Please see our answers above regarding overexpression of *lypABC* which likely increases the level of auto-active LypABC complexes.

Finally I have a few questions/comments on the transcriptional regulation

- To avoid confusion, RogB should be called CdxB as in McLaughlin et al., 2023. A deep comparison with the transcriptomic data described in McLaughlin et al., 2023 should be done to keep at best only the original information.

We agree, and we have now renamed RogB to CdxB for consistency with published literature. Our ChIP-seq data are very similar to those presented in McLaughlin *et al.* and we identified the same consensus binding sequence for RogB (please see response to Reviewer 4). The RNA-seq data that we presented is slightly different since McLaughlin *et al.* compared a quadruple mutant ($\Delta rtrA \Delta rtrB \Delta cdxA \Delta cdxB$) to the wildtype strain, whereas in our study, we directly compare a single $\Delta rogB$ (*cdxB*) mutant to wildtype cells.

- There are several transcriptomic data (RNA-seq and ChIP-seq) about the RogB repressor. However, except *lypABC* and *gafYZ*, the authors do not describe at all the RogB regulon. Is there any COG enrichment in this regulon that might help understanding in which conditions RogB is inactivated? The authors could provide some further analysis of this regulon.

RogB has many binding sites on the *C. crescentus* genome and therefore likely functions as a global transcriptional regulator. Due to the large regulon, we chose to focus on the RogB targets of *gafYZ* and *lypABC* since these genes are directly linked to GTA function. As requested, we have now performed another COG (and GO) analysis, this time on the $\Delta rogB$ vs wildtype RNA-seq dataset. In the COG analysis, the top three most-enriched categories within the $\Delta rogB$ mutant were: 'function unknown', 'replication, recombination and repair', and 'intracellular trafficking, secretion and vesicular transport'. In the GO analysis, the top three categories were: 'endonuclease activity', 'protein secretion', and 'acyl-phosphate glycerol-3-phosphate acyltransferase activity'. Inspection of the COG and GO analysis did not immediately reveal clear insights into how RogB might be inactivated. However, we have incorporated the GO analysis into **Supplementary Fig. 2c** which is referenced within the manuscript.

- We easily understand why GTA has to be tightly repressed given the toxicity of GTA and Lyp proteins. However, the system has to be de-repressed in some conditions. Based on previous publication, entry to stationary phase increased GTA production and DNA transfer. The authors could therefore check whether the protein levels of RogA and/or RogB decrease at this growth stage. If so, a proteolytic mechanism could be investigated. Alternatively, RogB could be inactivated by protein:protein interactions, so the authors could look for protein interacting with RogB.

Are there other conditions known to induce GTA? If so, similar experiments should be done in the same conditions.

We agree that the GTA system must be de-repressed under certain environmental condition(s), however, we have yet to identify the specific stimuli that lead to de-repression – a non-trivial task. Entry to stationary phase is necessary but not sufficient for GTA activation. Nevertheless, as the reviewer suggests, we performed immunoblots to determine whether RogA/RogB levels differ between exponential phase and stationary phase. We observed that the levels of both proteins were in fact higher in stationary phase (**Response to reviewers Fig. 3**). Given that the stimuli required for GTA activation are still unknown and thus not supplied in this experiment, it is perhaps unsurprising that repressor levels do not decrease in stationary phase. Regarding potential RogB inactivation through protein-protein interactions, we agree this is formally possible, though it is only one of many plausible scenarios. Unfortunately, we cannot address this open-ended experiment satisfactorily within the revision timeline set by the editor.

Response to reviewers Fig. 3. The production of RogA and RogB increases in stationary phase. Immunoblots of total cell lysates from *C. crescentus* to probe the levels of either RogA (**a**) or RogB (**b**) during log phase and stationary phase. A monoclonal anti-FLAG antibody was used to detect *rogA-flag*, whereas RogB was detected using polyclonal antisera. Separate Coomassie-stained SDS-PAGE gels serve as loading controls. Immunoblots are representative of three independent experiments.

Reviewer #3 (Remarks to the Author):

Recently discovered gene transfer agent in *Caulobacter crescentus* (hereafter, CcGTA) is not yet as well understood as its counterpart in another alphaproteobacterium, *Rhodobacter capsulatus* (RcGTA). In this manuscript, Banks et al. identify and characterize genes and proteins that are required for the release of CcGTA particles. Their findings are exciting for two reasons. First, the CcGTA release mechanism is very different from that of RcGTA (which uses holin-endolysin system that is typically employed by bona fide phages.) Second, the CcGTA system appears to be co-opted from a phage defense system, which is an unexpected and novel finding. Thus, I think the manuscript would be of interest not only to “GTA enthusiasts”, but to a broader audience of microbiologists and evolutionary biologists. I support publication of this work in *Nature Microbiology*.

Thank you very much for your support.

Overall, the manuscript is well-written and easy to follow, with an informative introduction and the results that are logically described and augmented by well-organized figures. Although, I think the authors could broaden and strengthen their Discussion by highlighting the importance of their findings to more general (i.e., beyond GTA particle release) understanding of how GTAs evolved from their viral ancestors. Because the CcGTA core GTA cluster is indisputably homologous to RcGTA head-tail cluster (first observed in Shakya et al., 2017; PMID 29250433), they must have shared a common viral ancestor. But, clearly, *Rhodobacter* and *Caulobacter* diverged in the “solution” for GTA particle release. I think this is worth mentioning in the Discussion (but I leave this for the authors to decide).

This is an excellent suggestion and we have added this point and the associated reference to the discussion as follows: “*The R. capsulatus and C. crescentus GTA systems appear to have evolved from a common α -proteobacterial prophage ancestor⁶⁶, and retain several shared features including homology within core cluster proteins and a similar activation factor (GafA in *R. capsulatus* being homologous to GafYZ in *C. crescentus*). Given these similarities, it is intriguing that *C. crescentus* and *R. capsulatus* have nevertheless evolved diverging solutions for GTA particle release.*”

Additional requests for a few minor modifications:

Lines 97-99: please list % of the cells, to be consistent with the previous sentence that list the data. It is not possible to extract the % from Figure 1B itself due to small numbers.

Done.

Line 99: there should be a comma before “i.e.”

Corrected.

Lines 196-197: “While structural modelling of LypA with ATP and Mg²⁺ by AlphaFold3 suggested potential ATP-binding ability (Figure S6A), canonical Walker A/B motifs are absent”. Could this putative ATPase domain contain ATP-grasp fold (the only other known ATP-binding fold) instead of P-loop? I believe this is possible to check bioinformatically.

Structural homology searches with the LypA ATPase domain consistently returned AAA+ ATPases (e.g., Orc1-like ATPases) hits rather than ATP-grasp fold proteins. Moreover, a model of the LypA ATPase domain aligned poorly with ATP-grasp fold enzymes such as D-Ala–D-Ala ligase and biotin carboxylase, in comparison to strong alignments with P-loop ATPases.

To assess whether LypA adopts a P-loop-like configuration, we re-examined the AlphaFold3 model complexed with ATP and Mg²⁺. The predicted nucleotide-binding pocket exhibits a β -strand→loop→ α -helix topology, consistent with the canonical arrangement of the P-loop motif. Although the loop is not glycine-rich and lacks the canonical Walker residues, we identified a

threonine (T356) that could form a polar contact with an ATP phosphate, as well as a serine (S360) in the adjacent α -helix that may also contact the phosphate group (**Extended Data Fig. 6a**). A phenylalanine (F361) within the same helix is positioned in parallel to the adenine base and may aid in nucleotide positioning.

To test whether these residues are functionally important, we generated a triple point mutant ($xy/X::P_{xy}/lypA$ (T356A, S360A, F361A)-FLAG). The protein variant was stably produced *in vivo* and microscopy analysis revealed that it failed to complement a $\Delta rogA\Delta lypA$ deletion mutant i.e. was unable to restore cell lysis (**Fig. 4d**). Collectively, these data suggest that LypA employs a highly divergent P-loop/AAA+ ATPase architecture to bind ATP, and that this activity is important for GTA particle release. We have expanded the Results accordingly to report these data.

Lines 218-219: The reference to Figure 8B is incorrect here, not sure which figure the authors wanted to refer to here.

Corrected.

Line 227: I believe "CARDS" should be "CARDS".

Corrected.

Lines 603-608: This section should provide more details about parameters and databases used in searches. Specifically,

a) Please list database(s) against which BLASTP searches were carried out. List version of the BLAST program used. If the search was carried out via web, date of search should be listed, as databases change over time. Default parameters of the BLAST programs change over time, so it would be preferable for reproducibility if the authors would list the parameters such as E-value cutoff, substitution matrix and low complexity filtering. Insert reference for the BLAST program (PMID: 9254694).

b) For FoldSeek, please list databases and search mode used

c) For InterProScan, please provide dates of searches as their databases change over time.

d) For HHPred, please list database(s) and parameters used.

We have now added further information to the methods.

Supplementary Figures and Tables:

a) Supplementary Figure S12 should be referenced in the text (it is currently not).

b) Supplementary Figure S13 is only referenced in the legends of the Figures 6 and 7. Ideally, it should be referenced in the text.

c) Supplementary Materials and Methods should be referenced in the text (they are currently not).

d) Tables S1-S5 should have table caption as the top line in the Excel spreadsheet, to improve readability.

All done.

Reviewer #4 (Remarks to the Author):

The paper by Banks et al., addresses the mechanism of regulation of LypABC, to study how bacterial defence systems are involved in the release of GTAs. The authors used genetic screenings to identify bacterial immunity operons involved in GTA-mediated host cell lysis in *C. crescentus* and validated *lypABC* as a crucial operon regulating the GTA particle release. LypABC is a homologue of caspase recruitment domain-nucleotide binding leucine-rich repeat (CARD-NLR) anti-phage defence systems that trigger abortive infection. They also showed how the overexpression of LypABC is highly toxic which together with other observations was interpreted as a putative immunity-related activity.

This is a solid and well written manuscript that deconstructs different aspects of the regulation of GTAs. The main criticism I have is that lypABC was not validated as a defence system and while a homologue to other CARD-NLR, lypABC is actually non-canonical with an effector that seems non-essential for the phenotype; without experimental validation the work feels lacking. Given the functional variety and plasticity of these mobile operons it could still be plausible that lypABC is just another functional cog in the GTA machine...

We thank the reviewer for their positive comments and feedback on our manuscript. We agree that it is not possible to validate LypABC as a defence system without an experimental assay demonstrating that the genes confer protection against phage infection. For this reason, we claim only that LypABC resemble an anti-phage defence system and have carried out several additional analyses and experiments (detailed in our response to other reviewer points) to strengthen this conclusion. We have also now removed any reference to LypABC potentially being repurposed as a lysis system within the Introduction/Results text and confined this speculative suggestion to the last part of the discussion.

Main comments:

The FoldSeek searches that are shown also involve AF predictions, I was curious to know what were the results against the PDB?

For LypA, top PDB matches included the serine protease MucD from *Pseudomonas syringae* (8K2Y; score 324, 19% identity (ID)), HtrA1 from humans (3TJN; 317, 19% ID), and DegS from *E. coli* (2QF0; 202, 17.5% ID).

For LypB, the highest-scoring hits were the NLR MalT from *E. coli* (8BOB; 207, 17.1% ID), the AAA ATPase VcHTnsC from *Vibrio cholerae* (7UFI; 188, 14.3% ID), and the ATPase ORC2 from *Aeropyrum pernix* (1W5S; 171, 12.3% ID).

For LypC, top matches were the serine proteases SplA from *Staphylococcus aureus* (2W7S; 281, 21.5% ID) and Rv3671c from *Mycobacterium tuberculosis* (3K6Z; 269, 16.6% ID), and the exonuclease ExoG from human (6IID; 273, 21.2% ID).

These results from searches against an experimentally validated structure database are consistent with the previous Foldseek searches against the AlphaFold database, further strengthening the conclusion that LypA indeed contains a trypsin-like peptidase domain, LypB contains an NLR-like AAA+ ATPase domain, and LypC contains both trypsin-like peptidase and endonuclease domains.

Page 6: Given that the active site is non-canonical it would be helpful to provide in Fig. 4 of S6 a detailed view of the predicted ATP binding mode and its coordination, how are the base and 3P coordinated and how does it compares with known binding modes, are there obvious catalytic residues?

This now forms **Extended Data Fig. 6a**. Please see our response to Reviewer 3 on the same point.

As part of the characterization of LypC, the protease domain was inactivated by replacing a catalytic Ser which correlates with the loss of lytic function, however when this was tried for the endonuclease domain the single point substitutions were not sufficient to change the phenotype. For some enzymes active site residues are redundant and some single substitutions are not enough to switch off the activity. So would it be possible to validate the effect of the mutations in vitro? Perhaps combine several? In the same way, some systems repurpose enzymatic domains as regulatory domains, given that in *C. endophyticus* the endonuclease is not present, could the authors evaluate the effect of a truncate. These are necessary controls that need to be provided to support the claim that the cell lysis is endonuclease independent.

We thank the reviewer for these suggestions. To provide more clarity regarding the function of the LypC endonuclease domain, we generated a strain that combined both the H471A and N505A catalytic substitutions, however, this double mutant phenocopied the single mutants, complementing a $\Delta rogA\Delta lypC$ mutant to restore cell lysis (**Fig. 4d**). Thus, we are confident that the endonuclease activity of LypC is not required for cell lysis. We additionally generated an endonuclease domain truncated variant (also suggested by Reviewer 2) but it was unstable *in vivo*, preventing us from drawing a firm conclusion. Nevertheless, the instability of the LypC Δ endonuclease variant suggests a possible contribution towards protein folding/stability/structural integrity. These data were added to the Results section.

One of the most interesting findings of the paper is the discovery that RogB is a transcriptional regulator of the system that also controls the *gafYZ* operon. The authors use SPR as a qualitative method to show the specific binding to the *lypABC* promoter, however very little is then done or pursued in terms of the actual characterization of the interaction of RogB and DNA. 1) Is there a reason why the K_d for DNA could not be measured with SPR?

We have now performed additional SPR experiments to determine the K_D of RogB binding to the promoters of both *lypABC* and *gafYZ* (28 nM and 9 nM, respectively) (**Extended Data Fig. 9b-c**). These quantitative measurements extend the earlier work of McLaughlin *et al.*, (which mainly employed genetic analyses but not biochemical characterisation).

2) What type of DNA sequence is RogB recognizing (inverted repeats, palindromes)?

To address this question, we used the top 100 most enriched peaks from our RogB ChIP-seq dataset to perform a MEME-ChIP analysis using MEME Suite. We identified a consensus binding motif that is non-palindromic, containing one conserved half-site and a second weakly-conserved degenerate half-site (now added to **Extended Data Fig. 9d**). This is also consistent with the binding motif identified in McLaughlin *et al.* 2023.

3) Is there a sequence motif shared between the promoter of *lypABC* and *gafYZ*? And if different, given the dual repressor hypothesis suggested in the manuscript, are such differences translated in different affinities? Given that RogB can be produced, even simple EMSA experiments can provide significant insights to the regulatory process.

The binding affinities determined by SPR might suggest that RogB binds more tightly to the promoter of *gafYZ* than the promoter of *lypABC* (28 nM vs 9 nM), but the difference is just three-fold i.e. relatively small. Guided by the MEME-generated consensus binding motif, we searched both promoter regions but were unable to pinpoint the exact core binding site within the 50-bp DNA duplex used for SPR analyses. This is perhaps not surprising given that the MEME consensus sequence is already degenerate (please see also **Extended Data Fig. 9d**).

Minor issues:

Ref 28 can now be updated, the work was published in May 2025.

The paper was duplicated in the references; we have now removed the preprint version, keeping only the published paper.

In page 3 at the end of the introduction the authors state: “AlphaFold-Multimer and bacterial two-hybrid assays indicate that LypA and LypB form a co-complex mediated by the CARD-like domain of LypA”. This should be rephrased, AF-multimer is not an experimental method and should not be used as an indication of a complex-formation no matter how high the cofolding scores are.

We agree that AF-Multimer only provides a prediction of protein-protein interaction, and we have now further validated the LypA-LypB interaction through extensive bacterial two-hybrid experiments (**Extended Data Fig. 7c-d**). This has now been described in detail in the Results section. For the

sentence within the introduction, we simplified it to: *“Bacterial two-hybrid assays indicate that LypA and LypB form a co-complex mediated by the CARD-like domain of LypA”*.

The same can be say about the final sentence of the introduction: “In summary, we present evidence that immune systems are versatile and can perform new biological functions that are beneficial, rather than antagonistic, to MGEs.” This is a very general and strong statement, however the authors didn’t actually show that lypABC is involved in defence, so without experimental validation this needs to be tone down, the paper has other significant achievements that are supported by experiments...

We have re-written the sentence to tone down the conclusion: *“In summary, we have identified a CARD-NLR-like system that may benefit MGEs and promote horizontal gene transfer.”*